# Targeting myoferlin in ER/Golgi vesicle trafficking reprograms pancreatic cancer-associated fibroblasts

Raphaël Peiffer [1,7], Emilie Laverdeur [1], Anthoula Gaigneaux [2], Yasmine Boumahd[1], Charlotte Gullo[1], Gilles Rademaker[1,8], Rebekah Crake[3], Arnaud Lavergne [4], Naïma Maloujahmoum[1], Ferman Agirman[1], Michael Herfs [5], Atsushi Masamune [6], Elisabeth Letellier [2], Akeila Bellahcène [1] & Olivier Peulen [1✉]

## Abstract

**Pancreatic adenocarcinoma (PAAD) cells exploit vesicle trafficking proteins, such as myoferlin (encoded by MYOF), to fuel tumor aggressiveness, yet the presence and function of myoferlin-dependent vesicles in cancer-associated fibroblasts (CAFs) remain unknown. By combining PAAD whole-tumor and single-cell transcriptomic analyses with immunohistochemistry and 2D/3D in vitro models, we link stromal myoferlin to tumor aggressiveness. We identify CAF-specific functions of myoferlin, as *MYOF*-depleted CAFs exhibit reduced activity and impaired extracellular matrix (ECM) production. Analysis of intracellular vesicles shows that myoferlin depletion results in a TGFß-receptor 1 (TGFBR1) trafficking blockade at the ER/Golgi interface upon myoferlin depletion, leading to altered TGFBR1 activation, impaired signal transduction, loss of ECM production and reduced CAF contractility. Both genetic depletion of myoferlin in the murine tumor stroma and the pharmacological targeting of myoferlin alike reduced tumor desmoplasia in orthotopic mouse model of pancreatic ductal adenocarcinoma. Based on these findings, we propose TGFBR1 trafficking as a potential target for reprogramming CAFs, controlling desmoplasia, and tackling these aggressive features in pancreatic cancer.**

**Keywords** Pancreatic Cancer; Cancer-associated Fibroblasts; Myoferlin; Desmoplasia; COPII-vesicle Trafficking
**Subject Categories** Cancer; Membranes & Trafficking; Signal Transduction

## Introduction

Despite decades of research, pancreatic adenocarcinoma (PAAD) remains an extremely aggressive cancer and is the most common malignant neoplasm of the pancreas (Conroy et al, 2023). The mortality of PAAD is strikingly high, with a five-year survival rate of 13%, according to the SEER database. The presence of an extensive tumor microenvironment (TME), accounting for up to 80% of PAAD tumor volume, contributes significantly to tumor aggressiveness by supporting growth, invasion, metastasis, and therapy resistance (Murakami et al, 2019). Targeting the TME has seen major advancements in the past 10 years, as immunotherapy is increasingly implemented in clinical practice. However, immunotherapy has not proven efficient in all cancer types, considering that immune cells can sometimes be a minority of stromal cells when compared to other cell types (Valkenburg et al, 2018). PAAD is a prime example for a cancer type that resists to newly proposed therapies (Sarantis et al, 2020), as the pancreatic TME is considered immune cell exclusive but heavily populated by cancer-associated fibroblasts (CAFs) (Karamitopoulou, 2019).

The landscape of pancreatic CAFs is evolving rapidly and presents high temporal and spatial heterogeneity (Pereira et al, 2019). Multiple studies have agreed on at least two functionally distinct human CAF subtypes: (1) myofibroblast-like CAFs (myCAFs) and (2) inflammatory CAFs (iCAFs) (Caligiuri and Tuveson, 2023), both of which can be modelled in vitro using pancreatic stellate cells (PSCs) (Öhlund et al, 2017). In a general context, myCAFs promote ECM production and desmoplasia, whereas iCAFs participate in immune modulation, altogether promoting immune cell exclusion and drug resistance (Sahai et al, 2020). CAFs represent one of the most promising stromal targets in PAAD, because of their excessive abundance and their implication in nearly every aspect of tumor biology.

Intracellular vesicle trafficking is an essential molecular route for the transport of membrane proteins and important for plasma membrane maintenance of all cells (Mellman and Nelson, 2008). Neoplastic cells alter fundamental vesicle trafficking routes for the delivery of specific membrane proteins to the cell surface, promoting tumor cell transformation and migration (Goldenring, 2013). During tumorigenesis, vesicle trafficking proteins are upregulated to support cellular flexibility and adaptation, necessary

[1]Metastasis Research Laboratory, GIGA Cancer, University of Liège, Liège, Belgium. [2]Molecular Disease Mechanisms Group, University of Luxembourg, Belvaux, Luxembourg. [3]Laboratory of Tumor Biology and Development, GIGA Cancer, University of Liège, Liège, Belgium. [4]GIGA Bioinformatics Platform, University of Liège, Liège, Belgium. [5]Laboratory of Experimental Pathology, GIGA Cancer, University of Liège, Liège, Belgium. [6]Division of Gastroenterology, Tohoku University Graduate School of Medicine, Sendai, Japan. [7]Present address: Personalized Oncology Division, The Walter and Eliza Hall Institute of Medical Research, Parkville, VIC, Australia. [8]Present address: Department of Anatomy, University of California, San Francisco, CA, USA. ✉E-mail: olivier.peulen@uliege.be

for tumor progression and dissemination in various cancer types, including PAAD (Yang et al, 2011; Hou et al, 2008; Gupta et al, 2020). One vesicle trafficking protein that is overexpressed in PAAD is myoferlin (Turtoi et al, 2011), a membrane protein physiologically involved in myoblast fusion and intracellular trafficking of membrane receptors such as IGF1R (Demonbreun et al, 2010; Doherty et al, 2005). Several pro-oncogenic properties of myoferlin in PAAD tumor cells have been described previously, including the implication of myoferlin in lysosome integrity and metabolic flexibility (Gupta et al, 2021; Anania et al, 2024; Rademaker et al, 2018).

To date, our knowledge of intracellular vesicle trafficking in PAAD is limited to cancer cells, and a potential importance of vesicle trafficking proteins, such as myoferlin, in pancreatic stromal cells remains to be elucidated. This work aims to fill this knowledge gap and to investigate the importance of myoferlin-mediated intracellular vesicle trafficking in pancreatic CAFs.

# Results

## Stromal myoferlin drives tumor aggressiveness is linked to desmoplasia in pancreatic cancer patients

Myoferlin is recognized as an emerging oncoprotein (Gupta et al, 2021; Blomme et al, 2017; Rademaker et al, 2019), highlighted by the overexpression of myoferlin in several cancer types, including PAAD with the highest expression and the greatest overexpression compared to healthy pancreas (Fig. EV1A). While several studies describe myoferlin functions in pancreatic cancer cells (Gupta et al, 2021; Rademaker et al, 2022, 2018; Anania et al, 2020), the stromal role of myoferlin remains unknown. To understand the potential stromal function of myoferlin in PAAD, we first evaluated *MYOF* mRNA expression in patient samples (TCGA PAAD cohort, $n = 178$). Patients with confirmed ductal phenotype ($n = 146$) were segregated into low-, intermediate-, and high *MYOF* patients, according to their respective *MYOF* z-score (Fig. 1A). *MYOF*high patients showed significantly shorter median overall survival (OS, 12.95 months) than intermediate- (19.62 months) and *MYOF*low patients (20.61 months), while demographic and clinical attributes including TNM staging were equivalent between patient groups (Figs. 1B and EV1B).

To link PAAD stromal features to myoferlin expression in an unbiased approach, we performed gene set enrichment analysis on differentially expressed genes between *MYOF*high and *MYOF*low patients. We found among the most enriched gene sets in *MYOF*high patients several fibrosis- and ECM-related gene sets (Figs. 1C and EV1C). Next, to investigate tumor cellularity as potential bias during GSEA analysis, we assessed tumor cellularity according to *MYOF* expression since tumor samples are heterogeneous and constituted by multiple different cell types. Using transcriptomic stromal and immune gene signatures (ESTIMATE) (Yoshihara et al, 2013), we found that *MYOF* expression did not affect stromal and immune cell content in TCGA samples (Figs. 1D and EV1D). Similar results were obtained when assessing ABSOLUTE (Carter et al, 2012) tumor purity across groups (Fig. 1E). However, in agreement with increased expression of ECM-related genes, *MYOF*high tumors displayed higher expression of hypoxia related genes (Buffa et al, 2010) than intermediate- and *MYOF*low tumors

(Fig. 1F). Further, with regard to inter-patient heterogeneity, high stromal ECM abundance in PAAD tumors is described as characteristic for activated stroma subtypes and linked to impaired survival. Using transcriptomic data (Moffitt et al, 2015) we assessed myoferlin expression in virtually microdissected human PAAD tumors and found that tumors with activated stroma (i.e. increased collagen and activated fibroblast gene expression) had higher *MYOF* expression than normal or low stroma subtypes (Fig. 1G). Of note, *MYOF*high patients contained higher proportions of quasi-mesenchymal/basal/squamous subtypes compared to *MYOF*low patients (Fig. EV1E–G). This finding fits with a previously suggested role of myoferlin in tumor differentiation (Wang et al, 2013). Importantly, all major PAAD-characteristic matrisome components (Tian et al, 2019), namely *COL1A1*, *COL1A2*, *COL3A1*, *COL6A3* and *FN1* were higher expressed in intermediate- and *MYOF*high tumors compared to *MYOF*low tumors (Fig. 1H). Altogether, this transcriptomic data suggests a link between *MYOF* expression, stromal activation, and tumor fibrosis in PAAD patients. We further extended these transcriptomic findings via myoferlin IHC and Masson trichrome staining of tumor sections from an internal PAAD patient cohort. Using 99 subTME regions with variations in collagen content (loose stroma vs dense stroma), we found that stromal myoferlin abundance significantly correlated with collagen deposition, as subTME regions with high stromal myoferlin abundance presented denser fibrotic tissue (Fig. EV1H,I).

Finally, we aimed to elucidate specifically the impact of stromal myoferlin on PAAD patient survival. Using our internal PAAD patient cohort, we quantified stromal myoferlin staining ranging from weak to intense and segregated patients in MYOFhigh stroma vs MYOFlow stroma (Fig. 1I), while demographic and clinical parameters were homogenous between groups (Fig. EV1J). Patients with a predominant MYOFlow stroma showed nearly doubled median overall survival (31.86 months) compared to patients with MYOFhigh stroma (16.09 months) (Fig. 1J), supporting the implication of myoferlin in tumor fibrosis and potential tumor-promoting functions of the *MYOF*high TME. We additionally quantified myoferlin staining in cancer cells on PAAD sections (Fig. 1K). Strikingly, we were unable to find an association of cancer cell myoferlin with overall patient survival (Fig. 1L). This was further supported by the absence of a correlation between stromal myoferlin scores and cancer cell myoferlin scores (Fig. EV1K).

Collectively, these results link stromal myoferlin expression to increased tumor fibrosis (i.e., tumor desmoplasia) and suggest stromal myoferlin rather than cancer cell myoferlin as driver for tumor aggressiveness in PAAD patients.

## Myoferlin is expressed pleiotropically in pancreatic CAFs

Fibrotic matrisome components (collagens, glycoproteins, proteoglycans) have been shown to be predominantly secreted by stromal cells (Tian et al, 2019), as tumor fibrosis is heavily driven by activated CAFs in the tumor microenvironment (Peiffer et al, 2023). Owing to the correlation between *MYOF* expression and tumor fibrosis in PAAD tumors, we used human PAAD scRNAseq data to assess the expression of *MYOF* across stromal cells, including CAFs, immune- and endothelial cells (Fig. 2A). Upon broad cell type clustering (Fig. EV2A), we found that *MYOF* was only poorly expressed in immune cells (B-, T-, NK-, myeloid- and mast cells), while fibroblasts, epithelial cells and endothelial cells

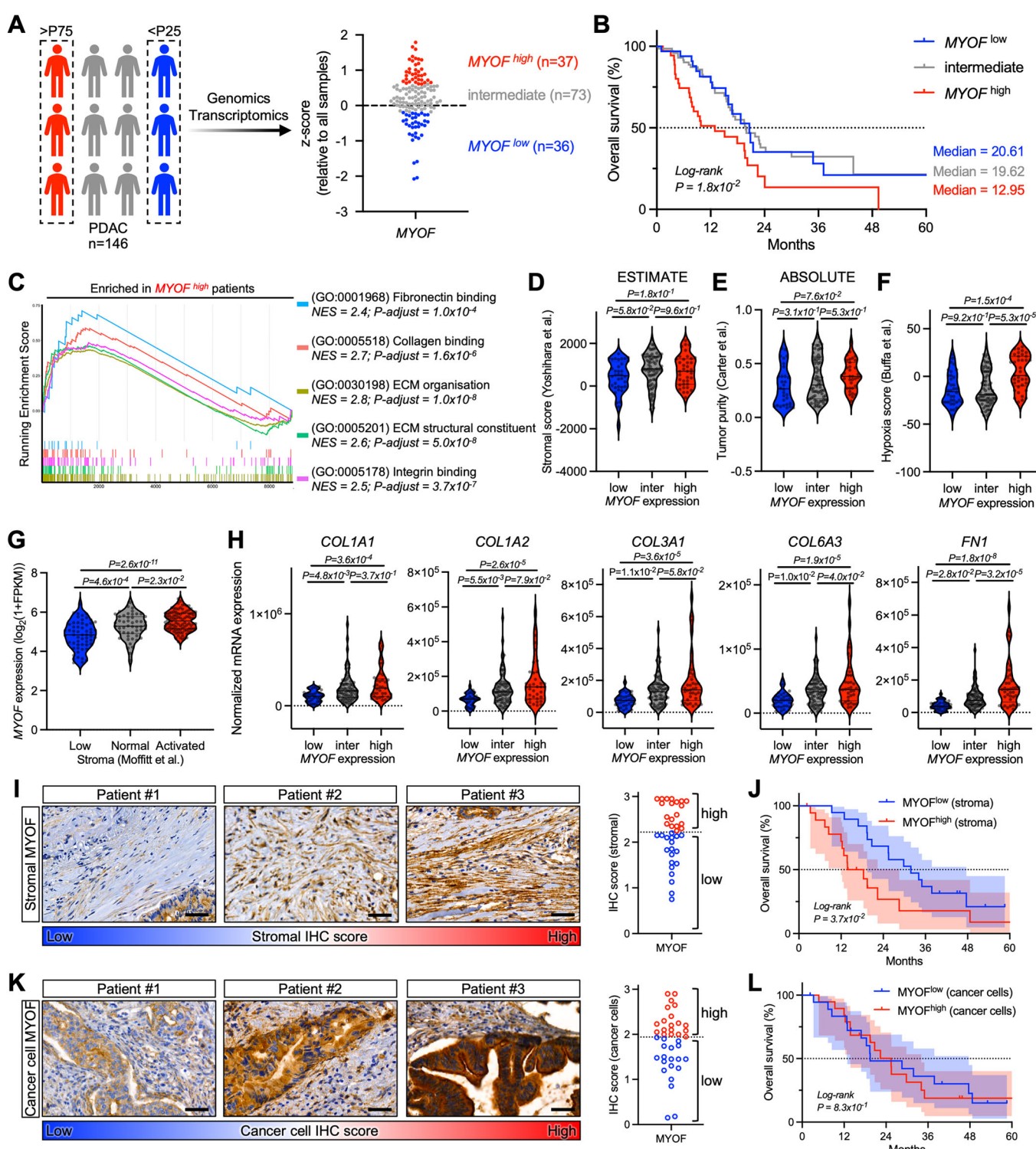

were the clusters with highest *MYOF* expression (Fig. 2B). These results not only validate previous reports describing myoferlin in epithelial (Gupta et al, 2021; Rademaker et al, 2022) and endothelial cells (Bernatchez et al, 2009), but also reveal a yet unknown expression of myoferlin in CAFs. To further investigate *MYOF* expression in CAFs, which are known to be composed of

heterogeneous populations (Biffi and Tuveson, 2021), we performed CAF subclustering (Fig. EV2B). We were able to subcluster CAFs into myofibroblast-like CAFs (myCAFs), inflammatory CAFs (iCAFs), antigen-presenting CAFs (apCAFs) and pericytes (Fig. EV2C), the latter frequently contaminate CAF clusters due to overlapping markers (Nurmik et al, 2020). *MYOF* expression was

**Figure 1. Stromal myoferlin drives tumor aggressiveness and is linked to desmoplasia in pancreatic cancer patients.**

(A) TCGA PAAD cohort patients with confirmed ductal phenotype ($n = 146$) were split in three groups based on *MYOF* z-score expression quartiles: high (>P75; $n = 36$), intermediate (P25-P50; $n = 73$), low (<P25; $n = 37$). (B) Kaplan–Meier plot showing 5-year overall survival rates (including median survival) of TCGA PAAD cohort patients ($n = 146$). Strata: *MYOF* expression. Log rank (Mantel–Cox) test. (C) GSEA analysis of differentially expressed genes ($P < 0.05$) between *MYOF*[high] patients (>P75; $n = 36$) and *MYOF*[low] patients (<P25; $n = 37$) from the TCGA PAAD cohort. Benjamini–Hochberg procedure. NES = normalized enrichment score. (D) ESTIMATE stromal scores in TCGA PAAD cohort patients ($n = 146$) according to *MYOF* expression. Violin plot, one-way ANOVA (Tukey's test). (E) ABSOLUTE tumor purity in TCGA PAAD cohort patients ($n = 146$) according to *MYOF* expression. Violin plot, one-way ANOVA (Tukey's test). (F) Hypoxia scores in TCGA PAAD cohort patients ($n = 146$) according to *MYOF* expression. Violin plot, one-way ANOVA (Tukey's test). (G) *MYOF* expression in human stroma-specific PAAD subtypes: Low ($n = 57$), Normal ($n = 50$), Activated ($n = 99$). Violin plot, one-way ANOVA (Tukey's test). (H) *COL1A1, COL1A2, COL3A1, COL6A3* and *FN1* expression in TCGA PAAD cohort patients ($n = 146$) according to *MYOF* expression. Violin plot, one-way ANOVA (Tukey's test). (I) Myoferlin IHC staining of PAAD patient sections ($n = 38$) and quantification of stromal myoferlin abundance. Representative images are shown. Scale bar = 50 µm. (J) Kaplan–Meier plot showing 5-year overall survival rates of PAAD patients ($n = 38$). Strata: stromal myoferlin score. Log rank (Mantel–Cox) test. (K) Myoferlin IHC staining of PAAD patient sections ($n = 38$) and quantification of cancer cell myoferlin abundance. Representative images are shown. Scale bar = 50 µm. (L) Kaplan–Meier plot showing 5-year overall survival rates of PAAD patients ($n = 38$). Strata: cancer cell myoferlin score. Log rank (Mantel–Cox) test. Source data are available online for this figure.

not statistically different across myCAF, iCAF and apCAF clusters (Fig. 2C). Next, investigated *MYOF* expression in different CAF subtypes that show only partial overlap with myCAFs and iCAFs, namely pro-tumoral CD105[pos] (*ENG*[+]) and anti-tumoral CD105[neg] (*ENG*[-]) CAFs (Hutton et al, 2021). In agreement with the myCAF/iCAF subclustering, *MYOF* was equally expressed between *ENG*[+] and *ENG*[-] CAFs (Fig. 2D). Altogether, these results suggest a panCAF expression of myoferlin with a potential pleiotropic function across CAFs.

Knowing that myoferlin was indeed abundantly expressed in human CAFs, we wondered whether this would also be the case in PAAD mouse models. Therefore, we investigated myoferlin expression in a murine PAAD scRNAseq dataset (Dominguez et al, 2020) that not only recapitulates distinct CAF subtypes (iCAF/myCAF), but also assesses CAF lineages during tumor progression (Fig. EV2D–F). *Myof* expression across all cell types showed a close correlation with pan-CAF markers (*Pdpn*, *Pdgfra* and *Fap*) and poorer correlation to EMT- and epithelial markers (*Alcam* and *Krt18*, respectively) (Fig. 2E). Despite this close similarity between our murine findings and human PAAD (i.e., high *Myof* expression in fibroblasts and tumor cells), we did notice discrepancies, such as the absence of *Myof* in mouse endothelial cells (Fig. EV2G).

Next, we evaluated *Myof* expression across distinct CAF subclusters and lineages. Dominguez et al describe 2 major CAF lineages, both transitioning from non-tumoral fibroblasts (ntFib) via early CAFs (eCAF) towards late CAFs. The myCAF lineage is dependent of TGFß stimulation (TGFß CAFs), while iCAF lineage progression is driven by IL1 (IL1 CAFs) (Dominguez et al, 2020). We noticed an increase of *Myof* expression in the myCAF lineage during tumor progression (Fig. 2F,G), absent in the iCAF lineage (Fig. 2H,I). Finally, late stage myCAFs and iCAFs from established tumors (TGFß-CAFs and IL1-CAFs, respectively) showed equal *Myof* expression (Fig. EV2G). Similar results were obtained when comparing *Eng*[+] and *Eng*[-] CAFs (Fig. EV2H) and murine CyTOF-isolated CD105[pos] and CD105[neg] CAFs (Fig. EV2I).

Collectively, these findings in human and murine PAAD reveal that myoferlin is pleiotropically expressed in CAFs across all subtypes. Consistent with the correlation between stromal myoferlin and ECM abundance (Fig. EV1H,I), we show that myCAFs, key actors during tumor fibrosis (Öhlund et al, 2017; Bachem et al, 2005), indeed express myoferlin, implying a potential importance of myoferlin for myCAF activity.

## Myoferlin knockdown impairs ECM production and deposition

During tumorigenesis, several residential cell types such as pancreatic stellate cells (PSCs), mesenchymal stem cells or residual fibroblasts are educated by tumor cells and give rise to activated CAFs (Öhlund et al, 2014; Manoukian et al, 2021). Among these cells, PSCs have been proven in vitro and in vivo to give rise to several CAF subtypes (Öhlund et al, 2017; Fujita et al, 2010). To understand and to study the function of myoferlin in pancreatic myCAFs, we used a well described culture model for PSCs (Biffi et al, 2019). PSC-derived myCAFs were obtained by 2D culture, while quiescent PSCs (qPSCs) were obtained via Matrigel embedding (Fig. EV3A). myCAF markers (Öhlund et al, 2017) (*CCN2, COL1A1, ACTA2*) were enriched in 2D-cultured myCAFs, while quiescence markers (Jesnowski et al, 2005) such as lipid droplets and lipid droplet-related genes (*PLIN1, PLIN2*) were lost in myCAFs when compared to qPSCs (Fig. EV3B,C). Intriguingly, *MYOF* expression was identical between qPSCs and myCAFs (Fig. EV3D), implying a potential function of myoferlin in both cell states, in agreement with a pleiotropic function suggested by scRNAseq findings described above. We then knocked down *MYOF* (*MYOF*[KD]) in PSC-derived myCAFs and performed genome-wide transcriptomic profiling (Fig. 3A). Due to the important plasticity described in CAFs (Biffi et al, 2019; Peiffer et al, 2023), we first asked whether *MYOF*[KD] affected the myCAF phenotype of CAFs in vitro. However, myCAF- and iCAF-related gene expression (Öhlund et al, 2017) in our transcriptomic data did not reveal a shift to a specific CAF subtype in *MYOF*[KD] CAFs (Fig. EV3E). A potential shift from myCAFs towards quiescence was also not observed, as *MYOF*[KD] myCAFs retained proliferation rates (Fig. EV3F), alpha-SMA expression and structure (Fig. EV3G–I) without acquiring lipid droplets (Fig. EV3J,K).

To further explore the function of myoferlin in myCAFs, we performed gene set enrichment analysis on differentially expressed genes ($P < 0.05$) between *MYOF*[KD] myCAFs and CTRL myCAFs. Importantly, we found that several ECM-related gene sets were significantly downregulated upon *MYOF*[KD] (Fig. 3B). Using RT-qPCR, we confirmed that indeed all major PAAD matrisome genes (*COL1A1, COL1A2, COL2A1, COL3A1, COL6A3* and *FN1*) were significantly less expressed in *MYOF*[KD] myCAFs (Fig. 3C). This loss of ECM gene transcription was also prominent at the protein level, as collagen 1 and fibronectin were both less abundant in whole-cell

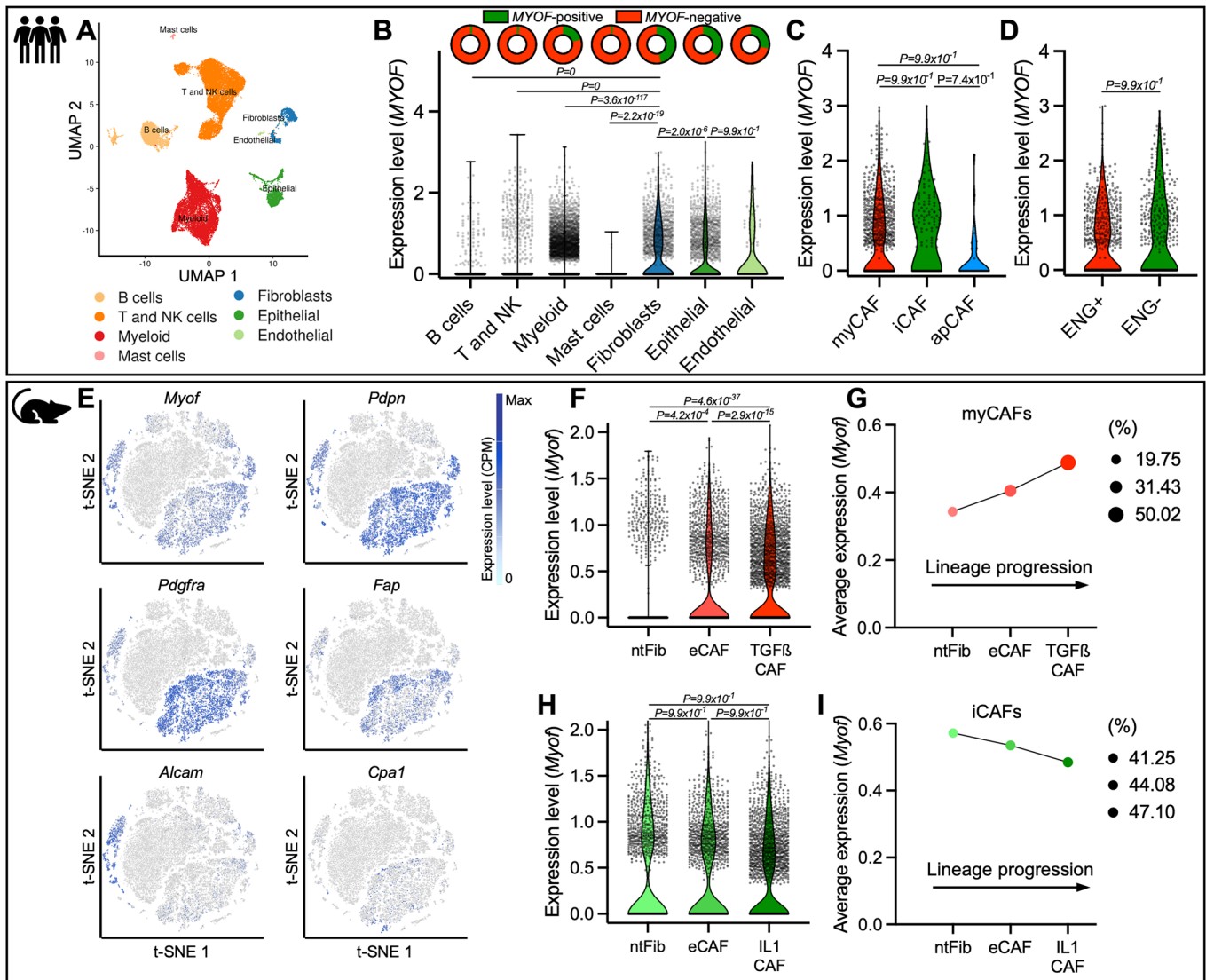

**Figure 2. Myoferlin is expressed pleiotropically in pancreatic CAFs.**

(A) UMAP plot and broad cluster labelling from human PAAD scRNAseq data. (B) Normalized *MYOF* expression levels and *MYOF*-positive cell (>0.05 expression level) proportions (donut chart) in broad clusters including B cells (n = 3364), T and NK cells (n = 10922), Myeloid cells (n = 9547), Mast cells (n = 151), Fibroblasts (n = 1465), Epithelial cells (n = 1945) and Endothelial cells (n = 123). Violin plot, pairwise comparisons using Wilcoxon Rank test (Bonferroni correction). (C) Normalized *MYOF* expression levels in fibroblasts subclusters, including myCAFs (n = 903), iCAFs (n = 130) and apCAFs (n = 33). Violin plot, pairwise comparisons using Wilcoxon Rank test (Bonferroni correction). (D) Normalized *MYOF* expression levels in CAFs clustered according to CD105 (*ENG*) expression (0.05 threshold for ENG + ). ENG-positive (n = 624) and ENG-negative (n = 442). Violin plot, pairwise comparisons using Wilcoxon Rank test (Bonferroni correction). (E) t-SNE plots for *Myof*, panCAF markers (*Pdpn*, *Pdgfra*, *Fap*), *Alcam* (EMT marker) and *Cpa1* (acinar marker) expression from murine PAAD scRNAseq data. Color scale representing gene expression levels (counts per million - CPM). (F) *Myof* expression levels in myCAF lineage-associated normal fibroblasts (ntFib, n = 1210), early CAFs (eCAF, n = 2101) and late CAFs (TGFß CAF, n = 2289). Violin plot, pairwise comparisons using Wilcoxon Rank test (Bonferroni correction). (G) Average *Myof* expression and *Myof*-positive (>0 expression level) cell proportions (%) in myCAF lineage-associated normal fibroblast - (ntFib), early CAF - (eCAF) and late CAF - (TGFß CAF) clusters. (H) *Myof* expression levels in iCAF lineage-associated normal fibroblasts (ntFib, n = 1435), early CAFs (eCAF, n = 1500) and late CAFs (IL1 CAF, n = 2102). Violin plot, pairwise comparisons using Wilcoxon Rank test (Bonferroni correction). (I) Average *Myof* expression and *Myof*-positive (>0 expression level) cells (%) in iCAF lineage-associated normal fibroblast - (ntFib), early CAF - (eCAF) and late CAF - (IL1 CAF) clusters. Source data are available online for this figure.

lysates from *MYOF*[KD] myCAFs compared to CTRL myCAFs (Figs. 3D,E and EV3L).

To determine whether the loss of ECM transcription and synthesis also affected ECM secretion, we evaluated the presence of collagen 1 and fibronectin in myCAF conditioned media (CM). Under *MYOF*[KD], myCAFs secreted significantly reduced amounts of collagen 1 and fibronectin into their medium (Fig. 3F,G), reflecting impaired ECM

gene transcription, protein translation and end-product secretion. Finally, we extended these findings in a 3D culture model more relevant to patient tumors. Additionally, a 3D spheroid model helped to elucidate whether our phenotype was dependent of cell/substrate interactions. The latter is of major importance, as stiff ECM (mimicked by cell culture plates) can promote ECM production in CAFs (Jesnowski et al, 2005). We therefore used *MYOF*[KD] CAFs to generate

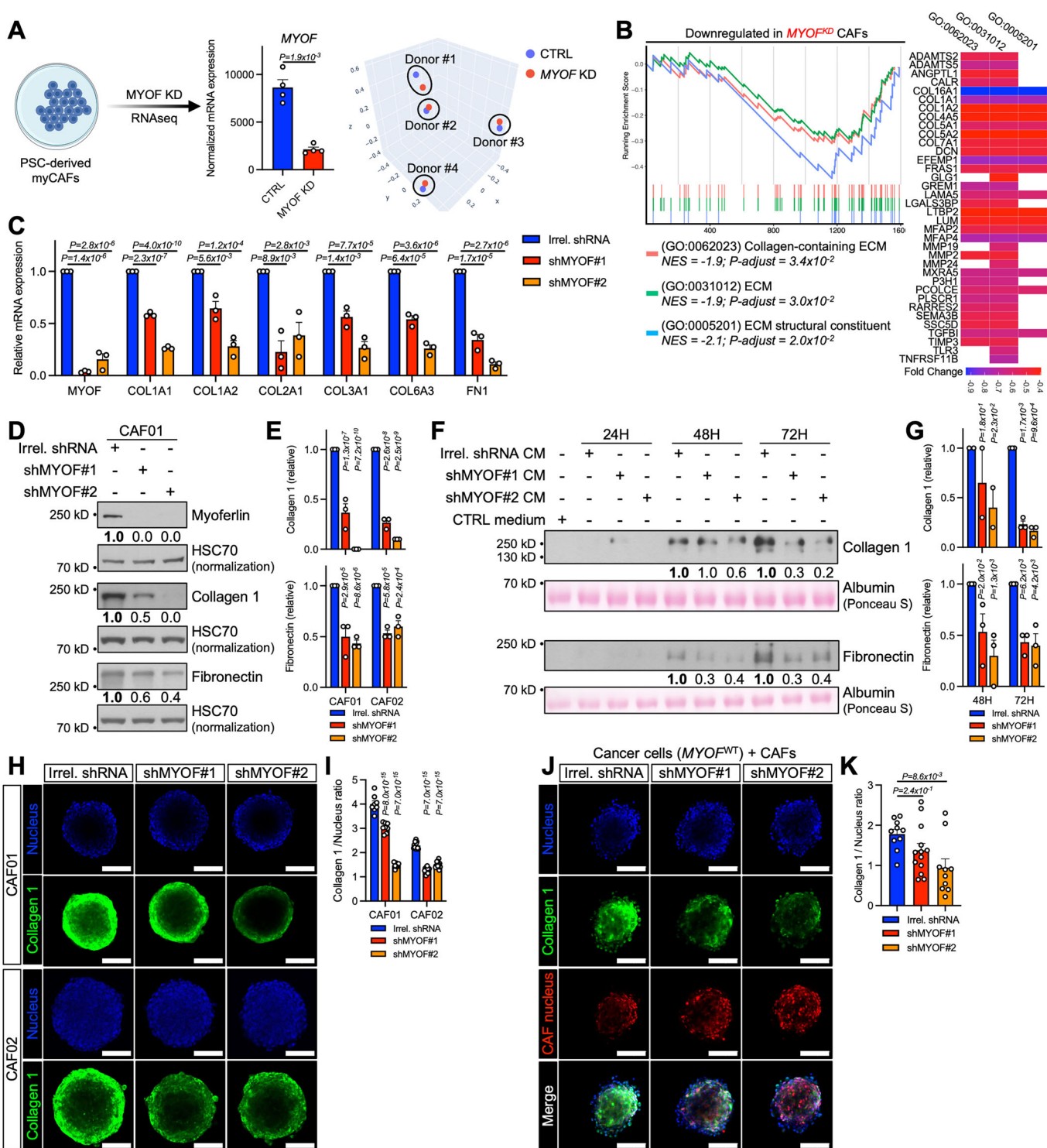

homotopic (CAFs alone) and heterotopic (CAFs + *MYOF*^WT cancer cells) spheroids, in which we evaluated collagen 1 abundance via confocal microscopy. In both models, *MYOF*^KD led to decreased abundance of collagen 1, independently of 2D or 3D culture conditions (Fig. 3H–J). Collectively, these results highlight that myoferlin knockdown interferes with myCAF activity through impaired ECM gene transcription as well as ECM protein synthesis and secretion.

## Myoferlin knockdown disrupts SMAD2/3-mediated TGFß signaling

Fibroblasts rely on external stimuli, such as growth factors and cytokines present in their cellular environment, for activation and transdifferentiation into CAF subtypes (Kalluri, 2016). Once in a myCAF state, CAFs abundantly secrete ECM components, perform

**Figure 3. Myoferlin knockdown impairs ECM production and deposition.**

(A) Bar plot of normalized *MYOF* mRNA expression levels and three-dimensional PCA plot of PSC-derived myCAFs silenced for *MYOF* ($n = 4$). Mean ± SEM, paired *T* test. (B) GSEA analysis and heatmap of differentially expressed genes ($P < 0.05$) between *MYOF*[KD] myCAFs (siMYOF#1, $n = 4$) and CTRL myCAFs ($n = 4$). Benjamini–Hochberg procedure. NES normalized enrichment score. (C) RT-qPCR analysis (CAF01) of *MYOF*, *COL1A1*, *COL1A2*, *COL2A1*, *COL3A1*, *COL6A3*, and *FN1* mRNA levels in CTRL myCAFs (Irrel. shRNA; $n = 3$) and *MYOF*[KD] myCAFs (shMYOF#1 and shMYOF#2; $n = 3$ each). Mean ± SEM, one-way ANOVA (Tukey's test). (D) Western blot analysis (CAF01) of total-cell lysates from CTRL myCAFs (Irrel. shRNA; $n = 3$) and *MYOF*[KD] myCAFs (shMYOF#1 and shMYOF#2; $n = 3$ each). One representative western blot of three independent experiments is shown, HSC70 was used as loading control. (E) Quantification of western blots shown in (D) and Fig. EV3L ($n = 3$). Mean ± SEM, one-way ANOVA (Tukey's test). (F) Western blot analysis (CAF01) of conditioned media (CM) from cell number-adjusted CTRL myCAFs (Irrel. shRNA; $n = 3$) and *MYOF*[KD] myCAFs (shMYOF#1 and shMYOF#2; $n = 3$ each). One representative western blot of three independent experiments is shown, albumin was revealed via Ponceau S staining and was used as loading control. (G) Quantification of western blots shown in (F). Bar plots represent quantifications of 48H ($n \geq 2$) and 72H ($n = 3$) measurements. Mean ± SEM, one-way ANOVA (Tukey's test). (H) Immunofluorescence microscopy of homotopic spheroids ($n \geq 8$) generated from CTRL myCAFs (Irrel. shRNA) or *MYOF*[KD] myCAFs (shMYOF#1 and shMYOF#2). Representative pictures are shown. Nuclei = blue, collagen 1 = green, scale bar = 100 μm. (I) Mean fluorescence quantification of spheroids ($n \geq 8$) shown in (H), collagen 1 intensity was normalized to DAPI. Mean ± SEM, one-way ANOVA (Tukey's test). (J) Immunofluorescence microscopy of heterotopic spheroids ($n \geq 10$) generated from PAAD cells (PANC-1) in coculture with mKate red-fluorescent CTRL myCAFs (CAF01, Irrel. shRNA) or *MYOF*[KD] myCAFs (CAF01, shMYOF#1 and shMYOF#2). Representative pictures are shown. Nuclei = blue, collagen 1 = green, myCAF nucleus = red, scale bar = 100 μm. (K) Mean fluorescence quantification of spheroids ($n \geq 10$) shown in (J), collagen 1 intensity was normalized to DAPI. Mean ± SEM, one-way ANOVA (Tukey's test). Source data are available online for this figure.

matrix remodeling and induce tumor fibrosis (Caligiuri and Tuveson, 2023). Several signaling pathways have been reported to stimulate pancreatic myCAF activity and transcriptionally converge to induce fibrosis, such as YAP/TAZ-mediated Hippo pathway (Shen et al, 2020; Morvaridi et al, 2015) and TGFß signaling (Löhr et al, 2001). Knowing that myoferlin silencing impaired myCAF activity and ECM gene transcription, we investigated intracellular signaling pathways involved in ECM production, potentially dysregulated upon *MYOF*[KD].

We initially focused on Hippo signaling, which was significantly enriched in *MYOF*[high] patients. However, only a minor subset of *MYOF*[low] patients showed decreased Hippo-related gene expression (Fig. EV4A) and *MYOF*[KD] myCAFs had neither altered actin structure (Fig. EV3H), nor altered protein abundance or phosphorylation of Hippo mediators (Fig. EV4B). Finally, YAP/TAZ target gene transcription was also unaffected by *MYOF* knockdown (Fig. EV4C), collectively challenging a major implication of Hippo signaling in myoferlin-dependent ECM production.

Focusing next on TGFß signaling, we found that TGFß signaling-related genes (R-HSA9006936) were significantly enriched in *MYOF*[high] patients. Furthermore, when segregating PAAD patients according to *MYOF* expression, we found a clustering of *MYOF*[low] patients with low expression of TGFß genes, while *MYOF*[high] patients clustered with highly expressed TGFß genes (Fig. 4A). In agreement with previous findings of this study, patients with low TGFß-related gene expression also presented higher overall survival and lower expression of ECM genes (*COL1A1*, *COL1A2*, *COL3A1*, *COL6A3*, *FN1*) (Fig. 4A). Collectively, patient data hereby links myoferlin expression to TGFß signaling and ECM production in PAAD.

To find out if the loss of ECM production in *MYOF*[KD] myCAFs was indeed linked to impaired TGFß signaling, we assessed c-terminal (S465/467) SMAD2 phosphorylation, a member of the canonical TGFß signaling pathway involved in tissue fibrosis (Hu et al, 2018). Strikingly, we found that SMAD2 was less phosphorylated in *MYOF*[KD] myCAFs, while total SMAD2/3/4 abundances were unaffected (Fig. 4B,C). Intrigued by these findings, we decided to thoroughly study the impact of *MYOF*[KD] on TGFß signaling in a ligand-dependent manner. We conducted pulse-chase experiments with recombinant TGFß1 on *MYOF*[KD]

myCAFs and assessed TGFß signal transduction kinetics (≤60 min) via c-terminal SMAD2/3 phosphorylation. Upon exogenous stimulation, TGFß response was impaired in *MYOF*[KD] myCAFs, as highlighted by reduced SMAD3 phosphorylation, particularly at the phosphorylation peak of 45 min (Figs. 4D,E and EV4D–F).

To understand if impaired TGFß signal transduction could indeed be the cause for reduced ECM gene transcription in *MYOF*[KD] myCAFs, we next used confocal microscopy to evaluate SMAD2/3 nuclear translocation, a process essential for TGFß target gene transcription (Hill, 2009). In the absence of exogenous TGFß we found that *MYOF*[KD] already presented increased cytosolic SMAD2/3 retention (Fig. 4F), consistent with reduced SMAD2/3 phosphorylation (Fig. 4B). Additionally, upon TGFß stimulation (up to 60 min), SMAD2/3 translocated less into the nucleus of *MYOF*[KD] myCAFs, while CTRL myCAFs underwent nearly complete SMAD2/3 nuclear translocation (Fig. 4F,G). Reduced SMAD2/3 nuclear translocation in *MYOF*[KD] myCAFs was further validated using subcellular fractionation (Fig. 4H,I).

Finally, we directly assessed ECM gene transcription using RT-qPCR upon TGFß stimulation and included *TGFBI* (also known as ßig-h3), a well described TGFß target gene (Skonier et al, 1994), as positive control. While all major PAAD matrisome genes (*COL1A1*, *COL1A2*, *COL3A1*, *COL6A3*, *FN1*) had SMAD3 binding motifs in their promoter regions (Fig. EV4G), we found that *COL1A1* and *FN1* were particularly induced over time upon TGFß stimulation, suggesting different induction timings among SMAD3 target genes (Fig. EV4H). When evaluating *COL1A1*, *FN1* and *TGFBI* expression in unstimulated *MYOF*[KD] myCAFs, we confirmed reduced expression when compared to control myCAFs. Strikingly, even under exogenous TGFß, *MYOF*[KD] myCAFs still showed lower expression of *COL1A1*, *FN1* and *TGFBI* than CTRL myCAFs (Fig. 4J).

Taken together, these results demonstrate that *MYOF*[KD] disrupts TGFß signaling as early as SMAD2/3 phosphorylation, impairing SMAD2/3 nuclear translocation and TGFß target gene transcription. Importantly, *MYOF*[KD]-induced disruption of TGFß signaling seems unaffected by exogenous TGFß stimulation, suggesting a myCAF-intrinsic mechanism rather than a ligand abundance-related effect linked to autocrine signaling.

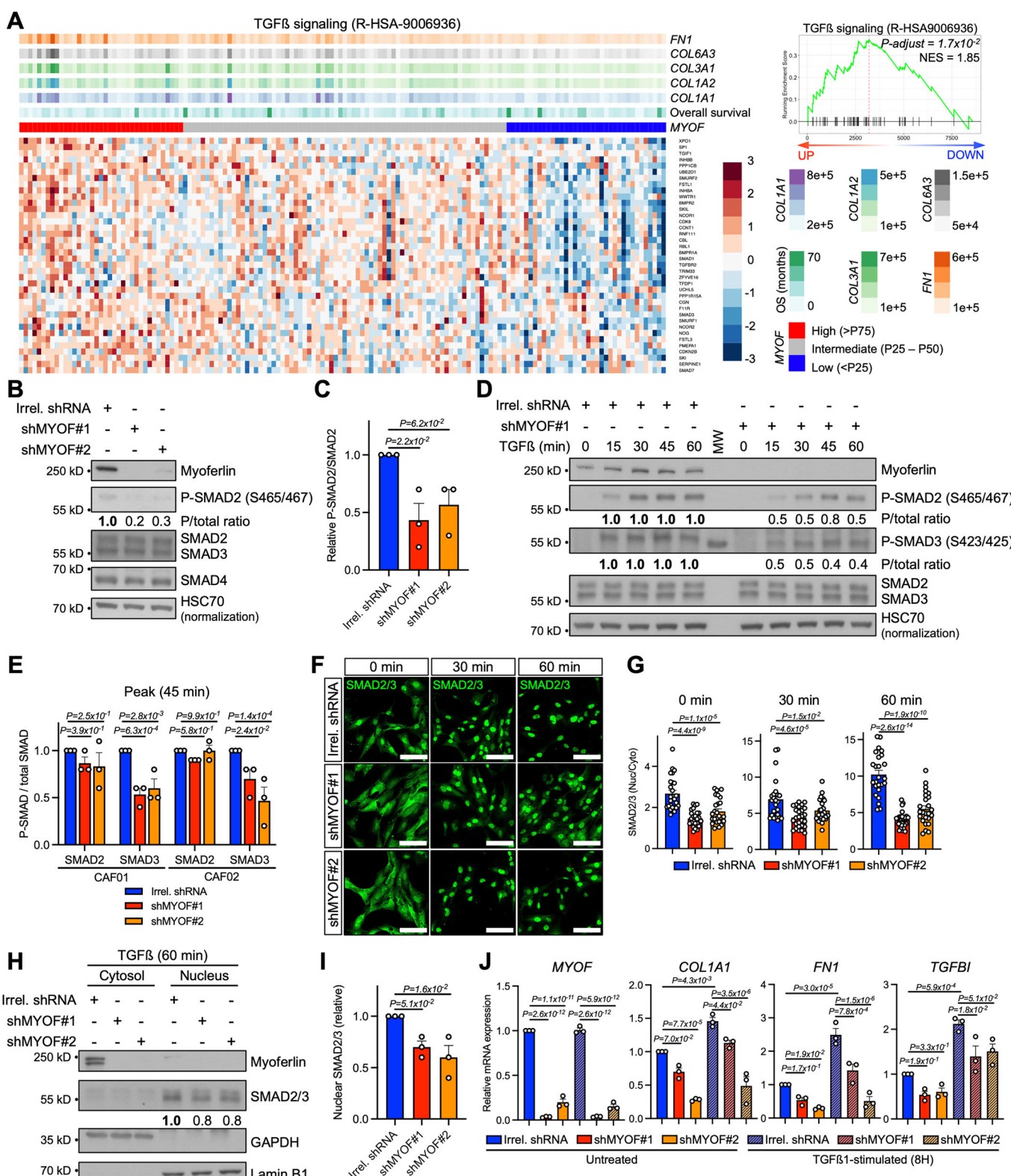

**Figure 4. Myoferlin knockdown disrupts SMAD2/3-mediated TGFß signaling.**

(A) Heatmap and GSEA analysis of TCGA PAAD cohort patients ($n = 146$) and TGFß-related genes ($P < 0.05$). Patients are segregated based on *MYOF* expression. *P* value assessed via Benjamini–Hochberg procedure. (B) Western blot analysis (CAF01) of total-cell lysates from CTRL myCAFs (Irrel. shRNA; $n = 3$) and *MYOF*$^{KD}$ myCAFs (shMYOF#1 and shMYOF#2; $n = 3$ each). One representative western blot of three independent experiments is shown, HSC70 was used as loading control. (C) Quantification of (B). Mean ± SEM, one-way ANOVA (Tukey's test), *P* value relative to control group (Irrel. shRNA). (D) Western blot analysis of total-cell lysates (CAF01) from CTRL myCAFs (Irrel. shRNA; $n = 3$) and *MYOF*$^{KD}$ myCAFs (shMYOF#1, $n = 3$) stimulated with human recombinant TGFß1 (5 ng/mL) for indicated timepoints. One representative western blot of three independent experiments is shown, HSC70 was used as loading control. (E) Quantification of (D) and Fig. EV4D,E. Mean ± SEM, one-way ANOVA (Tukey's test), *P* value relative to control group (Irrel. shRNA). (F) Immunofluorescent confocal microscopy (CAF01) of SMAD2/3 nuclear translocation upon TGFß1 stimulation (5 ng/mL) in CTRL myCAFs (Irrel. shRNA; $n > 24$) and *MYOF*$^{KD}$ myCAFs (shMYOF#1; $n > 25$ and shMYOF#2; $n = 25$). Representative pictures are shown. Nuclei (DAPI) = blue, SMAD2/3 = green, scale bar = 100 μm. (G) Quantification of (F). Nuclear and cytosolic mean fluorescence was quantified for individual cells across ≥3 fields of view. Mean ± SEM, one-way ANOVA (Tukey's test), *P* value relative to control group (Irrel. shRNA). (H) Western blot analysis of subcellular fractions (CAF01) from TGFß1 (5 ng/ml) stimulated CTRL myCAFs (Irrel. shRNA; $n = 3$) and *MYOF*$^{KD}$ myCAFs (shMYOF#1 and shMYOF#2; $n = 3$ each). One representative western blot of three independent experiments is shown, GAPDH and Lamin B1 were used as cytosolic and nuclear markers, respectively. (I) Quantification of (H). Mean ± SEM, one-way ANOVA (Tukey's test), *P* value relative to control group (Irrel. shRNA). (J) RT-qPCR analysis (CAF01) of *MYOF*, *COL1A1*, *FN1* and *TGFBI* mRNA levels in untreated or TGFß1-stimulated (5 ng/ml; 8 h) CTRL myCAFs (Irrel. shRNA; $n = 3$) and *MYOF*$^{KD}$ myCAFs (shMYOF#1 and shMYOF#2; $n = 3$ each). Mean ± SEM, one-way ANOVA (Tukey's test). Source data are available online for this figure.

## Myoferlin knockdown interrupts COPII-dependent TGFBR1 trafficking

We next sought to unveil a precise molecular mechanism, by which myoferlin depletion altered TGFß signaling and ECM production. Importantly, the phosphorylation on the c-terminal region of SMAD3 (S423/425), which leads to nuclear translocation and target gene transcription, is only mediated by the kinase TGFß receptor 1 (TGFBR1, also known as ALK5) upon TGFß binding (Tarasewicz and Jeruss, 2012). Owing to the fact that c-terminal SMAD3 serine 423 and 425 are less phosphorylated upon *MYOF*$^{KD}$ (Fig. 4D), we hypothesized that myoferlin is important for TGFBR1 function.

To investigate a potential implication of myoferlin in TGFBR1 function, we first evaluated TGFBR1 abundance and stability after myoferlin knockdown in myCAFs. We found that the total abundance of TGFBR1/2 was unaffected in *MYOF*$^{KD}$ myCAFs (Fig. EV5A, 0 h time point). Furthermore, by performing cycloheximide (CHX) chase experiments, blocking translational elongation and protein synthesis, in combination with recombinant TGFß stimulation, we revealed that the stability of TGFBR1/2 was also unchanged, and no increased TGFBR1/2 degradation was visible upon TGFß stimulation in *MYOF*$^{KD}$ myCAFs (Fig. EV5A).

As TGFBR1 abundance and degradation were not affected by myoferlin depletion, we next wondered whether the subcellular localization of TGFBR1 was altered in *MYOF*$^{KD}$ myCAFs. This idea was supported by the described implication of non-neoplastic myoferlin in vesicle trafficking and receptor recycling (Bernatchez et al, 2007; Demonbreun et al, 2010). Strikingly, genome-wide transcriptomic profiling of *MYOF*$^{KD}$ myCAFs revealed only one significantly downregulated gene set associated with vesicle trafficking, namely "COPII-coated ER to Golgi transport vesicle", in addition to other gene sets involved in the endoplasmic reticulum (ER) and Golgi interface (Fig. 5A). This finding particularly caught our attention, as COPII-coated vesicles are a specialized shuttle between the ER and Golgi apparatus (Fig. 5B), mediating the anterograde transport of a great variety of transmembrane and secreted cargo proteins (Sato and Nakano, 2007). The importance of COPII-coated vesicles in PAAD was further supported by the increased expression of COPII-related genes (*SAR1A*, *SEC13*, *SEC31A*, *SEC23A*, *SEC24A*, *SEC24B*, *SEC24C*) in PAAD tissue compared to matched healthy pancreas (Fig. EV5B).

We then aimed to assess the subcellular localization of COPII vesicles in *MYOF*$^{KD}$ myCAFs, and particularly the fusion of COPII vesicles with their destination—the Golgi apparatus. Therefore, we performed immunofluorescence confocal microscopy of the Golgi apparatus marker Golgin-97 and the COPII marker Sec24C, the latter being a core protein of the COPII coat and predominantly responsible for cargo selection (Adams et al, 2014). We quantified the colocalization between COPII vesicles (Sec24C) and the Golgi apparatus (Golgin-97) first in a pixel-based approach, calculating threshold-adjusted Mander's coefficients (tM2). While CTRL myCAFs presented strong colocalization between COPII vesicles and Golgi apparatus, implying a highly active anterograde ER/Golgi protein transport, we found reduced colocalization between COPII vesicles and the Golgi apparatus in *MYOF*$^{KD}$ myCAFs (Fig. 5C). Of note, total abundances of Sec24C and Golgin-97 were unchanged after *MYOF*$^{KD}$ (Fig. EV5C). As Mander's coefficients can be heavily biased by image noise, we validated our results using an object-based colocalization quantification—namely statistical object distance analysis (SODA) (Lagache et al, 2018) (Fig. 5C). Furthermore, to confirm that impaired ER/Golgi vesicle trafficking indeed resulted in reduced colocalization between COPII vesicles and the Golgi apparatus, we treated myCAFs with Brefeldin A (BFA), a common inhibitor of ER/Golgi transport (Fujiwara et al, 1988). BFA treatment mimicked *MYOF*$^{KD}$ in myCAFs, as colocalization between COPII vesicles and the Golgi apparatus was impaired (Fig. 5C), thus supporting that *MYOF*$^{KD}$ disrupts anterograde vesicle trafficking of COPII vesicles between the ER and Golgi apparatus.

As myoferlin has never been described in the context of COPII vesicles, we investigated if myoferlin was a potential member of the COPII complex and thus directly involved in COPII vesicle trafficking. Immunofluorescence colocalization between Sec24C and myoferlin in combination with Mander's coefficient calculation revealed a strong colocalization between both proteins, mainly in the perinuclear region (Figs. 5D,F and EV5D), which is described as the subcellular localization of the ER/Golgi interface (Brandizzi and Barlowe, 2013). A potential implication of myoferlin in COPII vesicle function was further supported via proximity ligation assay (PLA) between myoferlin and Sec24C or Sec31. The abundant presence of MYOF/Sec24C and MYOF/Sec31 PLA dots in myCAFs proved a close relationship (<40 nm distance) between myoferlin and COPII vesicles which was lost upon *MYOF*$^{KD}$ (Figs. 5E,G

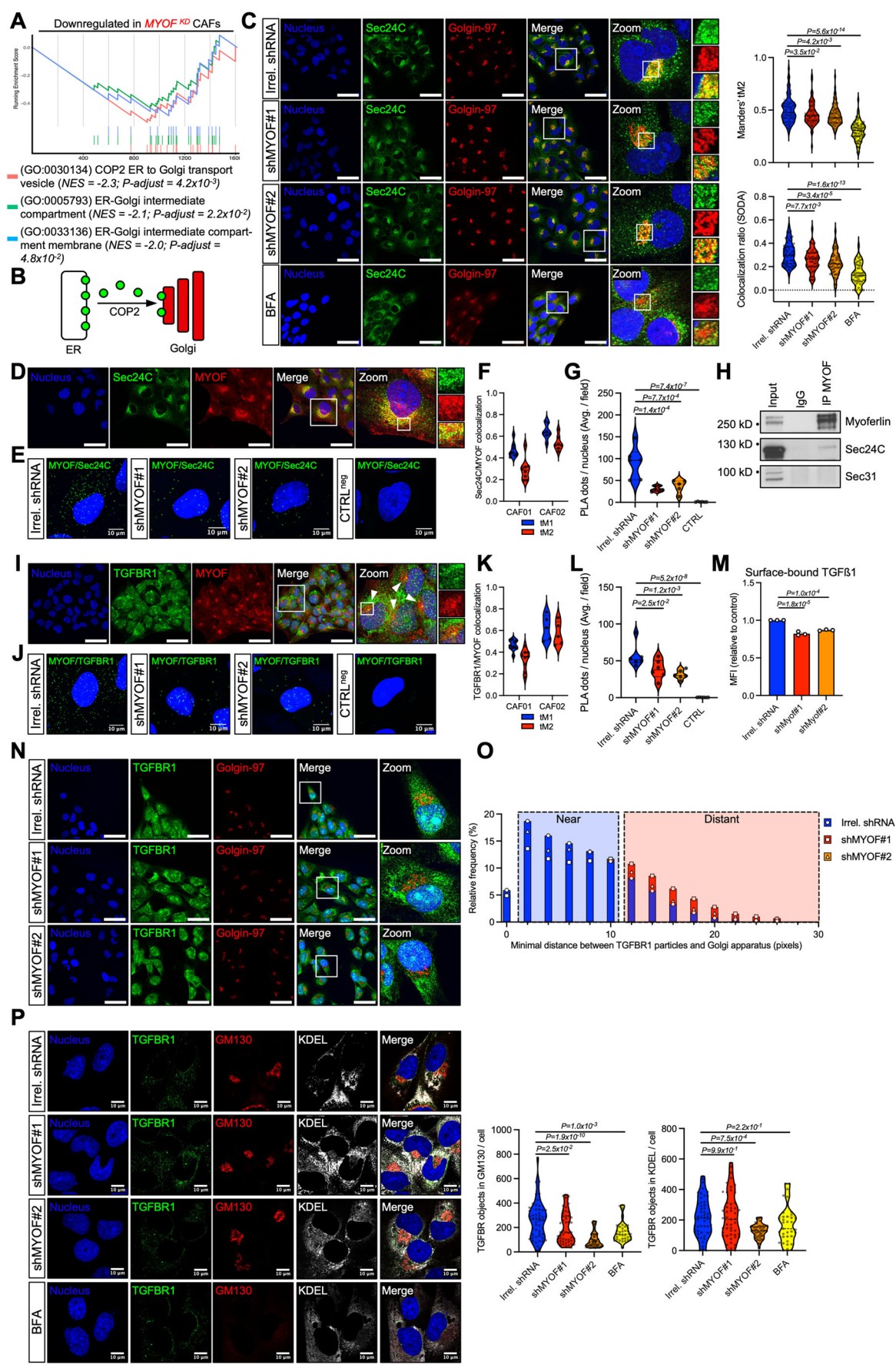

◄   **Figure 5.   Myoferlin knockdown interrupts COPII-dependent TGFBR1 trafficking.**

(A) GSEA analysis of differentially expressed genes (*P* < 0.05) between *MYOF*^KD^ myCAFs (siMYOF#1, n = 4) and CTRL myCAFs (n = 4). Benjamini–Hochberg procedure. NES normalized enrichment score. (B) Visualization of COPII vesicles as shuttle between the endoplasmic reticulum (ER) and Golgi apparatus. (C) Immunofluorescence microscopy (CAF01) with pixel-based colocalization quantification (tM2 Mander's coefficients after threshold regression) and object-based colocalization quantification (SODA) of COPII vesicles and the Golgi apparatus in CTRL myCAFs (Irrel. shRNA; n > 50), *MYOF*^KD^ myCAFs (shMYOF#1 and shMYOF#2; n > 50 each) and myCAFs treated with Brefeldin A (BFA; n ≥ 50). Representative pictures from 5 distinct observations across 2 independent wells are shown, n represents individual cells across the five fields of view. Nuclei = blue, COPII vesicles (Sec24C) = green, Golgi apparatus (Golgin-97) = red, scale bar = 50 μm. Violin plot, one-way ANOVA (Tukey's test). (D) Immunofluorescence microscopy of myCAFs (CAF01). Representative pictures are shown. Nuclei = blue, COPII vesicles (Sec24C) = green, myoferlin (MYOF) = red, scale bar = 50 μm. (E) Proximity ligation assay (PLA) (CAF01) between myoferlin (MYOF) and COPII vesicles (Sec24C) in CTRL myCAFs and *MYOF*^KD^ myCAFs (shMYOF#1 and shMYOF#2). PLA control (CTRL^neg^) without primary antibodies is included. Representative pictures are shown. Nuclei = blue, PLA dots = green, scale bar = 10 μm. (F) Quantification of colocalization (tM1 and tM2 Mander's coefficients after threshold regression) shown in (D) and Fig. EV5D. Violin plot. (G) Quantification of PLA dots shown in (E). Violin plot, one-way ANOVA (Tukey's test). (H) Western blot analysis of myoferlin co-IP eluants in myCAFs (CAF01). Input = total cell lysate. Irrelevant IgG was used as IP control. One representative western blot of two independent experiments is shown. (I) Immunofluorescence microscopy of myCAFs (CAF01). Representative pictures are shown. Nuclei = blue, TGFBR1 = green, myoferlin (MYOF) = red, scale bar = 50 μm. (J) Proximity ligation assay (PLA) (CAF01) between myoferlin (MYOF) and TGFBR1 in CTRL myCAFs and *MYOF*^KD^ myCAFs (shMYOF#1 and shMYOF#2). PLA control (CTRL^neg^) without primary antibodies is included. Representative pictures are shown. Nuclei = blue, PLA dots = green, scale bar = 10 μm. (K) Quantification of colocalization (tM1 and tM2 Mander's coefficients after threshold regression) shown in (I) and Fig. EV5E. Violin plot. (L) Quantification of PLA dots shown in (J). Violin plot, one-way ANOVA (Tukey's test). (M) Flow cytometry analysis (CAF02) of surface-bound TGFß1 (PE-CF594) in CTRL myCAFs (Irrel. shRNA; n = 3) and *MYOF*^KD^ myCAFs (shMYOF#1 and shMYOF#2; n = 3 each). Mean PE-CF594 signal was expressed as % of control (Irrel. shRNA). Mean ± SEM, one-way ANOVA (Tukey's test). (N) Immunofluorescence microscopy (CAF01) of CTRL myCAFs (Irrel. shRNA) and *MYOF*^KD^ myCAFs (shMYOF#1 and shMYOF#2). Representative pictures are shown. Nuclei = blue, TGFBR1 = green, Golgi apparatus (Golgin-97) = red, scale bar = 50 μm. (O) Object-based distance quantification (DiANA) between TGFBR1 particles and Golgi apparatus shown in Fig. 5N. 1 μm = 11 pixels. Histogram. (P) Immunofluorescence microscopy (CAF01) with object-based colocalization quantification (DIANA) of TGFBR1 vesicles and the Golgi apparatus or endoplasmic reticulum (ER) in CTRL myCAFs (Irrel. shRNA; n ≥ 50), *MYOF*^KD^ myCAFs (shMYOF#1 and shMYOF#2; n ≥ 50 each) and myCAFs treated with Brefeldin A (BFA; n ≥ 35). Representative pictures from 5 distinct observations across two independent wells are shown, n represents individual cells across the 5 fields of view. Nuclei = blue, TGFBR1 = green, Golgi apparatus (GM130) = red, endoplasmic reticulum (KDEL) = grey, scale bar = 10 μm. Violin plot, one-way ANOVA (Sidak's test). Source data are available online for this figure.

and EV5F,G). Finally, non-crosslink coimmunoprecipitation experiments of myoferlin confirmed that myoferlin and Sec24C are part of a same complex, while a close interaction with Sec31 was imperceptible (Fig. 5H). Altogether, these results strongly suggest a direct function of myoferlin in COPII vesicle trafficking likely mediated through Sec24C.

We subsequently evaluated if myoferlin was involved in the COPII-mediated ER/Golgi trafficking of TGFBR1, as the TGFBR1 protein harbors several Sec24C binding sequence (IxM, LxxL and RxxK) (Sato and Nakano, 2007; Chatterjee et al, 2021) in the cytosolic region, mandatory for COPII cargo selection (Fig. 5H). Of note, we were unable to identify Sec24C binding sequences in the cytosolic region of the myoferlin protein, making a bias in MYOF/Sec24C interaction due to myoferlin being a vesicle cargo unlikely. Using immunofluorescence staining of TGFBR1 and myoferlin in myCAFs we found indeed a colocalization between both proteins. Strikingly, the colocalization was restrained to the perinuclear region, supporting a role of myoferlin in TGFBR1 trafficking at the ER/Golgi interface, rather than in other subcellular regions such as the plasma membrane (Figs. 5I,K and EV5E). A close relationship between myoferlin and TGFBR1 in myCAFs was further validated via PLA (Fig. 5J,L), supporting the implication of myoferlin in TGFBR1 trafficking.

As impaired intracellular trafficking of TGFBR1 through myoferlin knockdown reduced SMAD3 phosphorylation upon TGFß stimulation (Fig. 4D), we stipulated that TGFBR1 mis-trafficking resulted in less TGFBR1 at the plasma membrane available for stimulation and signal transduction. Since efficient flow cytometry antibodies against human TGFBR1 were not available, we verified this hypothesis by quantifying the surface abundance of receptor-bound TGFß1. Despite TGFß1 being able to bind to other surface receptors such as ALK1 (Heldin and Moustakas, 2016), unable however to phosphorylate c-terminal SMAD3 (Tarasewicz and Jeruss, 2012), we still detected reduced

TGFß1 surface abundance in *MYOF*^KD^ myCAFs (Fig. 5M). Furthermore, using a phospho-antibody array in combination with TGFß stimulation, we found 3 out of 5 TGFBR1 phosphorylation targets (Vander Ark et al, 2018) were less phosphorylated in *MYOF*^KD^ myCAFs when compared to CTRL myCAFs (Fig. EV5I,J), excluding Shc which is unphosphorylated in both conditions. Surprisingly, SMAD3 S425 phosphorylation was not reduced upon *MYOF*^KD^ as reported earlier (Fig. 4D). This discrepancy can be explained by the simultaneous detection of S423 and S425 phosphorylation sites by western blot (Fig. 4D), while only S425 phosphorylation was evaluated in the phospho-antibody array (Fig. EV5I,J). Finally, we directly assessed the intracellular trafficking of TGFBR1 between the ER and Golgi apparatus. Using object-based distance analysis (DiANA) between TGFBR1 vesicles and the Golgi apparatus we proved that TGFBR1 indeed tightly transits through the Golgi apparatus, as majority of vesicles were located in close proximity ( <11 pixels, 1 μm) to Golgi membranes. Strikingly, however, *MYOF*^KD^ led to the accumulation of TGFBR1 vesicles, notably in a far periphery of the Golgi apparatus as highlighted by increased number of vesicles localized at further distance ( >11 pixels, 1 μm) (Fig. 5N,O), suggesting an incomplete trafficking of TGFBR1 to the Golgi apparatus in *MYOF*^KD^ myCAFs. Incomplete shuttling of TGFBR1 to the Golgi apparatus was further validated using an early Golgi marker (GM130) via quad-IF staining, while a retention of TGFBR1 on the ER (KDEL) was undetectable (Fig. 5P), implying a potential role of myoferlin during the fusion of TGFBR1-carrying COPII vesicles with the Golgi apparatus rather than during the budding from the ER.

Collectively, we hereby identify myoferlin as new component of the COPII vesicle trafficking machinery between the ER and Golgi. Furthermore, myoferlin knockdown leads to a disruption of COPII vesicle trafficking, resulting in impaired shuttling of TGFBR1 to the Golgi apparatus and cell membrane, responsible for reduced TGFß signal transduction.

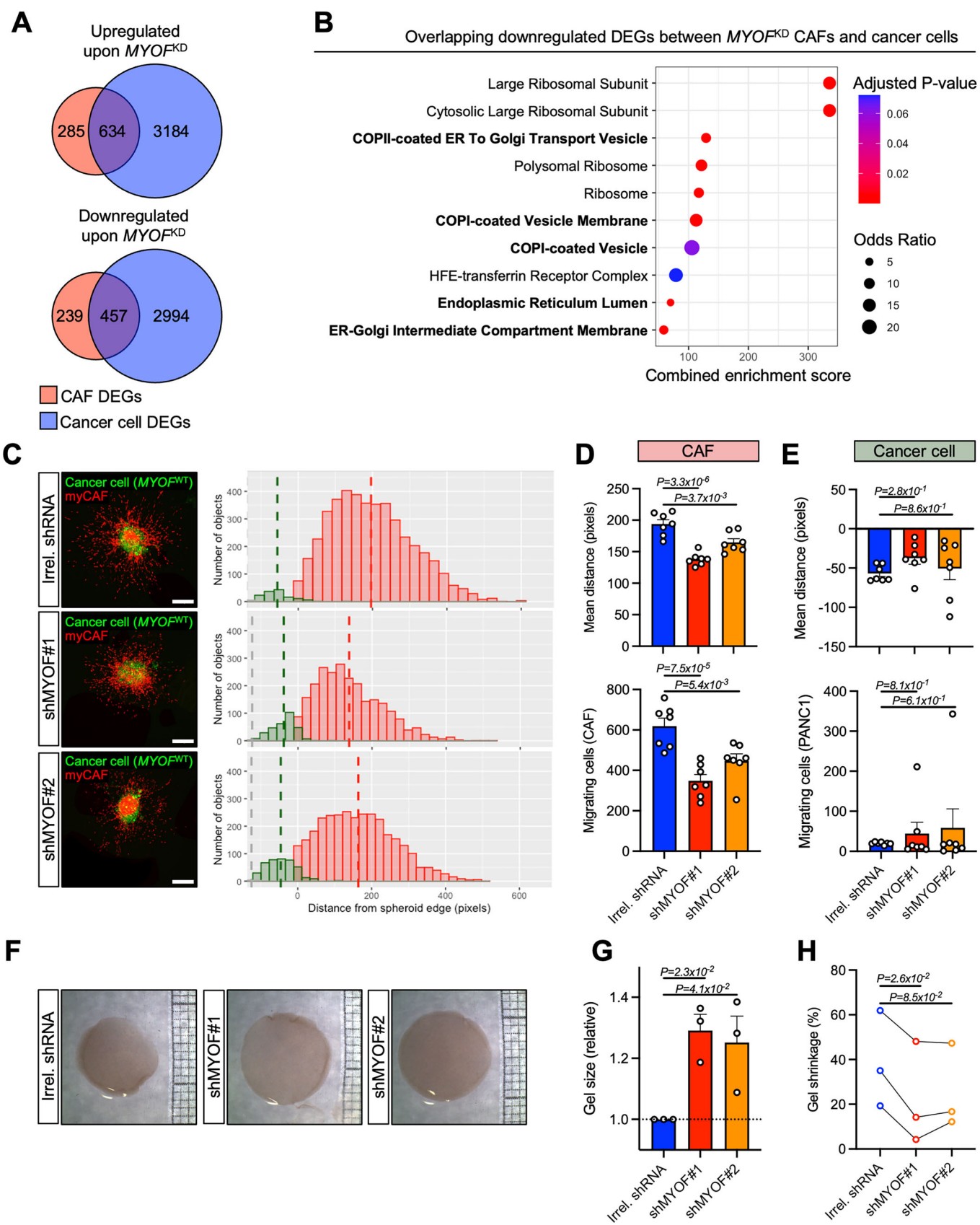

**Figure 6. Myoferlin has CAF-specific functions extending beyond ECM production.**

(A) Venn plots of upregulated and downregulated DEGs ($P < 0.05$) in CAFs and pancreatic cancer cells (PANC1) upon *MYOF*^KD. (B) ORA results of downregulated DEGs overlapping between CAFs and cancer cells. Genesets extracted gene ontology (GO— cellular compartments). (C) 3D-CAF migration assay of heterotopic spheroids generated from *MYOF*^WT cancer cells (PANC-1) in coculture with CTRL myCAFs (Irrel. shRNA; $n = 7$) or *MYOF*^KD myCAFs (shMYOF#1 and shMYOF#2; $n = 7$ each). Representative pictures are shown. Scale bar = 200 µm. (D) Bar plots showing mean migration distance and number of migrating cells quantifications of migrating CAFs from Fig. 6C, one-way ANOVA (Tukey's test). (E) Bar plots showing mean migration distance and number of migrating cells quantifications of *MYOF*^WT cancer cells (PANC-1) from Fig. 6C. One-way ANOVA (Tukey's test). (F) Collagen gel contraction assay of CTRL myCAFs (Irrel. shRNA, $n = 3$) and *MYOF*^KD myCAFs (shMYOF#1 and shMYOF#2, $n = 3$ each). Representative 24 h gel pictures are shown. (G) Quantification of collagen gels shown in (F). Bar plot showing gel size (relative) after 24 h, one-way ANOVA (Tukey's test). (H) Quantification of collagen gel contraction assay shown in (F). Bar plot showing gel shrinkage (%) after 24 h, RM one-way ANOVA (Dunnett's test). Source data are available online for this figure.

## Myoferlin has CAF-specific functions extending beyond ECM production

We unveiled that myoferlin supports COPII-vesicle trafficking and ECM production in CAFs, however myoferlin is also expressed in cancer cells. In contrast to stromal myoferlin, a link between cancer cell myoferlin and patient survival was absent (Fig. 1K,L). To understand whether the function of myoferlin in CAFs could be different to cancer cells, we performed transcriptomic profiling on *MYOF*^KD cancer cells (PANC1) and crossed the differentially expressed genes (DEGs) between *MYOF*^KD and CTRL cancer cells with DEGs from *MYOF*^KD CAFs (Fig. 6A). Overrepresentation analysis (ORA) of overlapping downregulated DEGs revealed that COPII vesicles and ER-related genes are both downregulated in CAFs and cancer cells upon *MYOF*^KD, suggesting that a COPII-related function of myoferlin may be shared between CAFs and cancer cells (Fig. 6B). However, while the overlapping DEGs represent a majority of DEGs in CAFs (65.6%), they represent only a minority of DEGs in cancer cells (13.2%). Accordingly, ORA on cancer cell-specific DEGs showed a downregulation of mitochondria-related genes, in agreement with the described mitochondrial role of myoferlin in pancreatic cancer cells (Rademaker et al, 2018; Anania et al, 2024) (Fig. EV6), suggesting that a COPII-related function of myoferlin is only minor in cancer cells. Taken together, this data supports a cell type specific function of myoferlin, as the main function of myoferlin in CAFs is to support COPII vesicle trafficking, while cancer cell myoferlin is predominantly linked to mitochondrial fitness.

Subsequently, we further investigated potential CAF-specific functions of myoferlin that contribute to tumor aggressiveness. One important aspect of CAF biology is the implication of migrating CAFs in metastasis development and tumor cell dissemination, that can be highly dependent of TGFß (Yoon et al, 2021). Owing to the effect of myoferlin knockdown on TGFß signal transduction in myCAFs, we asked whether myCAF migration was also affected. To address this question, we used a 3D-CAF migration assay by embedding heterotopic spheroids (myCAFs + *MYOF*^WT cancer cells) in a growth factor deprived collagen 1 gel that replicates the PAAD stroma. We indeed validated the important migratory capacity of myCAFs in our model and additionally found that *MYOF*^KD reduced myCAF migration. In fact, heterotopic spheroids generated with *MYOF*^WT cancer cells and *MYOF*^KD CAFs presented fewer migrating CAFs in combination with decreased migration distance compared to CTRL myCAFs (Fig. 6C,D). Importantly, impaired myCAF migration was not compensated with increased cancer cell migration in our model, as cancer cells presented only a poor migratory potential in all conditions (Fig. 6E). myCAFs also

contribute to tumor aggressiveness by acting on ECM stiffness. Like myofibroblasts, myCAFs harbor an actin/myosin contractile apparatus that promotes ECM contraction in response to TGFß signaling, notably through increased aSMA expression, and thus favoring ECM stiffness and increased interstitial fluid pressure (Erdogan and Webb, 2017). We therefore assessed the effect of myoferlin knockdown on myCAF contraction using a collagen 1 gel contraction assay. While CTRL myCAFs presented contractile abilities, which lead to a gel shrinkage above 50%, *MYOF*^KD myCAFs on the other hand presented impaired gel contraction potential (Fig. 6F–H).

Altogether our data suggests that the predominant function of myoferlin is cell type specific. Furthermore, the impact of myoferlin knockdown in CAFs on TGFß signal transduction extends beyond impaired ECM production, as *MYOF*^KD myCAFs bear significantly reduced migratory and contractile abilities, altogether supporting the implication of myoferlin in pro-tumoral stromal and CAF functions.

## Myoferlin targeting tackles tumor desmoplasia in orthotopic KPC mice

Owing to the suggested implication of myoferlin in tumor desmoplasia, we aimed to assess the impact of *MYOF*^KD myCAFs on PAAD tumor biology in vivo. We injected murine pancreatic cancer cells (KPC) orthotopically in *Myof*^WT or *Myof*^KO C57BL/6 mice, resulting in a tumor model with Myof^WT cancer cells and *Myof*^KO stromal cells (Figs. 7A and Fig. EV7A). At sacrifice, no macroscopic metastases were visible (data not shown) and the pancreas weight was unaffected by stromal *Myof*^KO (Fig. 7B). While stromal *Myof*^WT tumors were characterized by intense myoferlin abundance in cancer cells and stromal cells, stromal *Myof*^KO tumors lacked stromal myoferlin (Fig. 7C). Importantly, stromal *Myof*^KO tumors presented significantly reduced desmoplasia, while the abundance of myCAFs (aSMA-positive cells) was not affected (Fig. 7C–E), thus supporting the in vitro data that showed a loss of ECM production without induced cell death in *MYOF*^KD myCAFs (Figs. 3D and EV3F).

As stromal *Myof*^KO impaired tumor desmoplasia in mice, we aimed to assess the effect of myoferlin targeting in a pharmacological model, thus enabling an initial tumor establishment prior to treatment administration (Fig. 7G). Myoferlin inhibitor WJ460 is a small molecule that was first discovered for its anti-metastatic potential in breast cancer(Zhang et al, 2018). While the specificity of myoferlin targeting has already been assessed previously in pancreatic cancer cells (Rademaker et al, 2022), we validated that WJ460 indeed impaired ECM production dose-dependently in

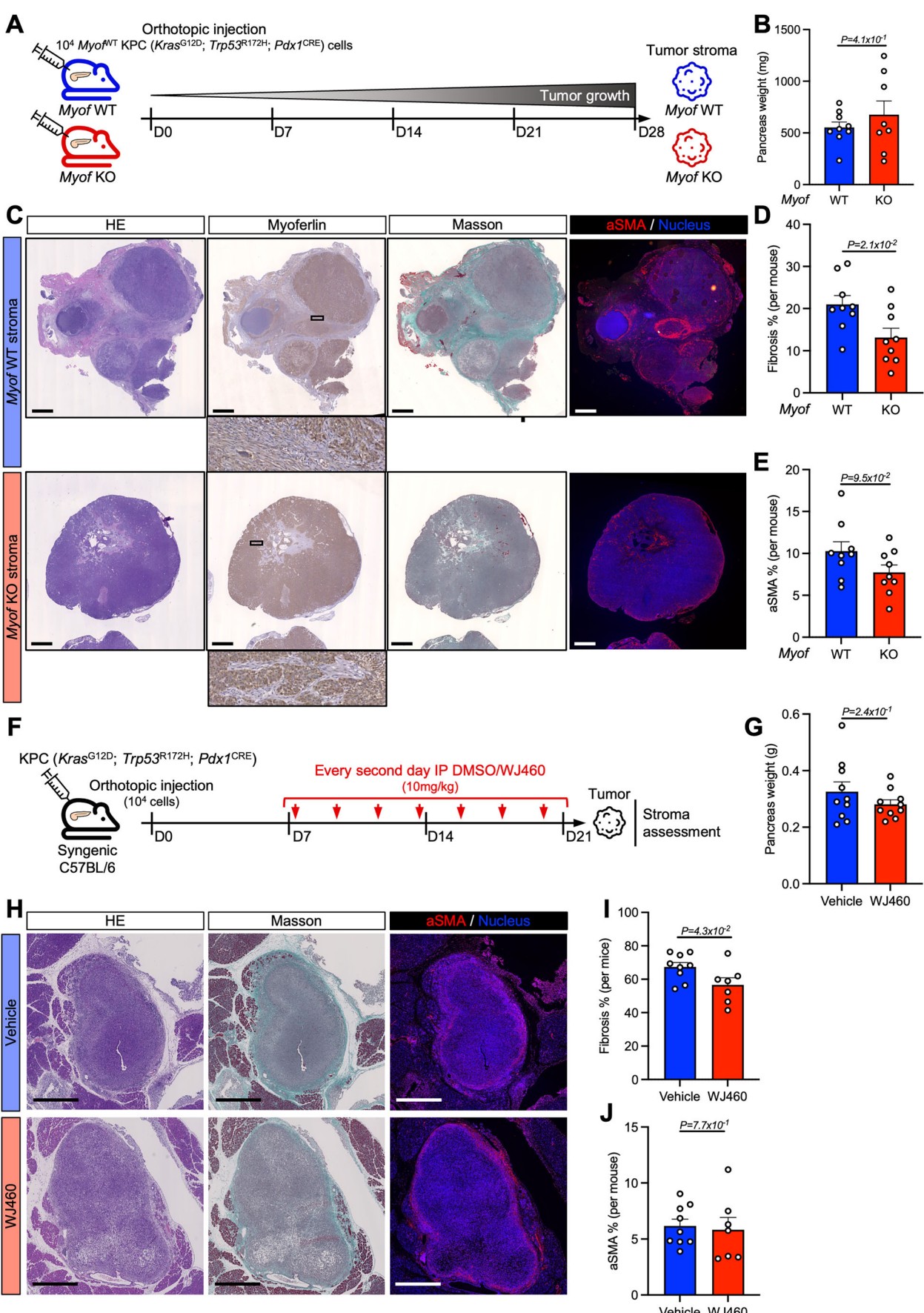

**Figure 7. Myoferlin targeting tackles tumor desmoplasia in orthotopic KPC mice.**

(A) Schematic representation of orthotopic KPC allografts into $Myof^{WT}$ or $Myof^{KO}$ C57BL/6 mice. (B) Pancreas weight of $Myof^{WT}$ ($n = 8$; one mouse succumbed on D27 and tumor was not weighed) and $Myof^{KO}$ ($n = 9$) mice after resection (D28). (C) IHC and immunofluorescence staining of $Myof^{WT}$ ($n = 9$) and $Myof^{KO}$ ($n = 9$) tumor sections. HE = hematoxylin & eosin, Myoferlin, Masson, aSMA = alpha-smooth muscle actin. Scale bar = 1 mm. (D) Quantification of tumor fibrosis (%, Masson-positive area) in $Myof^{WT}$ ($n = 9$) and $Myof^{KO}$ ($n = 9$) tumors. Mean ± SEM. Unpaired $T$ test. (E) Quantification of aSMA-positive area (%) in $Myof^{WT}$ ($n = 9$) and $Myof^{KO}$ ($n = 9$) tumors. Mean ± SEM. Unpaired $T$ test. (F) Schematic representation of orthotopic KPC allografts in C57BL/6 mice and intraperitoneal (IP) DMSO/WJ460 treatment timeline. (G) Pancreas weight (g) of vehicle (DMSO, $n = 10$) and WJ460-treated ($n = 10$) KPC allograft tumor after resection (D21). Unpaired $T$ test. (H) IHC and immunofluorescence staining of vehicle-treated ($n = 9$) and WJ460-treated ($n = 7$) tumor sections. HE = hematoxylin & eosin, Myoferlin, Masson, aSMA = alpha-smooth muscle actin. Scale bar = 1 mm. (I) Quantification of tumor fibrosis (%, Masson-positive area) in vehicle-treated ($n = 9$) and WJ460-treated ($n = 7$) tumors. Mean ± SEM. Unpaired $T$ test. (J) Quantification of aSMA-positive area (%) in vehicle-treated ($n = 9$) and WJ460-treated ($n = 7$) tumors. Mean ± SEM. Unpaired $T$ test. Source data are available online for this figure.

myCAFs, in accordance with $MYOF^{KD}$ (Fig. EV7B–F). WJ460 administration was well tolerated by mice (Fig. EV7G) and did not result in increased pancreas weight, in contrast to what has been previously observed with CAF ablation (Rhim et al, 2014) (Figs. 7G and EV7F). WJ460-treated tumor regions were characterized by reduced tumor fibrosis without impairing aSMA-positive myCAF abundance (Fig. 7H–J), thus supporting the observations made in tumors composed of a $Myof^{KO}$ stroma (Fig. 7C–E). Altogether, the in vivo results above underline the desmoplastic role of myoferlin in CAFs and suggest the pharmacological targeting of myoferlin as new therapeutic approach to tackle tumor desmoplasia in PAAD.

## Discussion

Intracellular vesicle trafficking via the Golgi apparatus is an evolutionarily conserved process with emerging importance in cancer biology (Bui et al, 2021). In this work, we described a pleiotropic expression of the vesicle trafficking protein myoferlin across CAF subtypes in the pancreatic tumor microenvironment and identified stromal myoferlin as driver for tumor aggressiveness. We then revealed a specific function of myoferlin in myofibroblast-like CAFs (myCAFs). Myoferlin participated in COPII-mediated TGFß-receptor 1 (ALK5) trafficking between the ER and Golgi apparatus, and a COPII vesicle blockade via myoferlin knockdown led to a disruption of canonical TGFß signaling and impaired ECM production. Furthermore, we found that myoferlin was involved in several pro-tumoral CAF functions such as migration and contraction. Accordingly, the genetic depletion of myoferlin in the tumor stroma and the pharmacological targeting of myoferlin alike reduced tumor desmoplasia in orthotopic KPC mice (Fig. 8).

We initially found that $MYOF^{high}$ patients presented increased ECM-related gene expression, which was further validated in subTME sections of PAAD patients. Due to the cellular hetero-geneity of the TCGA database, a concise interpretation of transcriptomic data is difficult. However, our results confidently suggest increased ECM abundance in $MYOF^{high}$ patients, as tumor purity and cellularity were homogenous between $MYOF^{low}$ and $MYOF^{high}$ patients. A further investigation of $MYOF$ expression in microdissected pancreatic RNA sequencing data could be helpful to link additional stroma-specific functions to myoferlin (Grünwald et al, 2021). Furthermore, tumor desmoplasia is still regarded as double-edged sword, as excessive ECM deposition has been associated with tumor-promoting and tumor restraining functions (Armstrong et al, 2004; Tian et al, 2021). Our results highlight a correlation between myoferlin and several pro-tumoral ECM components (collagen 1 and fibronectin). A further investigation

of myoferlin in the context of tumor-restraining ECM components, such as collagen XV (Clementz et al, 2013) and biglycan (Weber et al, 2001) is necessary to validate a specific tumor-promoting ECM function of myoferlin. We further elucidated that stromal myoferlin abundance, in contrast to cancer cell myoferlin abundance, impaired patient survival in pancreatic cancer. The absence of a link between cancer cell myoferlin abundance with patient survival was very surprising to us, as previous studies demonstrated that myoferlin ablation in tumor cells impaired tumor size (Gupta et al, 2021; Rademaker et al, 2022). However, it is noteworthy that these studies used subcutaneous xenografts of PAAD cells rather than orthotopic transplantations and thus only poorly reflect PAAD stromal biology. Further research is indeed necessary to assess whether tumor cell and stromal myoferlin collectively contribute to PAAD tumor aggressiveness.

We next identified myoferlin as pleiotropically expressed across CAF subtypes in human and in mice. Intriguingly, myCAFs presented an increase of $Myof$ expression upon lineage progression, potentially because $Myof$ gene transcription is induced by megakaryoblastic leukemia 1 and 2 (MKL1/2) transcription factors (Hermanns et al, 2017), described as crucial transcriptional regulators during TGFß-induced myofibroblast differentiation (Scharenberg et al, 2014). Thus, making $Myof$ a potential response gene of myofibroblast differentiation. The homogeneous expression of myoferlin across distinct CAF subtypes could be highly beneficial during myoferlin targeting in PAAD. In fact, past studies aimed at depleting specific CAF subtypes, such as aSMA+ or Hedgehog+ CAFs, and found increased tumor aggressiveness (Rhim et al, 2014; Özdemir et al, 2014), most likely due to CAF plasticity and tumor infiltration by other CAF subtypes (Steele et al, 2021). Targeting myoferlin in PAAD, and thus impairing multiple CAF subtypes simultaneously, could solve the issue of CAF plasticity. However, the inevitable targeting of myoferlin in tumor restraining CAFs, such as CD105$^{neg}$ CAFs (Hutton et al, 2021), remains a challenge and deserves to be addressed in the future.

In this work, we described a functional implication of myoferlin during TGFß signaling in CAFs, as $MYOF^{KD}$ impaired TGFß signaling and ECM production. However, a link between myoferlin and TGFß signaling has been described previously in breast cancer cells (Barnhouse et al, 2018). In contrast to our findings in pancreatic CAFs, authors showed that SMAD2/3 phosphoryla-tion was unaffected upon myoferlin silencing but revealed an impact of myoferlin on TGFß secretion and autocrine signaling in breast cancer cell lines. Distinct myoferlin functions between neoplastic and non-neoplastic cells are not surprising, due to a highly divergent transcriptional and epigenetic profile between cells.

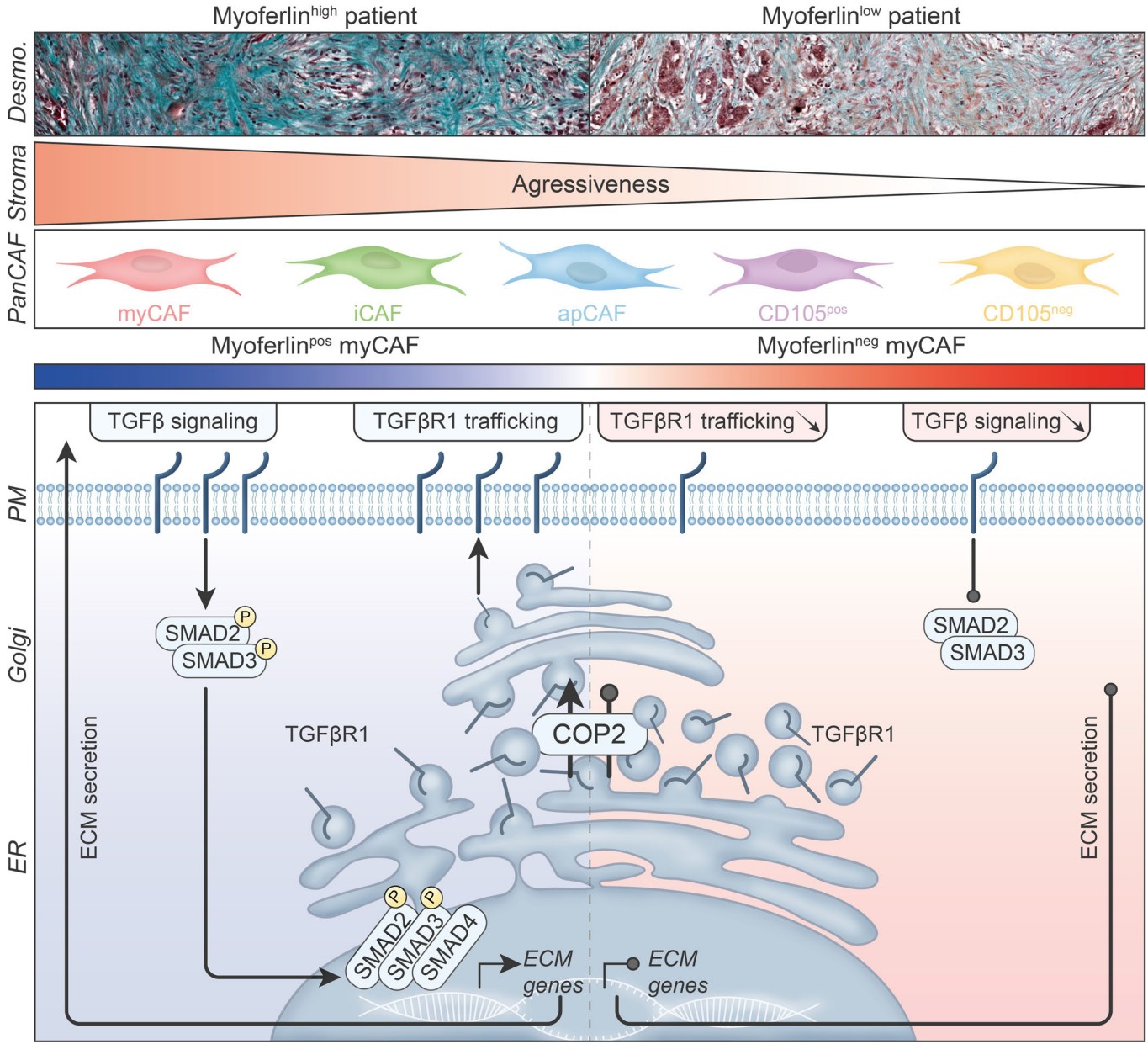

**Figure 8.** Summary of the findings.

We identified myoferlin as member of the COPII vesicle machinery and suggested that TGFBR1 trafficking was impaired by reduced trafficking of COPII vesicles to the Golgi apparatus. Even though our work revealed TGFBR1 in silico and in vitro as a cargo protein of COPII vesicles, the possibility that only TGFBR1 trafficking is affected by a COPII vesicle blockade is highly surprising and unlikely. In fact, it is conceivable that a great variety of transmembrane cargo proteins are affected by myoferlin knockdown. We hypothesize that on whole-cell scale, $MYOF^{KD}$-mediated COPII blockage results in a phenotype dependent of the main transmembrane proteins synthesized by the cell in question. In other words, myCAFs are responsive to TGFß and specialized in ECM production, therefore myoferlin knockdown in

myCAFs perturbed TGFß signaling simply because of the dependency of myCAFs on TGFß receptors. This hypothesis links the effect of myoferlin knockdown in distinct cell types not specifically to myoferlin but rather to the membrane receptor dependency of the cell of interest and deserves to be assessed in the future.

We validated on multiple occasions that ECM gene transcription was indeed impaired in $MYOF^{KD}$ myCAFs, resulting in reduced ECM production. However, it is noteworthy that COPII vesicles are also described to act as ECM carriers and participate in procollagen 1 and fibronectin secretion (Chatterjee et al, 2021; Gorur et al, 2017). Therefore, impaired procollagen 1 and fibronectin secretion caused by $MYOF^{KD}$ mediated COPII blockade could extend the

transcriptional effect of impaired TGFß signaling to a secretory level.

The discussion above suggests that targeting COPII vesicles, or targeting universal vesicle trafficking using Brefeldin A, could be an interesting multilayer approach to tackle cancer, which has indeed been proposed in the past (Goldenring, 2013). However, intracellular vesicle trafficking involves universally expressed proteins such as Rab GTPases (Stenmark, 2009). Targeting intracellular vesicle trafficking in patients could lead to severe side effects caused by impaired vesicle trafficking necessary in healthy tissue, such as synapses in the central nervous system that rely on synaptic vesicle trafficking for neurotransmitter release (Rizo, 2018). Targeting vesicle trafficking through a protein that is specific or overexpressed in cancer, such as myoferlin, represents an innovative approach to target tumoral vesicle trafficking and could potentially reduce the severity of side toxicity.

Finally, we assessed the anti-desmoplastic potential of a stromal $Myof^{KO}$ and the myoferlin-targeting small molecule WJ460 in an orthotopic KPC mouse model and showed indeed reduced tumor desmoplasia. Of note, while depleting stromal myoferlin and impairing tumor desmoplasia alone could result in more aggressive tumors due to improved vascularization and promoted tumor cell dissemination (Rhim et al, 2014; Olive et al, 2009; Provenzano et al, 2012), we did not observe increased tumor burden upon stromal myoferlin depletion. However, the use of cytotoxic agents in combination with myoferlin targeting is likely mandatory for efficient tumor regression. Furthermore, the assessment of immune cell infiltration into WJ460-treated tumors is crucial due to the broad importance of TGFß in the TME and could open valuable opportunities for therapeutic combinations in the future.

# Methods

### Reagents and tools table

| Reagent/resource | Reference or source | Identifier or catalog number |
|---|---|---|
| **Experimental models** | | |
| C57BL6/J (*M. musculus*) | Charles River | N/A |
| CD1 (*M. musculus*) | Charles River | N/A |
| HPSC (*H. sapiens*) | Gift from Atsushi Masamune (Tohoku University, Japan) | N/A |
| KPC (*M. musculus*) | Hingorani et al (2005) Gift from Jonas Van Audenaerde (UAntwerp, Belgium) | N/A |
| Lenti-X 293 T (*H. sapiens*) | Clontech | 632180 |
| PANC-1 (*H. sapiens*) | ATCC | CVCL-0480 |
| **Recombinant DNA** | | |
| Control shRNA plasmid | This study | Vector Builder |
| pLKO.1-puro-CMV-TagRFP | Merck | SHC012 |
| pSPAX2 | Addgene | 12260 |
| shRNA plasmid | This study | Merck |

| Reagent/resource | Reference or source | Identifier or catalog number |
|---|---|---|
| **Antibodies** | | |
| Mouse anti-aSMA (1A4) | Sigma | A5228 |
| Mouse anti-aSMA (1A4) (APC-conjugated) | R&D systems | IC1420A |
| Rabbit anti-CK19 (EP1580Y) | Abcam | ab52625 |
| Rabbit anti-Collagen 1 (EPR7785) | Abcam | ab138492 |
| Rabbit anti-Fibronectin | Abcam | ab2413 |
| Rabbit anti-GAPDH (D16H11) | Cell Signaling Technologies | 5174 |
| Rabbit anti-GM130 (D6B1) | Cell Signaling Technologies | 12480 |
| Mouse anti-Golgin-97 (CDF4) | Invitrogen | 14-9767-82 |
| Mouse anti-HSC70 (B-6) | Santa Cruz | sc-7298 |
| Rabbit anti-IgG (AF488-conjugated) | Invitrogen | A-21206 |
| Mouse anti-IgG (AF546-conjugated) | Invitrogen | A-11030 |
| Mouse anti-IgG (AF750-conjugated) | Invitrogen | A-21037 |
| Rabbit anti-IgG (AF750-conjugated) | Invitrogen | A-21039 |
| Rabbit anti-IgG (biotin-conjugated) | Vector Lab | BA-1000 |
| Mouse anti-IgG (HRP-conjugated) | Dako | P0260 |
| Rabbit anti-IgG (HRP-conjugated) | Cell Signaling Technologies | 7074 |
| Mouse anti-IgG1 Isotype Ctrl (PE-CF594-conjugated) (X40) | BD | 562292 |
| Mouse anti-IgG2a Isotype Ctrl (APC-conjugated) (G155-178) | BD | 555576 |
| Rabbit anti-KDEL (AF488-conjugated) (EPR12668) | Abcam | ab184819 |
| Rabbit anti-Lamin B1 (EPR22165-121) | Abcam | ab229025 |
| Rabbit anti-MST1 | Cell Signaling Technologies | 3682 |
| Rabbit anti-MST2 | Cell Signaling Technologies | 3952 |
| Rabbit anti-Myoferlin | Sigma | HPA014245 |
| Mouse anti-Myoferlin (D-11) | Santa Cruz | sc-376879 |
| Rabbit anti-P-SMAD2 S465/467 (138D4) | Cell Signaling Technologies | 3108 |
| Rabbit anti-P-SMAD3 S423/425 (C25A9) | Cell Signaling Technologies | 9520 |
| Rabbit anti-P-YAP S127 (D9WI) | Cell Signaling Technologies | 13008 |
| Rabbit anti-Sec24C (D9M4N) | Cell Signaling Technologies | 14676 |
| Rabbit anti-COPII (Sec31) | Invitrogen | PA1-069A |
| Rabbit anti-SMAD2/3 (D7G7) | Cell Signaling Technologies | 8685 |

| Reagent/resource | Reference or source | Identifier or catalog number |
|---|---|---|
| Rabbit anti-SMAD4 (D3R4N) | Cell Signaling Technologies | 46535 |
| Mouse anti-TEF-1 (31/TEF-1) | BD | 610922 |
| Mouse anti-TGFB1 (PE-CF594-conjugated) (TW4-9E7) | BD | 562422 |
| Rabbit anti-TGFBR1 | Sigma | SAB4502958 |
| Rabbit anti-TGFBR2 (E5M6F) | Cell Signaling Technologies | 41896 |
| Rabbit anti-YAP | Cell Signaling Technologies | 4912 |
| Rabbit anti-YAP/TAZ (D24E4) | Cell Signaling Technologies | 8418 |
| **Oligonucleotides and other sequence-based reagents** | | |
| qPCR primers | This study | Appendix Table S1, Eurogentec |
| sgRNA | This study | Synthego |
| siRNA | This study | Eurogentec |
| **Chemicals, enzymes and other reagents** | | |
| Bodipy 493/593 | Invitrogen | D3922 |
| Brefeldin A | BioLegend | 420601 |
| Cas9 | IDT | 1081058 |
| Cycloheximide | Sigma | C7698 |
| Cytofix/Cytoperm kit | BD | 554714 |
| CytoPainter green | Abcam | ab138891 |
| DMEM | Biowest | L0101 |
| DMEM-F12 | Biowest | L0093 |
| dNTP mix | Thermo Fisher | R0191 |
| Duolink In Situ Detection Reagents Green | Merck | DUO92014 |
| Duolink In Situ PLA probes | Merck | DUO92002 DUO92004 |
| FBS | Sigma | F7524 |
| Fixable Viability Stain 780 | BD | 565388 |
| Fluorescence Mounting Medium | Agilent | S3023 |
| Fluoromount-G | Invitrogen | 00-4958-02 |
| Hoechst 33258 | Thermo Fisher | H1398 |
| iSpacers | Sunjin Lab | IS317 |
| L-glutamine | Biowest | X0550 |
| Lentivirus Titration (Titer) Kit | ABM | LV900 |
| Lipid (Oil Red O) staining kit | Sigma | MAK194 |
| M-MLV reverse transcriptase | Promega | M1701 |
| M-MLV RT buffer | Promega | M5312 |
| Masson-Goldner kit | Merck | 100485 |
| Matrigel | Corning | 354234 |
| mRNA Prep Ligation kit | Illumina | 20040532 |
| MycoAlert™ PLUS | Lonza | LT07-710 |
| non-fat dry milk | AppliChem | A0830 |

| Reagent/resource | Reference or source | Identifier or catalog number |
|---|---|---|
| NucleoSpin RNA plus | Macherey-Nagel | 740984 |
| Phalloidin | Invitrogen | R415 |
| Pierce BCA protein assay kit | Thermo Fisher | 23227 |
| Pierce ECL Western Blotting Substrate | Thermo Fisher | 32106 |
| Ponceau S solution | Sigma | P7170 |
| PrestoBlue Cell Viability reagent | Invitrogen | A13261 |
| protein A/G magnetic beads | Thermo Fisher | 88803 |
| Puromycine | Sigma | P9620 |
| PVDF membranes | Sigma | 03010040001 |
| Random hexamers | Thermo Fisher | SO142 |
| Rat tail collagen 1 | Merck | 08-115 |
| Ribolock RNase inhibitor | Thermo Fisher | EO0381 |
| RNA 6000 Nano kit | Agilent | 5067-1511 |
| Rompun (xylazine) | Bayer | N/A |
| Super RX films | Fujifilm | |
| SuperSignal West Atto | Thermo Fisher | A38554 |
| Takyon Low ROX SYBR master mix | Eurogentec | UF-LSMT-B0701 |
| TGF beta signaling phospho antibody array | Tebubio | PTG176 |
| TGFß1 | Bio-Rad | PHP143B |
| TRIzol reagent | Thermo Fisher | 15596026 |
| Vectastain ABC | Vector | PK-4000 |
| WJ460 | MedChem Express | HY-124632 |
| **Software** | | |
| Cellranger v7.0.1 | https://www.10xgenomics.com | |
| Clusterprofiler v3.18.1 | https://www.bioconductor.org | |
| Coloc2 v3.1.0 | https://imagej.net/plugins/coloc-2 | |
| DESeq2 v4.0 | https://www.bioconductor.org | |
| DiANA | https://imagej.net/plugins/distance-analysis | |
| Dplyr v1.1.4 | https://cran.r-project.org | |
| Enrichplot v1.10.2 | https://www.bioconductor.org | |
| Fiji v2.9.0 | https://imagej.net/ | |
| ggplot2 v3.4.4 | https://cran.r-project.org | |
| Graphpad Prism v10 | https://www.graphpad.com | |
| Harmony v0.1 | https://github.com/immunogenomics/harmony | |
| Icy v2.5.2.0 | https://icy.bioimageanalysis.org/ | |
| Incucyte analysis software | Sartorius | |

| Reagent/resource | Reference or source | Identifier or catalog number |
|---|---|---|
| nf-core RNAseq v3.0 | https://nf-co.re/rnaseq | |
| Pheatmap v1.0.12 | https://cran.r-project.org | |
| Plyr v1.8.9 | https://cran.r-project.org | |
| QuPath v0.5.1 | https://qupath.github.io | |
| R v4.0.4 and v4.3 | https://www.r-project.org | |
| Seurat v3 and v5 | https://satijalab.org/seurat | |

## Patient samples

Paraffin-embedded tissues from PAAD patients were stored at the institutional biobank (Biothèque Hospitalo-Universitaire de Liège, BHUL) with informed patient consent. According to Belgian law, patients were informed that the residual surgical material could be used for research purposes and the consent is presumed if the patient does not oppose. Access to patient samples for immuno-histological analysis was granted by the Human Ethic Committee of the University of Liège and from the University Hospital for project number 2016-270.

## Mice and in vivo experiments

5-week-old male C57BL/6J mice and were purchased from Charles River. Mice were enrolled in experiments between 6 and 10 weeks of age. All animals were kept in pathogen-free ventilated cages at the animal facility of the University of Liège. Animals were allowed unrestricted access to food and water; room temperature was stable at 21 °C and circadian rhythm was fixed by a 12 h light/dark cycle. Animal well-being was monitored daily. All animal experimentations were approved by the Institutional Animal Care and Ethics Committee of the University of Liège, under file number 20-2271. Orthotopic transplants were performed by injecting mice with $10^4$ murine KPC cells at 6 weeks. Mice that were used for WJ460 (HY-124632, MCE) treatments were randomly assigned to one experimental group and intraperitoneally injected with 50 μL of 2% DMSO (vehicle group) or 10 mg/kg WJ460 (treated group) diluted in corn oil 1 week after surgery. Mice were treated every second day for 2 weeks until euthanasia. For tumors in $Myof^{KO}$ mice, tumors were collected 28 days post orthotopic transplantation of KPC cells.

### CRISPR-Cas9 mediated generation of $Myof^{KO}$ mice
C57BL/6J mice were used for embryo collection and pseudo-pregnant recipient preparation. All animal procedures to generate F1 hetero-zygous animals were approved by the Institutional Animal Care and Use Committee of the University of Liège, under file number 20-2271.

Two $Myof$-specific single guide RNAs (sgRNAs) were designed using Chopchop (sgRNA1: TGACTTGAGGGGGGATACCAC, sgRNA2: AAGTTCAATAGCTTCGCGGA) and purchased from Synthego. Fertilized one-cell embryos were collected from super-ovulated female mice. A mixture of Cas9 protein (IDT, 4 μM) and both sgRNA (4 μM) was electroporated using the Gene X Cell Electroporator (Bio-Rad). Injected zygotes were cultured to the 2-cell stage and transferred into the oviducts of pseudo-pregnant CD1 females to obtain F0 founders.

Genomic DNA was extracted from tail biopsies using the NucleoSpin Tissue kit (Macherey-Nagel). The targeted region was PCR-amplified and analyzed by Sanger sequencing to confirm indels and/or deletions at the CRISPR target sites.

### Orthotopic transplantations
Orthotopic injections were performed using a previously described procedure (Lee et al, 2021). The day prior to surgery, cells were seeded into 60 mm dishes to obtain 80% confluency the following day. The day of surgery, cells were detached using trypsin and counted. Cells were washed once in PBS, $10^4$ KPC cells were resuspended in 50 μL PBS and kept on ice until injection. 15 min prior to surgery, mice were injected intraperitoneally with 100 μL of the analgesic 2% Rompun xylazine (Bayer), further diluted 20× in PBS. Mice were then anesthetized using isoflurane/$O_2$ mixture and kept anesthetized while being immobilized on a heating pad during the entire procedure, eyes were kept humidified using sterile PBS. The left abdominal flank was shaved and disinfected using 70% ethanol. The spleen was localized through the skin and a median laparotomy was performed by making incisions through the skin and peritoneum using surgical scissors and forceps. Then, using forceps, the spleen was externalized to expose the tail of the pancreas. 50 μL of cell suspension were injected into the tail of the pancreas using 26 G/1 mL syringes (303176, BD Plastipak), successful injection was evaluated by the formation of a bleb within the pancreas. The needle was slowly removed to avoid leakage and the pancreas was left for 30 s. Mice that presented clear signs of leakage were excluded from the experiment. The organs were carefully replaced into the abdominal cavity before the peritoneum and skin were sutured using Mersilk (FW528, Ethicon). The site of incision was cleaned up with isobetadine and mice were kept on a heating pad during recovery. Mice weight and health was monitored daily. At the time of experimental endpoint, mice were anesthetized and euthanized by cervical dislocation. Tumors were resected, weighted, and stored for further analysis.

## Immunohistochemistry and immunofluorescence of human and murine tumor samples

Human PAAD tumor samples were obtained from the BHUL biobank as 5 μm thick formalin-fixed paraffin-embedded (FFPE) sections. Fresh murine tumors were fixed overnight in 4% paraformaldehyde and then embedded in paraffin. 5 μm FFPE sections were cut on a microtome and mounted on Superfrost slides. For IHC staining, FFPE sections were dehydrated overnight at 56 °C, then deparaffinated and rehydrated using consecutive baths as follows: $3 × 5$ min xylene, 3 min 100% methanol, 3 min 95% methanol, 3 min 70% methanol, 3 min 50% methanol, 5 min $H_2O$, 10 min PBS-0.25% Triton X-100, 2 min PBS. Then, endogenous peroxidases were inhibited by washing slides for 1 min in 100% methanol, followed by a 30 min incubation in the dark with 3% $H_2O_2$. Slides were then washed for 5 min in $H_2O$. Heat-induced epitope retrieval was performed by incubating slides in a citrate buffer (10 mM sodium citrate, pH 6) for 40 min at 95 °C. Then, non-specific binding sites were blocked using serum from the same species as the secondary antibody (1.5% serum, 0.1% Tween, PBS) for 30 min. After blocking, slides were incubated with primary antibodies diluted in 1.5% serum-supplemented PBS overnight at 4 °C. After washing, slides were incubated with biotinylated secondary antibodies diluted in 1.5% serum-supplemented PBS for 30 min at room temperature. During secondary antibody incubation, avidin biotin

complex (ABC) was prepared as described by the manufacturer (Vectastain ABC kit, PK-4000, Vector Laboratories). After secondary antibody incubation, slides were washed and incubated in ABC solution for 30 min at room temperature. Finally, slides were incubated with $DAB/H_2O_2$ solution for 3 min and then washed for 5 min under tap water. After washing, slides were counterstained with hematoxylin for 40 s, washed with $H_2O$ and mounted.

Masson's trichrome staining was performed using the Masson-Goldner kit (100485, Merck) according to the manufacturer's recommendations.

For immunofluorescence staining of murine tumor FFPE sections, the original IHC protocol was slightly adapted. Briefly, after deparaffination and rehydration, heat-induced epitope retrieval was performed in a citrate buffer (10 mM sodium citrate, pH 6) by boiling slides for $2 \times 3$ min using a microwave. Sections were then permeabilized for $2 \times 10$ min in permeabilization buffer (PBS, 0.2% gelatin, 0.25% Triton X-100, pH 7.4). Next, slides were blocked in 5% BSA-supplemented permeabilization buffer for 1 h. Primary antibodies were diluted in 1% BSA-supplemented permeabilization buffer overnight at 4 °C. After washing, slides were incubated with fluorescent secondary antibodies diluted in 1% BSA-supplemented permeabilization buffer for 45 min at room temperature. Hoechst 33258 (H1398, Thermo Fisher) was used as nuclear stain. Slides were then washed $3 \times 10$ min with PBS and washed further 10 min in wash buffer (10 mM $CuSO_4$, 50 mM $NH_4Cl$). Finally, slides were rinsed with $H_2O$ and mounted with Fluoromount-G mounting medium (00-4958-02, Invitrogen). Mounted slides were scanned using an Olympus VS200 slide scanner.

### Patient scoring

SubTME regions of patient slides stained for myoferlin abundance with IHC or with Masson Trichrome for collagen abundance were scored independently by three distinct investigators (RP, YB, OP). For each subTME a score between 1 (low staining intensity) and 3 (high staining intensity) was given.

To calculate whole-patient scores for stromal or cancer-cell myoferlin abundance, the staining intensity (scores 0 to 3) in combination with staining extent (%) were evaluated independently, and blindly from Masson Trichrome score, by 4 distinct investigators (RP, GR, OP, MH). Whole patient scores were obtained by multiplying the mean extent (ME) with each score:

$$\frac{(0 \times ME(0)) + (1 \times ME(1)) + (2 \times ME(2)) + (3 \times ME(3))}{100}$$

### Quantification of murine tumors

All quantifications were performed using QuPath v0.5.1 (Bankhead et al, 2017). Tumor regions in each pancreas were determined based on histology.

Global collagen and aSMA abundance were evaluated on Masson's Trichrome-stained slides or aSMA-stained slides in tumoral regions using the QuPath random tree pixel classifier, trained with 50 regions for each annotation. Each annotation extend was expressed as % of total tumor area for each tumor.

### Bioinformatic processing of human PAAD bulk RNAseq data

TCGA (Hoadley et al, 2018) transcriptomic and clinical data from PAAD patients was retrieved from cBioportal (https://

www.cbioportal.org) (Cerami et al, 2012). Patients with confirmed ductal PAAD ($n = 146$) were segregated into groups based on z-score normalized *MYOF* expression. Differentially expressed genes, including log$_2$FC and q-value, were obtained by comparing *MYOF*$^{high}$ patients with *MYOF*$^{low}$ patients in cBioportal. Kaplan–Meier curves were calculated based on clinical data from TCGA in Graphpad Prism (v10). Further processing of transcriptomic data was performed in R (v4.0.4) and gene set enrichment analysis (GSEA) of differentially expressed genes ($P < 0.05$) was carried out using the R packages clusterprofiler (v3.18.1) and enrichplot (v1.10.2). Heatmaps were generated using the R package pheatmap (v1.0.12). For each patient, hypoxia scores (Buffa et al, 2010) and stromal/immune scores (Yoshihara et al, 2013) were retrieved using cBioportal (Cerami et al, 2012) and the ESTIMATE database (https://bioinformatics.mdanderson.org/estimate/), respectively. ABSOLUTE (Carter et al, 2012) purity scores and tumor subtypes for each patient were extracted from the TCGA PAAD paper (Raphael et al, 2017).

Transcriptomic data from different PAAD stroma subtypes (Moffitt et al, 2015) was obtained on Gene Expression Omnibus (https://www.ncbi.nlm.nih.gov/geo/) using accession number GSE71729. *MYOF* expression in samples was retrieved using the NCBI tool GEO2R (https://www.ncbi.nlm.nih.gov/geo/geo2r/).

*MYOF* and COPII-related gene expression was compared between cancer tissue (TCGA) and respective healthy data (GTEx) for several organs on Gepia (https://gepia.cancer-pku.cn).

Differential gene expression data between CyTOF-isolated CD105$^{pos}$ and CD105$^{neg}$ CAFs was obtained from the source publication (Hutton et al, 2021).

### Bioinformatic processing of human PAAD scRNAseq data

Access to a published human PAAD scRNAseq dataset (Elyada et al, 2019), via accession number phs001840.v1.p1, was granted from NCBI dbGaP (https://www.ncbi.nlm.nih.gov/gap/) for project #36038. First, Fastq files were counted and aggregated in Cellranger (v7.0.1). Reads were then mapped to the hg38 reference genome (refdata-gex-GRCh38-2020-A) and Cellranger counts were used to create an object in Seurat (v5) with 29943 features across 28507 samples. Metadata was added to the Seurat object, linking sample names to clinical information. Source tables were: "SraRunTable.csv" and "sra/files/phs001840.v1.pht009149.v1.p1.c1.PDAC_CAFs_Sample_Attributes.GRU.txt.gz". Further quality control, processing and data analysis was performed in R (v4.3) using the Seurat package (v5). Out of 28507 initial cells, cells with less than 500 genes or 50000 counts were filtered out. Additionally, cells with more than 10000 genes or 15% mitochondrial genes were removed, resulting in 27517 cells. From 29943 genes, poorly expressed genes (<10 counts in all cells) were also filtered out, leaving 26571 genes for further analysis. For batch correction, the Seurat object was spit across samples, regressed, PCA reduced and integrated using Harmony (v0.1). Further dimensionality reduction was carried out via UMAP (principal components 1-30) and the data was clustered using the Louvain algorithm (Blondel et al, 2008) with a resolution of 0.4. Broad clusters were labelled with markers from the original publication (Elyada et al, 2019) (Fig. EV2A). For fibroblast subclustering (Fig. EV2B,C), the data was integrated by "subject_id" and reduced (dimensions = 30, Resolution = 0.4). To ensure correct labeling of fibroblast subclusters, CAF markers from Elyada et al (Elyada et al, 2019) were complemented with pericyte markers (*RGS5* and *MCAM*)

(Baek et al, 2022). CD105 positive CAFs were clustered based on *ENG* gene expression (Hutton et al, 2021). For further analysis, the data was log-normalized and scaled using default parameters. Differential gene expression between groups was analyzed using the "FindMarkers" function from Seurat and statistical significance of pairwise comparisons was assessed with Wilcoxon Rank test and Bonferroni correction. Finally, normalized *MYOF* expression levels were extracted from all cells and visualization was performed on GraphPad Prism (v10).

## Bioinformatic processing of murine PAAD scRNAseq data

A publicly available murine PAAD scRNAseq dataset (Dominguez et al, 2020) from KPP mice (*Pdx1*cre/+; *LSL-Kras*G12D/+; *p16/p19*flox/flox) was downloaded from the ArrayExpress database (https://www.ebi.ac.uk/biostudies/arrayexpress) using accession number E-MTAB-8483. Data analysis was performed in R (v4.0) using the Seurat package (v3). Quality control and preprocessing were carried out according to the original publication. First, from 14926 initial cells, only cells expressing between 1200 and 5000 genes were retained, while dead cells were filtered out (>3% mitochondrial DNA content), resulting in 13454 cells for further analysis. A potential batch effect between replicates was corrected using Harmony (v0.1). After batch correction, dimensionality reduction was performed via t-SNE (principal components 1–20) and clusters were identified using the default algorithm in Seurat (k = 40, Resolution = 0.7), which resulted in 13 clusters. Clusters were then labelled using markers described by Dominguez and colleagues (Dominguez et al, 2020) (Fig. EV2E–G). Differential gene expression between groups was analyzed using the "FindMarkers" function from Seurat and statistical significance of pairwise comparisons was assessed with Wilcoxon Rank test and Bonferroni correction. For the evaluation fibroblasts according to *Eng* expression, all fibroblast clusters were grouped and then segregated bason on *Eng* positivity. Finally, log-normalized *Myof* expression values were extracted from all cells and visualization was performed on GraphPad Prism (v10). t-SNE plots for specific marker genes were retrieved form the Single Cell Expression Atlas (https://www.ebi.ac.uk/gxa/sc/home).

## Cell culture

Immortalized human pancreatic stellate cells (PSCs) [HPSC127 (male IPMC, CAF01 hereafter, mKate red-fluorescent nucleus), HPSC21 (female PAAD, CAF02 hereafter), HPSC128 (male PAAD) and HPSC130 (female PAAD)] were established at Division of Gastroenterology, Tohoku University Graduate School of Medicine, Japan. Human pancreatic adenocarcinoma cells PANC-1 were purchased from ATCC (Manassas, VA). PSCs were cultured in DMEM-F12 (L0093, Biowest), supplemented with 2mM L-glutamine (X0550, Biowest) and 10% heat-inactivated FBS (F7524, Sigma), unless stated otherwise (hereafter referred to as CAF medium). For quiescent PSC generation, $4 \times 10^4$ PSCs were embedded in 50 µL Matrigel (354234, Corning) domes in low-binding 12 well plates to avoid attachment (174931, Thermo). PSCs were considered quiescent after 4 days. For myCAFs, PSCs were cultured as described previously (Biffi et al, 2019). Human PAAD cells PANC-1 (RRID: CVCL-0480) were maintained in high-glucose DMEM (L0101, Biowest) supplemented with L-glutamine

at 2 mM and 10% heat-inactivated FBS. PANC-1 cells were authenticated by STR profiling (DSMZ, Braunschweig, Germany). Mouse KPC (Hingorani et al, 2005) cells (*Kras*G12D; *Trp53*R172H; *Pdx1*CRE) were a gift from Dr Van Audenaerde (UAntwerp, Belgium). KPC cells were maintained in high-glucose DMEM (L0101, Biowest) supplemented with 2 mM L-glutamine and 10% heat-inactivated FBS. All cells were cultured at 37 °C in a humidified 5% $CO_2$ incubator, checked regularly for mycoplasma contamination and used between passage 1 and 10.

### 3D spheroid culture and coculture

For spheroids, $2.5 \times 10^3$ myCAFs were seeded per well in 96-well ultra-low attachment U-bottom plates (174925, Thermo) and centrifuged for 5 min at $200 \times g$. Spheroids were incubated at 37 °C with 5% $CO_2$. 12 h after seeding, when spheroids had formed, was considered as t0 for growth curve experiments. For immuno-fluorescent imaging, spheroids were fixed for 1 h at room temperature with 4% paraformaldehyde 72 h after seeding. For spheroid coculture, $2 \times 10^3$ myCAFs were seeded together with $0.5 \times 10^3$ cancer cells (PANC-1) and processed the same as spheroids monocultures. Spheroid cocultures were incubated for 96 h prior to fixation for immunofluorescent imaging.

### In vitro treatments

Myoferlin inhibitor WJ460 (HY-124632, MCE) stock solution was prepared by dissolving 50 mg in 250 µL DMSO (434 mM). A diluted stock solution was prepared in DMSO at 1 mM. A further dilution to 10 µM was performed in CAF medium, which was then used to treat cells at concentrations described in the figures. TGFß1 100 ng/µL stock solution was prepared by dissolving recombinant human TGFß1 (PHP143B, Bio-Rad) in $H_2O$ and sterilized through a 0.2 µm filter. TGFß1 stock solution was further diluted in PBS-BSA 1% to 1 ng/µL. For TGFß1 stimulations, cells were starved overnight with 0.5% FBS and stimulated the next day with TGFß1 at a 5 ng/mL final concentration. Cycloheximide (CHX) (C7698, Sigma) 5 mg/mL stock solution was prepared in $H_2O$ and sterilized using a 0.2 µm filter. Cells were treated at a 10 µg/mL final concentration. Brefeldin A (BFA) (420601, Biolegend) was resuspended as a stock solution at 5 mg/mL in DMSO. For TGFBR trafficking studies, cells were treated for 30 min with BFA working solution (5 µg/mL). All other treatments were performed between 0 h and 48 h.

## siRNA transfection and shRNA lentiviral transduction

For transient gene knockdown, cells were transfected with 40 nM (Irrel. siRNA: CUUACGCUGAGUACUUCGAT; *MYOF*-specific siRNA#1: CCCUGUCUGGAAUGAGAUUU; Eurogentec, Seraing, Belgium) using the calcium phosphate method. Briefly, siRNAs were diluted to 40 nM in sterile $H_2O$ supplemented with 250 µM $CaCl_2$ and vortexed intermittently ten times. Then, HBSP 2x (280 mM NaCl, 50 mM HEPES, 10 mM KCl, 12 mM D-glucose, 1.5 mM $Na_2HPO_4$) was added in a spiral motion and gently homogenized. After 1 min of incubation, the transfection mix was deposited on the cell monolayer. Sixteen hours after transfection, cells were washed three times with PBS and fresh medium was added. Another 48 h later, cells were stopped for further analysis.

To achieve stable gene knockdown, cells were transduced with shRNA plasmids. *MYOF* shRNA lentiviral plasmids were purchased from Merck (*MYOF*-specific shRNA#1: CCCTGTCTGGAATGAGATT and

shRNA#2: GAAAGAGCTGTGCATTATAAA). Two controls were also used: Irrelevant shRNA (Vector Builder - Target Sequence against luciferase TGTCCGGTTATGTAAACAATC) and pLKO.1-puro-CMV-TagRFP (SHC012, Merck) as transduction control. All plasmids contain a puromycin resistance gene under the control of a PGK promoter. Briefly, Lenti-X 293T cells (632180, Clontech) were co-transfected with pSPAX2 (Addgene) and VSV-G encoding vectors (Emi et al, 1991). Viral supernatants were collected 48 h, 72 h and 96 h post transfection, filtrated (0.2 μM) and concentrated 100x by ultracentrifugation. Lentiviral vectors were then titrated using the qPCR Lentivirus Titration (Titer) Kit (LV900, ABM). In total, $1 \times 10^5$ cells were transduced with lentiviral vectors (30 TU/cell) and then selected with 10 mg/mL puromycin (P9620, Sigma). The absence of replication competent lentivirus and mycoplasma in cell supernatant was confirmed using the qPCR Lentivirus Titration kit (LV900, ABM) and MycoAlert™ PLUS Mycoplasma Detection Kit (LT07-710, Lonza), respectively. Transduced cells were used as heterogeneous population as no clonal expansion was performed. Puromycin selection was repeated monthly.

## RNA isolation of monolayer and 3D Matrigel culture

For adherent culture, cells were washed 1× with PBS and total RNA was extracted using the NucleoSpin RNA plus kit (740984, Macherey-Nagel), according to manufacturer's recommendations.

For Matrigel-embedded qPSCs, total RNA was extracted using a combination protocol of TRIzol reagent (15596026, Thermo Fisher) and the NucleoSpin RNA plus kit. Briefly, Matrigel domes were dissociated using 1 mL TRIzol and incubated for 5 min at RT. Then, 200 μL chloroform were added, the mixture homogenized, and incubated for 2–3 min at RT. The lysate was centrifuged at $12,000 \times g$ for 15 min at 4 °C. In all, 350 μL of the upper aqueous phase were mixed with 100 μL BS buffer (Macherey-Nagel) and subjected to the NuceloSpin procedure. Finally, RNA quantity and purity were evaluated using a Nanodrop spectrophotometer.

## Bioinformatic analysis of *MYOF*$^{KD}$ CAF and cancer cell whole-genome transcriptomic profiling data

### mRNA library generation

For mRNA library generation, $5 \times 10^5$ cells were seeded in 60-mm dishes. myCAFs from four distinct donors were used to achieve increased biological significance. For cancer cells, PANC1 were seeded as three biological replicates. The following day, myoferlin knockdown was performed using siMYOF#1. Forty-eight hours after washing of wells, total RNA was extracted as described above. RNA quality was assessed by determining RNA integrity numbers for each sample using the Agilent RNA 6000 Nano kit (5067-1511, Agilent). Next, a mRNA library was generated using the Illumina stranded mRNA Prep Ligation kit (20040532, Illumina) according to manufacturer's instructions and sequencing was performed on an Illumina Novaseq6000 using a S4 flowcell with a depth of 2x20M reads of 150 bp per sample.

### Bioinformatic processing

Novaseq reads were processed using the nf-core RNAseq pipeline (Ewels et al, 2020) (v3.0; https://nf-co.re/rnaseq). Briefly, read alignment and quantification were performed with the "STAR-Salmon" option, using the human GRCh38 genome build associated with annotations from the Ensembl release 097 (https://www.ensembl.org). Downstream analysis was carried out

using the DESeq2 package (Love et al, 2014) in R (v4.0), including clustering, principal component analysis and differential expression analysis between *MYOF*$^{KD}$ and CTRL cells. For *MYOF*$^{KD}$ CAF gene set enrichment analysis, differentially expressed genes ($P < 0.05$) including logFC values were analyzed using the R packages clusterprofiler (v3.18.1) and enrichplot (v1.10.2). Venn plots including overrepresentation analysis (ORA) of overlapping DEGs between *MYOF*$^{KD}$ CAFs and cancer cells were performed using the "Set Comparison Appyter" (https://appyters.maayanlab.cloud/#/CompareSets) and Enrichr (https://maayanlab.cloud/Enrichr/) from the Ma'ayan Laboratory.

## Oil Red O staining

Lipid droplets were visualized in 2D and 3D cultures using the Lipid (Oil Red O) staining kit (MAK194, Sigma). Monolayer (myCAF) and Matrigel cultures (qPSC) were fixed for 30 min with 4% paraformaldehyde. Cells were then washed twice for 5 min with 60% isopropanol and incubated with Oil Red O working solution for 15 min. Oil Red O working solution was freshly prepared by diluting Oil Red O stock solution to 60% using water. After staining, cells were washed five times with $H_2O$ and pictures were acquired under a brightfield microscope.

## Cell proliferation analysis

For cell proliferation analysis, $10^3$ cells were seeded per well into a 96-well plate. Wells were directly imaged using an IncuCyte S3 (Sartorius). Cell proliferation was quantified based on confluency. All quantifications were performed on the Incucyte analysis software (Sartorius).

## IC$_{50}$ analysis

Cells were seeded in 96-well plates at $7 \times 10^3$ cells/well. The next day, cells were treated with increasing concentrations (0–100 nM) of WJ460 for 24 h or 48 h. Cell viability was assessed using the PrestoBlue Cell Viability Reagent (A13261, Invitrogen) as recommended by the manufacturer. Log$_{10}$ normalizations and IC$_{50}$ calculations were performed on GraphPad Prism (v10).

## RT-qPCR

cDNA was synthesized using an in-house cDNA synthesis protocol. 4 μg RNA were mixed with 1 μM random hexamer primers (SO142, Thermo Fisher) in a final volume of 50 μL and incubated at 65 °C for 5 min. After chilling on ice, M-MLV RT buffer (M5312, Promega) at 1×, Ribolock RNase inhibitor (EO0381, Thermo Fisher) at 2U/μL, dNTP mix (R0191, Thermo Fisher) at 0.5 mM each and M-MLV reverse transcriptase (M1701, Promega) at 10U/μL were added to the RNA mix to obtain a final volume of 80 μL. The reaction was performed by incubating the mix for 5 min at 25 °C, 60 min at 42 °C and 5 min at 70 °C.

qPCR was performed in 96-well plates (AB1400, Thermo Fisher). For every reaction, 100 ng cDNA was mixed with 1 μM forward and reverse primers (Appendix Table S1) in Takyon Low ROX SYBR master mix (UF-LSMT-B0701, Eurogentec) in a final volume of 20 μL. Plates were read on a QuantStudio Real-Time PCR system (A28567, Applied Biosystems). Amplification conditions were initiated with 10 min denaturation at 95 °C, followed by

40 cycles of 15 s denaturation at 95 °C and 1 min annealing/elongation at 60 °C. Results were analyzed using the ΔΔCT method, 18S ribosomal RNA was used for normalization.

## Western blot

Proteins were extracted from adherent cells using 1% SDS supplemented with phosphatase inhibitors (5 mM NaF, 1 mM $Na_3VO_4$) and protease inhibitors (3 μM aprotinin, 80 μM leupeptin, 60 μM pepstatin A and 40 μM PMSF). Cells were scraped in lysis buffer and protein lysates were sonicated for 2–3 s at 5 watts using a VibraCell ultrasonic liquid processor (Sonics). Protein concentration was determined using the Pierce BCA protein assay kit (23227, Thermo Fisher).

For SDS-PAGE, 15 μg of protein extracts were diluted with 1% SDS and supplemented with Laemmli's buffer in a final volume of 15 μL. Samples were denatured at 99 °C for 5 min and then loaded on polyacrylamide gels. Migration was performed at 100 V for 2 h, followed by wet transfer onto PVDF membranes (03010040001, Sigma) at 100 V during 90 min. Membranes were then blocked for 1 h in TBS-T supplemented with 5% non-fat dry milk (A0830, AppliChem). Primary antibodies were diluted according to manufacturer recommendation and PVDF membranes were incubated with diluted antibodies overnight at 4 °C under agitation. HRP-linked secondary antibodies were diluted 1/3000 and the membranes incubated for 1 h at room temperature under agitation. For revelation, membranes were incubated for 1 min with Pierce ECL western blotting Substrate (32106, Thermo Fisher) and then exposed in a dark room using radiographic films (Super RX, Fujifilm). HSC70 served as loading control. All signals were quantified via densitometric analysis using Fiji (v2.9.0) (Schindelin et al, 2012).

## Conditioned media experiments

For conditioned media collection, $6 \times 10^5$ cells were seeded into 60 mm dishes and cultured in reduced serum conditions (DMEM-F12, supplemented with 2mM L-glutamine and 0.5% heat-inactivated FBS). To remove dead cells and debris, conditioned media was centrifuged at $250 \times g$ during 5 min after collection and then stored at −20 °C. For western blot analysis of ECM components in conditioned media, 16 μL media samples were supplemented with 4 μL Laemmli's buffer and then boiled at 99 °C for 5 min. Subsequent western blot was performed as described above, except that SuperSignal West Atto (A38554, Thermo) was used as ECL substrate.

### Ponceau S staining

As loading control, membranes were stained with Ponceau S. Prior to blocking, PVDF membranes were washed twice in $H_2O$. For staining, membranes were incubated during 5 min under agitation in Ponceau S solution (P7170, Sigma) and then washed with $H_2O$ until background clearance. Membranes were scanned for image acquisition before undergoing blocking and antibody incubations as described above.

## Cycloheximide chase analysis

In all, $2 \times 10^5$ cells were seeded into six-well plates. Forty-eight hours later, cells were stimulated with TGFß1 (5 ng/ml) while protein synthesis was blocked with cycloheximide (CHX; 10 μg/ml). Cells were harvested at time points 0 h, 0.5 h, 1 h, 2 h, 4 h, 8 h and

proteins were extracted using SDS 1%, as described above. Protein half-life and stability was assessed via western blot.

## Prediction of transcription factor binding sites

Gene-specific promoter sequences were retrieved from the *Homo sapiens* reference genome (GRCh38) using the UCSC genome browser (Karolchik et al, 2004) (https://genome.ucsc.edu). Promoter regions were defined as 2000bp upstream of the first exon. Within each promoter sequence, transcription factor binding sites were predicted using Jaspar (Castro-Mondragon et al, 2022) (https://jaspar.genereg.net). Binding sites with a profile score below an 80% threshold were excluded from the analysis.

## Flow cytometry

### Bodipy staining of lipid droplets

Lipid droplets of myCAFs were stained with Bodipy 493/593 (D3922, Invitrogen) according to a described procedure (Qiu and Simon, 2016). Briefly, Bodipy stock solution was prepared at 1 mg/mL in ethanol. Adherent cells were stained for 15 min at 37 °C with Bodipy staining solution (2 μM in PBS). Then, cells were washed, trypsinized and mean FITC-A was quantified per cell using a Cytoflex flow cytometer.

### Staining of surface and intracellular markers

In all, $6 \times 10^5$ cells were seeded into 60-mm dishes and cultured for 48 h. At 80% confluency, cells were detached using trypsin and $3 \times 10^5$ cells were resuspended in 200 μL PBS per condition. Cells were then incubated with viability staining (1/1000) (565388, BD Horizon) and surface antibodies or isotype control (1/50) in 50 μL FACS buffer (PBS with 1% FBS and 1% EDTA) for 30 min at 4 °C. After incubation, cells were washed with 150 μL FACS buffer. Then, cells were fixed and permeabilized using the Cytofix/Cytoperm kit (554714, BD). Intracellular antibodies and isotype controls were diluted 1/50 in Cytoperm buffer (554714, BD) and cells were incubated for 30 min at 4 °C. Finally, cells were washed 2× in Cytoperm buffer and acquisition was performed on a FACSCanto II.

## Immunofluorescent confocal microscopy of spheroids

Immunofluorescent staining and confocal microscopy on spheroids was performed by adapting a described procedure (Gonzalez et al, 2021). Spheroids were fixed for 1 h at room temperature using 4% paraformaldehyde. Next, spheroids were incubated with permeabilization/blocking (PB) solution (PBS supplemented with 1% Triton X-100, 1% DMSO and 1% BSA) for 1 h at room temperature with gentle horizontal agitation (30–50 rpm). Primary antibodies were diluted 1/100 in diluted PB solution (10%) and spheroids were incubated for 24 h at 4 °C with gentle horizontal agitation (30–50 rpm). After primary antibody incubation, spheroids were washed with PBS-BSA 0.1% 5 times for 3 min and two times for 15 min. Fluorescent secondary antibodies were diluted 1/250 in diluted PB solution in combination with 1 μg/mL Hoechst 33258 (H1398, Thermo Fisher) for nuclear staining. Spheroids were incubated with secondary antibody/Hoechst solution for 24 h at 4 °C with gentle horizontal agitation (30–50 rpm) and protected from light. Spheroids were then washed with PBS-BSA 0.1% five times for 3 min and 2 times for 15 min and incubated in clearing

solution (85% glycerol in $H_2O$) for 12 h at room temperature in the dark. Finally, spheroids were mounted within clearing solution using iSpacers (IS317, Sunjin Lab) and kept in the dark until imaging. Spheroids were imaged on a Nikon AR1 or Leica Stellaris 8 confocal microscope using a ×20 objective (pinhole = 38.3 µm). For each spheroid, mean fluorescence intensity was quantified individually per channel on raw images using Fiji (v2.9.0) (Schindelin et al, 2012).

## Immunofluorescent confocal microscopy of adherent cells

In all, $4 \times 10^4$ cells were seeded on sterile glass coverslips. After 48 h, cells were either directly fixed or treated according to experimental conditions and then fixed using 4% paraformaldehyde for 10 min at room temperature. Cells were then permeabilized (PBS 1% Triton X-100) for 5 min and blocked (PBS-BSA 1%) for 30 min. Primary antibodies were diluted 1/100 in PBS-BSA 1%, and coverslips were incubated overnight at 4 °C in a wet chamber. After washing, fluorescent secondary antibodies were diluted 1/1000 in PBS-BSA 1% with 1 µg/mL Hoechst 33258 (H1398, Thermo Fisher) for nuclear staining. Coverslips were incubated for 45 min at room temperature in a wet chamber and protected from light. For cytoskeleton imaging, cells were stained with Phalloidin, as described by the manufacturer (R415, Invitrogen). Finally, coverslips were mounted onto glass slides using fluorescence mounting medium (S3023, Agilent). Coverslips were imaged on a Nikon A1R or Leica Stellaris 8 confocal microscope using a 63x oil-immersion objective. Image analysis was performed on raw images using Fiji (v2.9.0) (Schindelin et al, 2012).

For quad-IF, coverslips underwent three consecutive cycles of fixation and antibody incubation. First, coverslips were fixed with 4% paraformaldehyde for 10 min at room temperature, permeabilized (PBS 1% Triton X-100) for 5 min and blocked (PBS-BSA 1%) for 30 min. Coverslips were incubated with the first primary antibody (TGFBR1) overnight at 4 °C and then exposed to the secondary antibody for 45 min. For the second cycle, coverslips were fixed again with 4% paraformaldehyde and then incubated with the second primary antibody (GM130) overnight at 4 °C, before being incubated with the second secondary antibody for 45 min. Finally, a third cycle was performed by fixing the coverslips and incubating with the third primary antibody (KDEL-AF488) overnight at 4 °C before being incubated with Hoechst, mounted and imaged.

### Bodipy staining of lipid droplets
Lipid droplets of myCAFs were stained with Bodipy 493/593 (D3922, Invitrogen) according to a described procedure (Qiu and Simon, 2016). Briefly, Bodipy stock solution was prepared at 1 mg/mL in ethanol. Adherent cells were stained for 15 min at 37 °C with Bodipy staining solution (2 µM in PBS). Then, cells were washed and FITC-A signal was imaged using a Leica Stellaris 8 confocal microscope.

### Pixel-based colocalization quantification
All colocalization studies were performed on raw confocal images. Mander's coefficients were calculated after Costes threshold regression using the Fiji plugin Coloc 2 (https://imagej.net/plugins/coloc-2). For colocalization with Golgi apparatus, each Golgi apparatus was selected as individual ROI prior to analysis.

### Object-based colocalization quantification (SODA)
As pixel-based colocalization studies can by heavily biased by background noise, we also performed cell-by-cell object-based colocalization using the statistical object distance analysis (SODA) (Lagache et al, 2018) suite in Icy (de Chaumont et al, 2012) (v2.5.2.0). Colocalization quantification was performed as described by Lagache and colleagues, with COPII vesicles in channel 1 and Golgin97 in channel 2. Ripley's K function was used to compare object distribution with a random distribution. The maximal distance for the analysis was fixed to 5 pixels. Colocalization ratios for each cell were exported from Icy and visualized in Graphpad Prism (v10).

### Object-based distance quantification
All distance quantifications were performed on raw confocal images using the Fiji plugin DiANA (Gilles et al, 2017) (https://imagej.net/plugins/distance-analysis). For efficient object segmentation, image channels were split, denoised using median filtering (radius=1.0) and binarized with a global threshold (TGFBR1 = 60; Golgin97 = 30). Objects touching XY edges or objects with a size smaller than 3 pixels were excluded from segmentation. Next, object-based distance analysis was performed by measuring the center-center distance (pixels) of each object in channel A (TGFBR1) to the nearest object in channel B (Golgin97). Frequency distribution and histogram plotting for each condition was performed in Graphpad Prism (v10). For TGFBR1, GM130 and KDEL colocalization, a specific pipeline including auto-thresholding was used (https://gitlab.uliege.be/Olivier.Peulen/jython_colocalization_gm130_kdel_tgfbr). For TGFBR1 channel thresholding was preceded by an 8 pixel top-hat filtering and an object segmentation based on maximum intensity detection. Minimal distance between each TGFBR1 object and reticulum (KDEL) or Golgi (GM130) object edge was calculated.

## Proximity ligation assay

For proximity ligation assay (PLA), $4 \times 10^4$ cells were seeded on sterile glass coverslips. PLA was performed 48 h after seeding, as described by the manufacturer, using Duolink In Situ PLA probes (DUO92002 and DUO92004, Merck) and Duolink In Situ Detection Reagents Green (DUO92014, Merck). Briefly, coverslips were fixed using 4% paraformaldehyde for 10 min at room temperature, then permeabilized (PBS 1% Triton X-100) for 5 min at room temperature and finally blocked (Duolink blocking solution) for 30 min at 37 °C in a wet chamber. Primary antibodies were diluted 1/100 in Duolink Antibody Diluent and coverslips were incubated overnight at 4 °C in a wet chamber. For negative controls, coverslips were incubated with Duolink Antibody Diluent without primary antibodies. The next day, coverslips were washed (Wash Buffer A) and incubated with Duolink PLA probes for 1 h at 37 °C. After 2 more washings (Wash Buffer A), coverslips were incubated with ligase for 30 min at 37 °C. Subsequently, coverslips were washed 2 additional times (Wash Buffer A) and incubated with polymerase for 100 min at 37 °C. After amplification, coverslips were washed twice (Wash Buffer B) and incubated with Hoechst 33258 (1 µg/ml in PBS; H1398, Thermo Fisher) during 15 min at room temperature for nuclear staining. Finally, coverslips were washed 3 times with diluted Wash Buffer B (1/100) and mounted onto glass slides with fluorescent mounting medium (S3023, Agilent). Images were acquired on a Leica Stellaris 8

confocal microscope using a ×63 oil-immersion objective. Image analysis was performed on Fiji (v2.9.0) (Schindelin et al, 2012).

### PLA dots quantification

All studies were performed on raw confocal images using Fiji (v2.9.0). First, channels were split, and the green channel was binarized using the Otsu threshold (Otsu, 1979). Binary pictures were then processed with median filtering (pixel radius = 2) to reduce noise. Finally, PLA dots were counted using the "Analyze Particles" command (Size = 0-infinity; Circularity = 0.0-1.0). For each field, nuclei were counted manually. PLA dots and nuclei that were in contact with image borders were excluded from the analysis. Average PLA dots per nucleus were calculated for each field by dividing the total number of PLA dots by the total amount of nuclei.

### Phospho-antibody array

For phospho-antibody array analysis, $5 \times 10^6$ myCAFs (Irrel. shRNA and shMYOF#1) were seeded into 10 cm dishes. 48 h after seeding, cells were stimulated with human recombinant TGFß1 for 8 h. After treatment, cells were washed with ice cold PBS, detached using trypsin and $5 \times 10^6$ cells were pelleted per condition. The cell pellet was washed 3 times with ice cold PBS and finally stored at −80 °C until analysis.

TGF beta signaling phospho antibody array (PTG176, Tebubio) was carried out by Tebubio as recommended by the manufacturer. For analysis, data was normalized to the median signal of the slide. For each pair of phospho-antibody (P) and non-phospho-antibody (N), the P/N ratio was determined using the average signal intensity of replicate spots. Finally, P/N ratio fold change was calculated by comparing shMYOF#1 samples to irrel. shRNA. Targets with a fold change between ±0.01 were considered irrelevant and excluded from the analysis.

### Subcellular fractionation

In all, $5 \times 10^6$ cells were starved overnight in 0.5% FBS and stimulated the following day with human TGFß1 (5 ng/mL) for 60 min. Then, cells were washed 1× in PBS, trypsinized, counted and centrifuged (5 min at $250 \times g$). For plasma membrane lysis (cytosolic fraction) the cell pellet was resuspended in 400 µL/$10^7$ cells of buffer A (10 mM HEPES pH 7.9, 10 mM KCl, 2 mM MgCl$_2$, 0.1 mM EDTA pH8, 0.2% NP-40, 1 mM DTT, 1× protease/phosphatase inhibitor cocktail). After a 10 min incubation on ice, the lysate was centrifuged at 14000 g for 5 min at 4 °C and the supernatant (cytosolic fraction) was collected. Next, the pellet was washed 1x with Buffer A and then resuspended in 150 µL/$10^7$ cells of buffer B (20 mM HEPES pH 7.9, 1.5 mM MgCl$_2$, 0.2 mM EDTA pH8, 630 mM NaCl, 25% glycerol, 0.5 mM DTT, 1x protease/phosphatase inhibitor cocktail). For nuclear lysis, the lysate was incubated for 2 h at 4 °C under rotation. Finally, the lysate was centrifuged for 5 min at $14,000 \times g$ at 4 °C and the nuclear fraction) was collected. Prior to SDS-PAGE, the protein concentration in samples is quantified via BCA assay (23227, Thermo Fisher).

### Coimmunoprecipitation

In all, $5 \times 10^6$ cells were lysed on ice with 300 µL freshly prepared ND buffer (20 mM Tris-HCl pH 8, 137 mM NaCl, 1% NP-40, 2 mM EDTA pH8, 1x protease inhibitor cocktail). For mechanical lysis, the cell lysate was run 10× through a Dounce tissue grinder (Wheaton) on ice and then incubated for 30 min at 4 °C under rotation. After incubation, the lysate was centrifuged at $14,000 \times g$ for 10 min at 4 °C and the supernatant was recovered. The protein concentration of the lysate was determined via BCA assay (23227, Thermo Fisher). For coimmunoprecipitation, protein A/G magnetic beads (88803, Thermo Fisher) were washed 1x in ND buffer and then blocked overnight with BSA at 4 °C under rotation. Simultaneously, 500 µg of lysate were incubated with 1 µg target antibody (Myoferlin antibody, mouse IgG1 clone D-11, sc-376879, Santa Cruz) or control IgG (mouse IgG isotype control, 02-6502, Invitrogen) overnight at 4 °C under rotation. The next day, 30 µl of protein A/G magnetic beads were added to the lysate-antibody mix and were incubated for further 2 h at 4 °C under rotation. Using a magnet, the protein-antibody-bead complex was washed 2x in ice-cold wash buffer (1% Triton X-100, 2 mM EDTA, 20 mM Tris-HCl pH 8, 100 mM NaCl). For elution, the protein-antibody-bead complex was resuspended in 30 µL of Laemmli's Buffer (supplemented with 1% SDS) and heated for 10 min at 99 °C. Finally, beads were removed using a magnet and the samples subjected to western blot.

### 3D invasion assay

While myCAFs were obtained as stable transduced cell line with mKate (red fluorescent), PANC-1 cells ($6 \times 10^6$ cells/mL) were stained using CytoPainter green (ab138891, Abcam) for 30 min. For spheroid generation, 2000 myCAFs were seeded together with 500 PANC-1 cells per well into 96-well ultra-low attachment U-bottom plates (174925, Thermo) and centrifuged for 5 min at $200 \times g$. Twelve hours after seeding, once spheroids had formed, spheroids were embedded in a DMEM-based 2.5% collagen 1 (rat-tail, 08-115, Merck) gel, without serum (due to the acidic properties of collagen 1, color-based pH adjustment was done using NaOH). During collagen gel solidification, culture plates were flipped 3 times for 1 min to avoid spheroid precipitation and attachment. Collagen gels were cultured in 35 mm imaging dishes (81156, Ibidi) using FBS-supplemented (10%) CAF culture medium, acting as chemoattractant. Seventy-two hours after spheroid embedding, cell migration was evaluated using a Nikon A1R confocal microscope. To reduce bias related to limited migrated cells per focal plane, images were acquired as Z-stack, covering a maximum of spheroid depth. Image stacks were flattened by maximum-intensity Z-projection and migrated cells were counted in each channel using a Fiji script. Quantification and visualization of cell migration was done in R (v4.0).

### Gel retraction assay

In all, $3 \times 10^5$ cells were embedded in 330 µL of a DMEM-based rat-tail collagen (08-115, Merck) gel (2.5% collagen, 1% Pen/Strep, color-based pH adjustment was done with NaOH). Gels were poured into round sterile plastic forms within a 6-well plate. The plate was incubated 3 min at 37 °C to launch the polymerization process. Then the plate was flipped upside down $3 \times 1$ min and incubated for further 20 min at 37 °C. Once the gel had polymerized, the plastic forms were retrieved and t0 pictures were taken. Finally, 2 mL CAF medium supplemented with 5 ng/mL TGFß1 were added into each well and the gels were detached from the wells using tips. Experimental endpoint was 24 h and

gel size was measured using Fiji (v2.9.0) (Schindelin et al, 2012). Gel shrinkage (%) was calculated using the following formula: $\left(\frac{a-b}{a}\right) \times 100$ in which a = initial gel size (μm²) and b = 24 h gel size (μm²).

## Statistical analysis

Statistical analyses were carried out in GraphPad Prism (v10) or on R (v4.0) using the packages ggplot2 (v3.4.4), dplyr (v1.1.4) and plyr (v1.8.9). Replicate numbers and appropriate statistical tests for each experiment can be found in figure legends. Globally, qualitative variables were treated with Fishers exact test, while quantitative variables were compared using one-way ANOVA (including Tukey's or Dunnet's multiple comparisons). Correlation coefficients were calculated in R using Pearson and Spearman procedures. Kaplan–Meier curves were generated in GraphPad Prism (v10), and $P$ values calculated with the Log rank (Mantel–Cox) test. Statistical significance for all experiments was defined as $P < 0.05$.

## Graphics

Some of the figure panels (Figs. EV3A, 3A and 7A,F) were created with www.biorender.com.

# Data availability

The bulk RNAseq data discussed in this publication have been deposited in NCBI's Gene Expression Omnibus and are accessible through GEO Series accession numbers: GSE306771; GSE306772.

The source data of this paper are collected in the following database record: biostudies:S-SCDT-10_1038-S44318-025-00570-6.

# Peer review information

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

## Acknowledgements

We kindly thank all patients that donated samples and made the human aspects of this study possible. The authors express their acknowledgement to the excellent support provided by the institutional Biobank (BHUL), the ULiège animal housing staff and GIGA technical facilities (University of Liège). Notably, Dr. Chantal Humblet from the Histology platform, Dr. Emmanuel Di Valentin with M. Francois Giroulle and Ms. Alexandra Revnic from the Viral Vectors platform, Dr. Sandra Ormenese with M. Alexandre Hego and M. Gaëtan Lefevre from the GIGA-Imaging platform, Dr. Wouter Coppieters from the Genomics platform, Dr. Arnaud Lavergne from the Bioinformatics platform. We also thank Dr. Helene Pendeville-Samain from the Gene Editing platform for the generation of *Myof*^KO mice via CRISPR. Furthermore, authors thank Dr. Jonas Van Audenaerde (UAntwerpen, Belgium) for gifting KPC cells and Prof. Patrick Jacquemin with Ms. Margaux Wulleman (De Duve Institute, UCLouvain, Belgium) for technical guidance with orthotopic pancreas transplantations in mice. The authors also thank M. Patrick Roncarati for assistance with QuPath image analysis. Finally, authors thank M. Louis Baudin (Animascience) for the design of the graphical summary. The results in this work are in part based on data generated by the TCGA Research Network: www.cancergenome.nih.gov. This work was supported by the "Fondation Léon Frédericq", the Julia Russe prize, the University of Liège "Fonds spéciaux crédits sectoriels (CSRV-SS)" and the "Fonds de la Recherche Scientifique (FNRS)" grant N° J.0167.24. RP is a Research Fellow (FNRS, grant N° 40010385), RC is a "Télévie" postdoctoral fellow, GR is a Postdoctoral Researcher (FNRS, grant 40024125), MH is a Research Associate (FNRS) and AB is a Research Director (FNRS). EL and AG are supported by the Luxembourg National Research Fund (FNR). AM is supported by a JSPS KAKENHI grant (N° 23K24312).

## Author contributions

**Raphaël Peiffer**: Conceptualization; Data curation; Formal analysis; Investigation; Visualization; Methodology; Writing—original draft; Writing—review and editing. **Emilie Laverdeur**: Validation; Investigation; Methodology. **Anthoula Gaigneaux**: Software; Formal analysis; Visualization. **Yasmine Boumahd**: Investigation. **Charlotte Gullo**: Investigation. **Gilles Rademaker**: Resources; Investigation. **Rebekah Crake**: Investigation. **Arnaud Lavergne**: Software; Formal analysis. **Naïma Maloujahmoum**: Resources. **Ferman Agirman**: Resources. **Michael Herfs**: Resources; Supervision; Methodology. **Atsushi Masamune**: Resources. **Elisabeth Letellier**: Resources; Supervision; Investigation. **Akeila Bellahcène**: Supervision; Funding acquisition. **Olivier Peulen**: Conceptualization; Resources; Supervision; Funding acquisition; Investigation; Project administration; Writing—review and editing.

Source data underlying figure panels in this paper may have individual authorship assigned. Where available, figure panel/source data authorship is listed in the following database record: biostudies:S-SCDT-10_1038-S44318-025-00570-6.

## Disclosure and competing interests statement

The authors declare no competing interests.

# Expanded View Figures

**Figure EV1. Myoferlin correlates with stromal features in pancreatic cancer.**

(A) *MYOF* expression in human matched healthy and neoplastic tissue, ranked according to tumor *MYOF* expression. PAAD = pancreatic adenocarcinoma, KIRC = kidney renal clear cell carcinoma, STAD = stomach adenocarcinoma, CHOL = cholangiocarcinoma, OV = ovarian serous cystadenocarcinoma, GBM = glioblastoma, THYM = thymoma, LAML = acute myeloid leukemia, DLBC = diffuse large B-cell lymphoma, LGG = lower grade glioma. Patient numbers are indicated for each group. Boxplot (P25-1.5*IQR; P25; Median; P75; P75 + 1.5*IQR). *$P < 0.01$. (B) TCGA PAAD cohort patient characteristics, low *MYOF* ($n = 36$) versus high *MYOF* ($n = 37$) patients. Chi-squared test for demographic characteristics and TNM stage; Wilcoxon test for diagnosis age. (C) Ridgeplot of most enriched gene sets ($P < 0.0001$) in *MYOF*high patients. (D) ESTIMATE immune scores in TCGA PAAD cohort patients ($n = 146$) according to *MYOF* expression. Violin plot, one-way ANOVA (Tukey's test). (E) Percentages of tumor subtypes (Collisson et al, 2011) according to *MYOF* expression (QM = quasi-mesenchymal). Stacked bar plot, Chi-squared test. (F) Percentages of tumor subtypes (Moffitt et al, 2015) according to *MYOF* expression. Stacked bar plot, Chi-squared test. (G) Percentages of tumor subtypes (Bailey et al, 2016) according to *MYOF* expression. Stacked bar plot, Chi-squared test. (H) Myoferlin IHC and Masson trichrome staining (collagens stained in blue) in human PAAD sections. Representative images from 99 subTME regions across 33 patients. Scale bar = 100 μm. (I) IHC analysis, correlation and linear regression between Masson trichrome scores and Myoferlin stromal scores ($n = 99$). Pearson (R) and Spearman (ρ) correlation. (J) Internal PAAD cohort patient characteristics, *MYOF*low stroma ($n = 19$) versus *MYOF*high stroma ($n = 19$) patients. Chi-squared test for demographic characteristics and TNM stage; Wilcoxon test for diagnosis age and death/censoring age. (K) Linear regression between stromal myoferlin IHC scores and tumor cell myoferlin IHC scores of PAAD patients ($n = 38$). Pearson (R) and Spearman (ρ) correlation. Source data are available online for this figure.

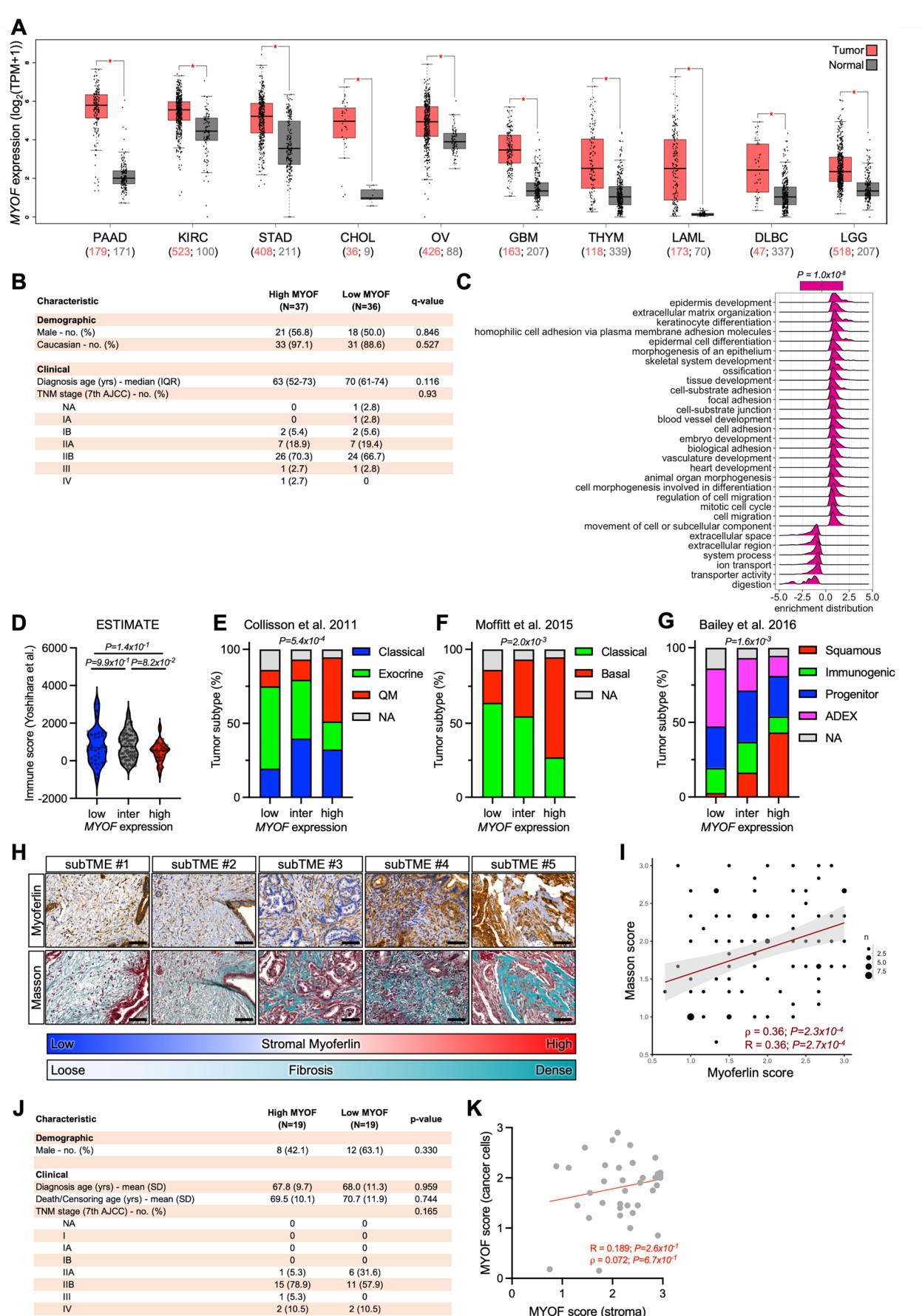

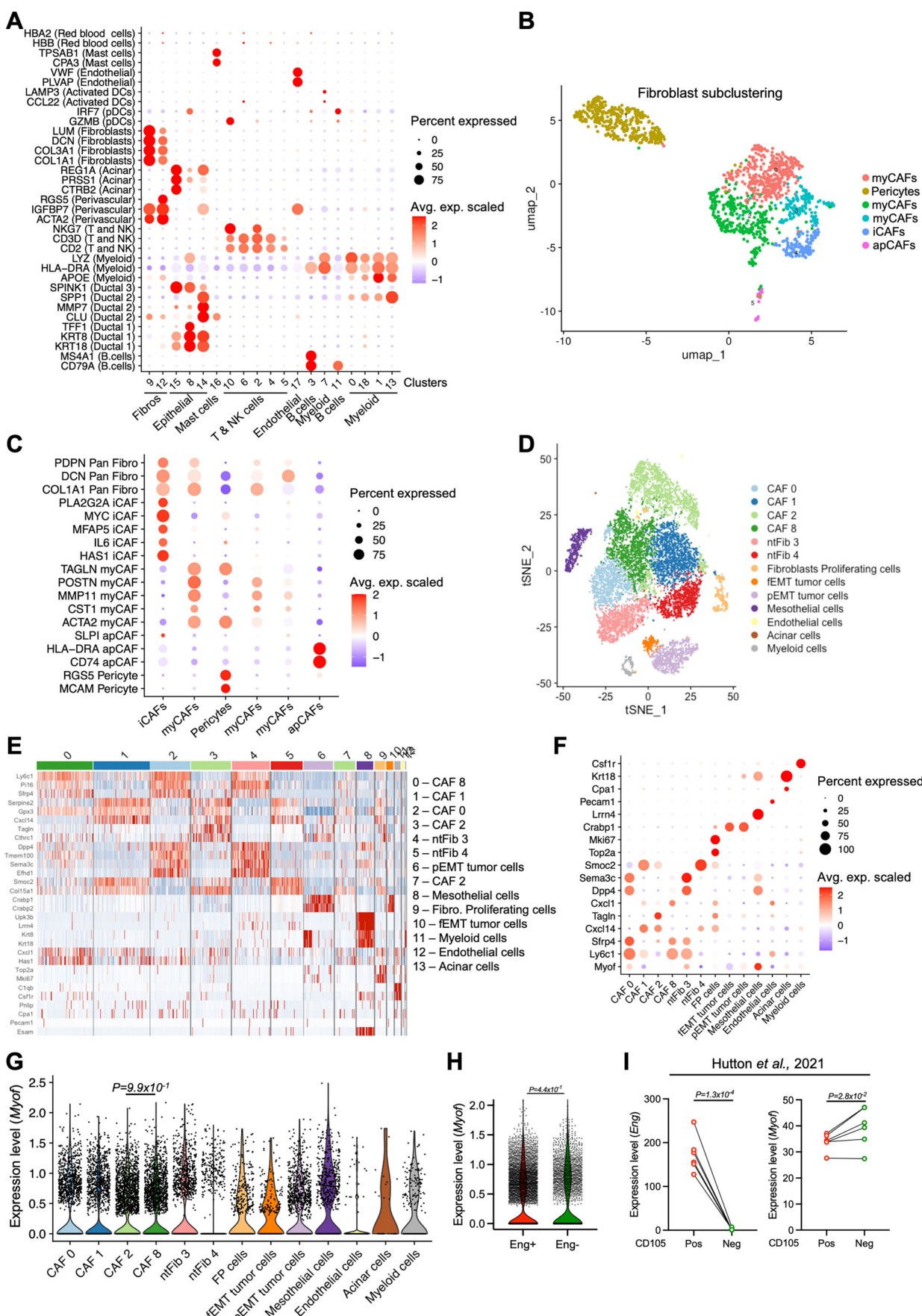

◀   **Figure EV2.   Myoferlin expression in pancreatic cancer at single cell resolution.**

(**A**) Marker genes for broad cell type labeling from Fig. 2A. Dot plot. (**B**) UMAP plot for fibroblasts subclustering from human PAAD scRNAseq data. (**C**) Marker genes for fibroblasts subclustering from (**B**). Dot plot. (**D**) t-SNE plot and cluster labeling for murine PAAD scRNAseq data. iCAF lineage: ntFib3 = normal Fibroblasts; CAF0 = early CAFs; CAF8 = IL1 CAFs. myCAF lineage: ntFib4 = normal Fibroblasts; CAF1 = early CAFs; CAF2 = TGFß CAFs. (**E**) Marker genes for clusters from (**D**). Heatmap. (**F**) *Myof* expression and marker genes for clusters from (**D**). Dot plot. (**G**) *Myof* expression in clusters from (**D**). Violin plot, pairwise comparisons using Wilcoxon Rank test (Bonferroni correction). (**H**) Normalized *Myof* expression levels in fibroblasts clustered according to CD105 (Eng) expression (0.05 threshold for Eng + ). Violin plot, pairwise comparisons using Wilcoxon Rank test (Bonferroni correction). (**I**) Normalized *Eng* and *Myof* expression levels in CyTOF-isolated murine CD105[pos] and CD105[neg] CAFs. Paired *T* test. Source data are available online for this figure.

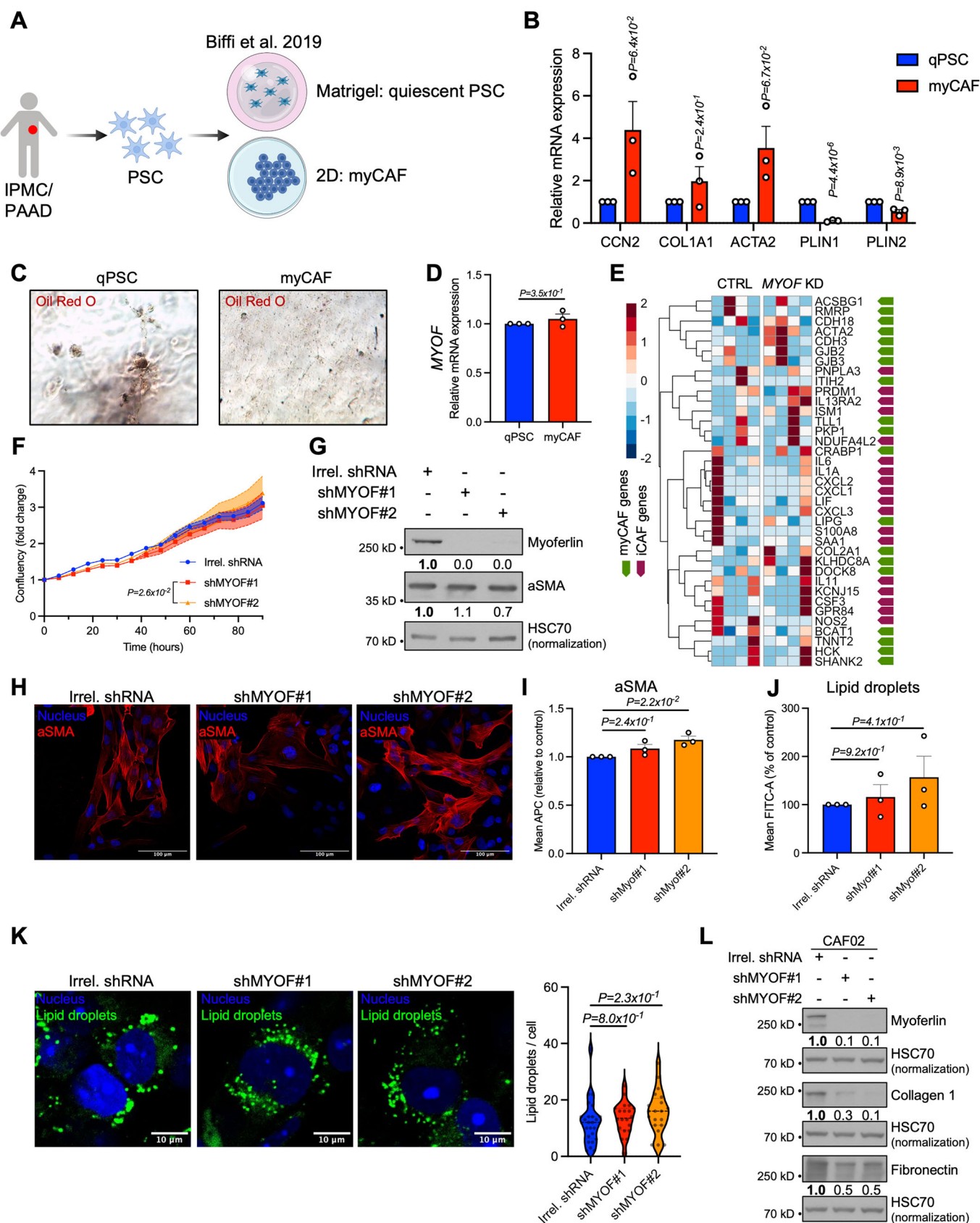

**Figure EV3.  Myoferlin knockdown does not induce CAF quiescence.**

(A) Culture model for PSC-derived myCAFs. (B) RT-qPCR analysis of *CCN2*, *COL1A1*, *ACTA2*, *PLIN1* and *PLIN2* mRNA levels in qPSCs ($n = 3$) and myCAFs ($n = 3$). Mean ± SEM, unpaired *T* test (*P* values relative to qPSC). (C) Oil Red O staining of lipid droplets in qPSCs and myCAFs. (D) RT-qPCR analysis of *MYOF* mRNA levels in qPSCs ($n = 3$) and myCAFs ($n = 3$). Mean ± SEM, unpaired *T* test. (E) Heatmap and unsupervised clustering of myCAF and iCAF signature genes in CTRL myCAFs ($n = 4$) and *MYOF*^KD myCAFs (siMYOF#1; $n = 4$). Gene expression values are z-score normalized. (F) Confluency-based cell proliferation analysis of CTRL myCAFs (Irrel. shRNA; $n = 3$) and *MYOF*^KD myCAFs (shMYOF#1 and shMYOF#2; $n = 3$ each). Mean ± SEM (dashed lines), two-way ANOVA (Tukey's test). (G) Western blot analysis of CTRL myCAFs (Irrel. shRNA) and *MYOF*^KD myCAFs (shMYOF#1 and shMYOF#2). HSC70 was used as loading control. (H) Immunofluorescence microscopy of CTRL myCAFs (Irrel. shRNA) and *MYOF*^KD myCAFs (shMYOF#1 and shMYOF#2). Representative pictures are shown. Nuclei = blue, aSMA = red, scale bar = 100 μm. (I) Flow cytometry analysis of alpha-SMA (APC-conjugated antibody) in CTRL myCAFs (Irrel. shRNA; $n = 3$) and *MYOF*^KD myCAFs (shMYOF#1 and shMYOF#2; $n = 3$ each). Mean APC signal was expressed as % of control (Irrel. shRNA). Mean ± SEM, one-way ANOVA (Dunnett's test). (J) Flow cytometry analysis of lipid droplets (FITC-Bodipy staining) in CTRL myCAFs (Irrel. shRNA; $n = 3$) and *MYOF*^KD myCAFs (shMYOF#1 and shMYOF#2; $n = 3$ each). Mean FITC-A signal was expressed as % of control (Irrel. shRNA). Mean ± SEM, one-way ANOVA (Dunnett's test). (K) Confocal microscopy analysis of lipid droplets (FITC-Bodipy staining) in CTRL myCAFs (Irrel. shRNA; $n = 24$ cells) and *MYOF*^KD myCAFs (shMYOF#1 and shMYOF#2; $n = 20$ each). Scale bar = 10 μm. Violin plot, one-way ANOVA (Sidak's test). (L) Western blot analysis (CAF02) of total-cell lysates from CTRL myCAFs (Irrel. shRNA; $n = 3$) and *MYOF*^KD myCAFs (shMYOF#1 and shMYOF#2; $n = 3$ each). One representative western blot of three independent experiments is shown, HSC70 was used as loading control. Source data are available online for this figure.

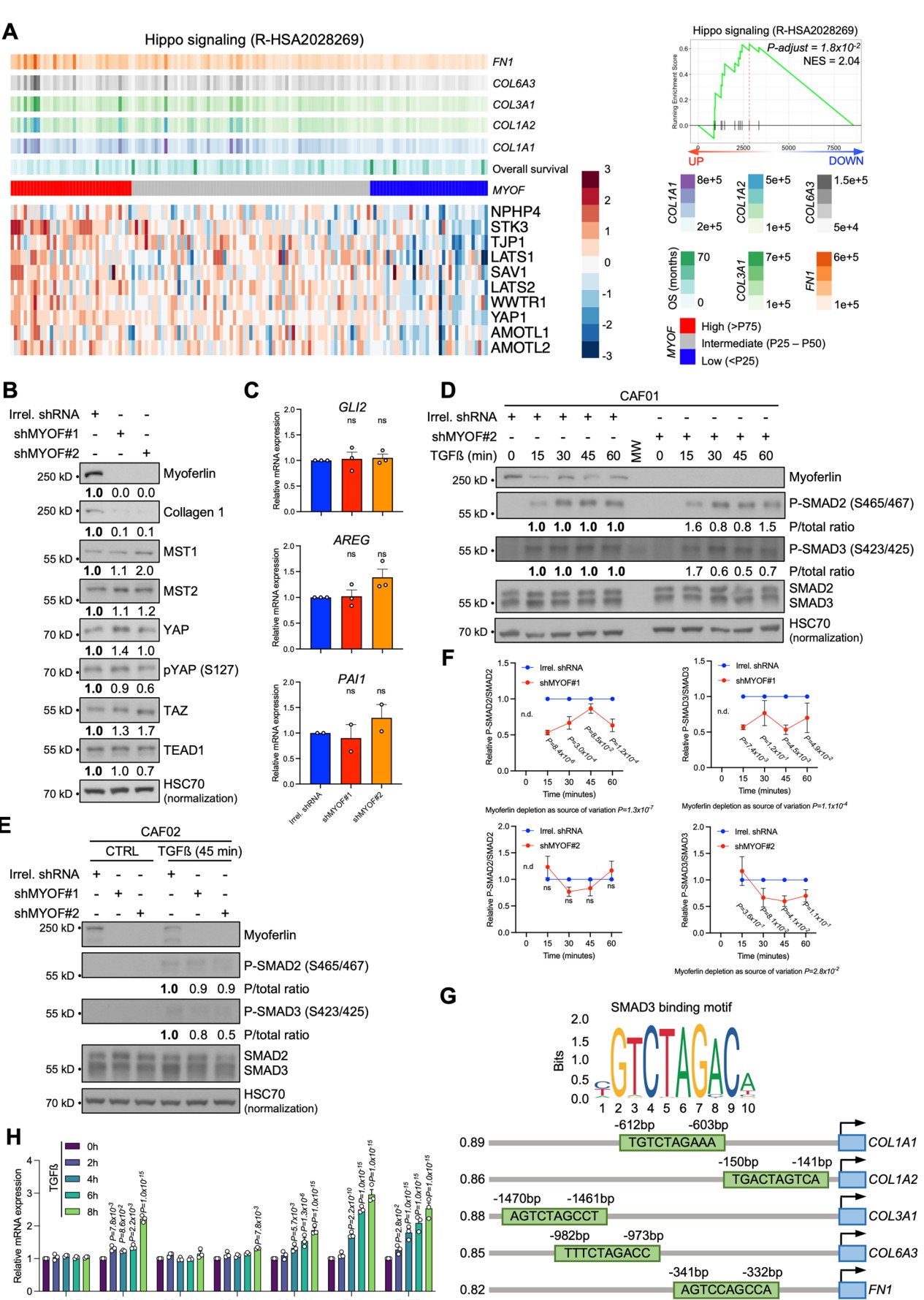

◀  **Figure EV4.  TGFß signaling but not Hippo signaling is altered upon myoferlin knockdown.**

(A) GSEA analysis and heatmap of TCGA PAAD cohort patients ($n = 146$) and Hippo-related genes ($P < 0.05$). Patients are segregated based on *MYOF* expression. *P* value assessed via Benjamini–Hochberg procedure. (B) Western blot analysis of total-cell lysates (CAF01) from CTRL myCAFs (Irrel. shRNA; $n = 1$) and *MYOF*<sup>KD</sup> myCAFs (shMYOF#1 and shMYOF#2; $n = 1$ each). HSC70 was used as loading control. (C) RT-qPCR analysis (CAF01) of *GLI2*, *AREG*, and *PAI1* mRNA levels in CTRL myCAFs (Irrel. shRNA; $n \geq 2$) and *MYOF*<sup>KD</sup> myCAFs (shMYOF#1 and shMYOF#2; $n \geq 2$ each). Mean ± SEM, one-way ANOVA (Tukey's test). (D) Western blot analysis of total-cell lysates (CAF01) from CTRL myCAFs (Irrel. shRNA; $n = 3$) and *MYOF*<sup>KD</sup> myCAFs (shMYOF#2, $n = 3$) stimulated with human recombinant TGFß1 (5 ng/mL) for indicated timepoints. One representative western blot of three independent experiments is shown, HSC70 was used as loading control. (E) Western blot analysis of total-cell lysates (CAF02) from CTRL myCAFs (Irrel. shRNA; $n = 3$) and *MYOF*<sup>KD</sup> myCAFs (shMYOF#1 and shMYOF#2, $n = 3$ each) stimulated with human recombinant TGFß1 (5 ng/mL) for 45 min. One representative western blot of three independent experiments is shown, HSC70 was used as loading control. (F) Quantification of Fig. 4D and (D). Mean ± SEM, two-way ANOVA (full model including time effect, shRNA effect and interaction effect). Individual pair comparisons were performed using the Fisher's LSD test, *P* value relative to control group (Irrel. shRNA). (G) SMAD3 consensus binding motif and predicted binding sites in promoter regions of *COL1A1*, *COL1A2*, *COL3A1*, *COL6A3* and *FN1* genes. Relative binding site scores were computed with Jaspar. (H) RT-qPCR analysis (CAF01) of *MYOF*, *COL1A1*, *COL1A2*, *COL3A1*, *COL6A3*, *FN1* and *TGFBI* mRNA levels in TGFß1 (5 ng/mL) stimulated CTRL myCAFs (Irrel. shRNA). Mean ± SEM (3 technical replicates), two-way ANOVA (Tukey's test; *P* values relative to 0 h condition for each gene). Source data are available online for this figure.

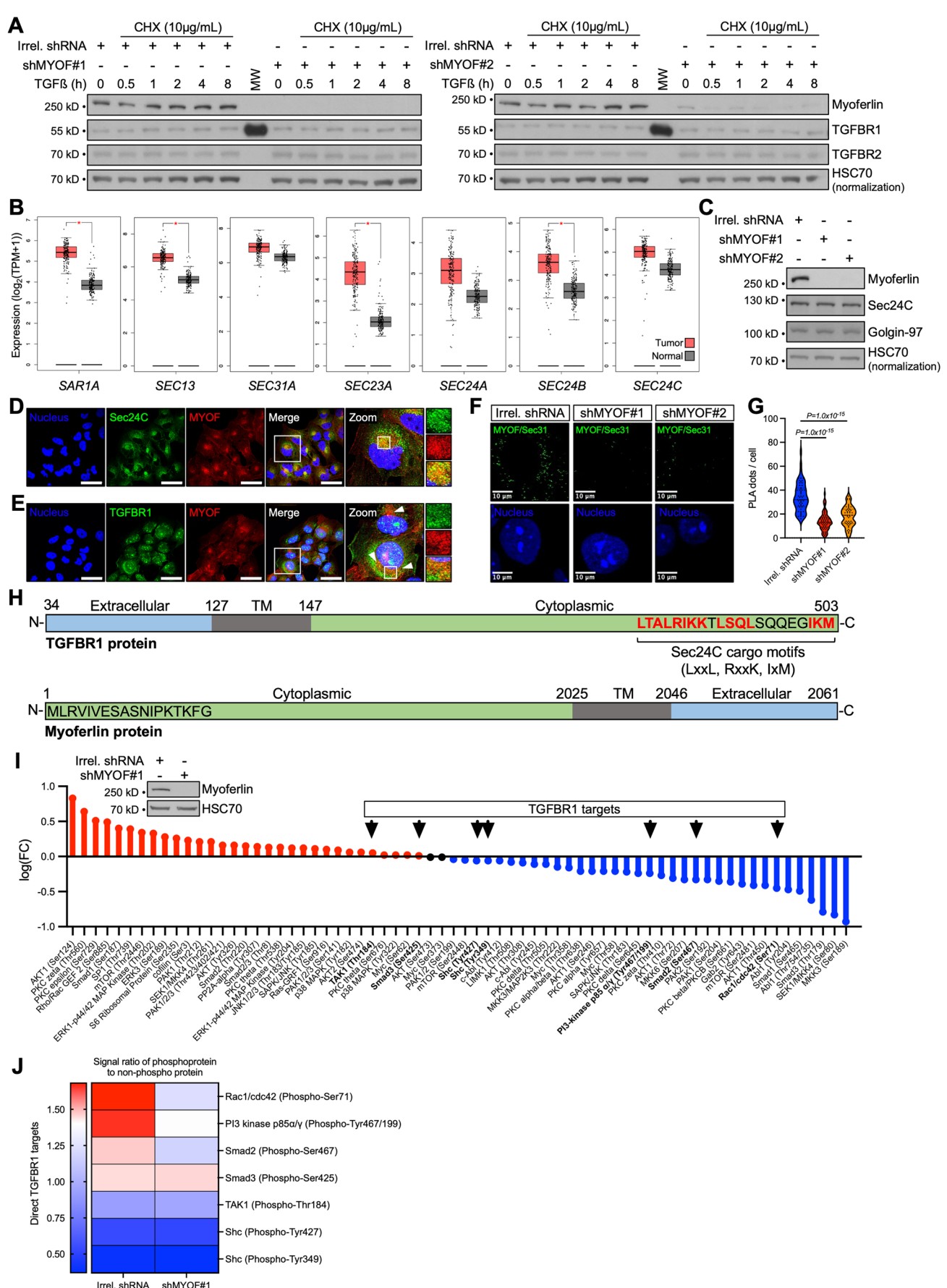

◀  **Figure EV5.  Myoferlin knockdown impairs TGFBR1 trafficking and activity.**

(**A**) Cycloheximide (CHX) chase upon TGFß1 stimulation and western blot of total-cell lysates (CAF01) from CTRL myCAFs (Irrel. shRNA) and *MYOF*^KD^ myCAFs (shMYOF#1 and shMYOF#2). HSC70 was used as loading control. (**B**) COPII vesicle trafficking gene expression (*SAR1A, SEC13, SEC31A, SEC23A, SEC24A, SEC24B, SEC24C*) in human healthy pancreas (grey) and PAAD tissue (red). Boxplot (P25-1.5*IQR; P25; Median; P75; P75 + 1.5*IQR). *$P < 0.05$. (**C**) Western blot analysis of total-cell lysates (CAF01) from CTRL myCAFs (Irrel. shRNA, $n = 1$) and *MYOF*^KD^ myCAFs (shMYOF#1 and shMYOF#2, $n = 1$ each). HSC70 was used as loading control. (**D**) Immunofluorescence microscopy of myCAFs (CAF02). Representative pictures are shown. Nuclei = blue, COPII vesicles (Sec24C) = green, myoferlin (MYOF) = red, scale bar = 50 μm. (**E**) Immunofluorescence microscopy of myCAFs (CAF02). Representative pictures are shown. Nuclei = blue, TGFBR1 = green, myoferlin (MYOF) = red, scale bar = 50 μm. (**F**) Proximity ligation assay (PLA) (CAF01) between myoferlin (MYOF) and COPII vesicles (Sec31) in CTRL myCAFs and *MYOF*^KD^ myCAFs (shMYOF#1 and shMYOF#2). PLA control (CTRL^neg^) without primary antibodies is included. Representative pictures are shown. Nuclei = blue, PLA-dots = green, scale bar = 10 μm. (**G**) Quantification of PLA-dots shown in Fig. 5E. Violin plot, one-way ANOVA (Tukey's test). (**H**) Visualization of TGFBR1 and myoferlin amino acid sequence with Sec24C cargo motifs highlighted in the cytoplasmic region. (**I, J**) Differentially phosphorylated proteins (CAF01) upon TGFß1-stimulation (8 h) between CTRL myCAFs (Irrel.shRNA) and *MYOF*^KD^ myCAFs (shMYOF#1). Proteins in bold mark TGFBR1 kinase targets. Source data are available online for this figure.

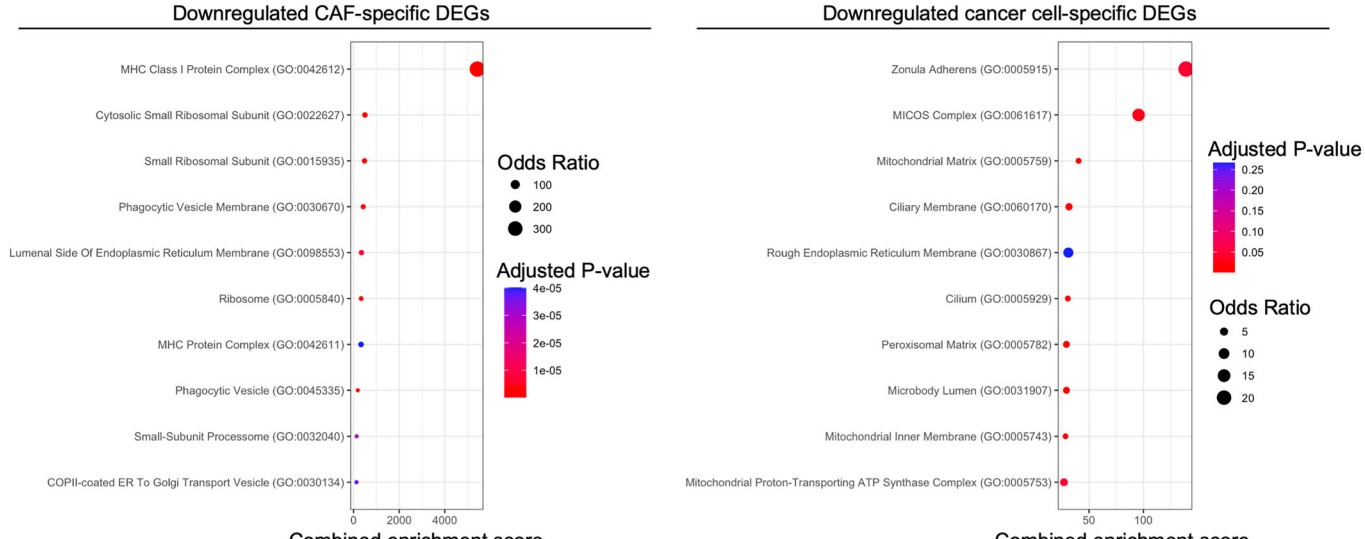

**Figure EV6.** **Downregulated genesets upon myoferlin knockdown in CAFs and cancer cells.**

ORA results of CAF-specific and cancer cell-specific downregulated DEGs. Genesets extracted gene ontology (GO – cellular compartments). Source data are available online for this figure.

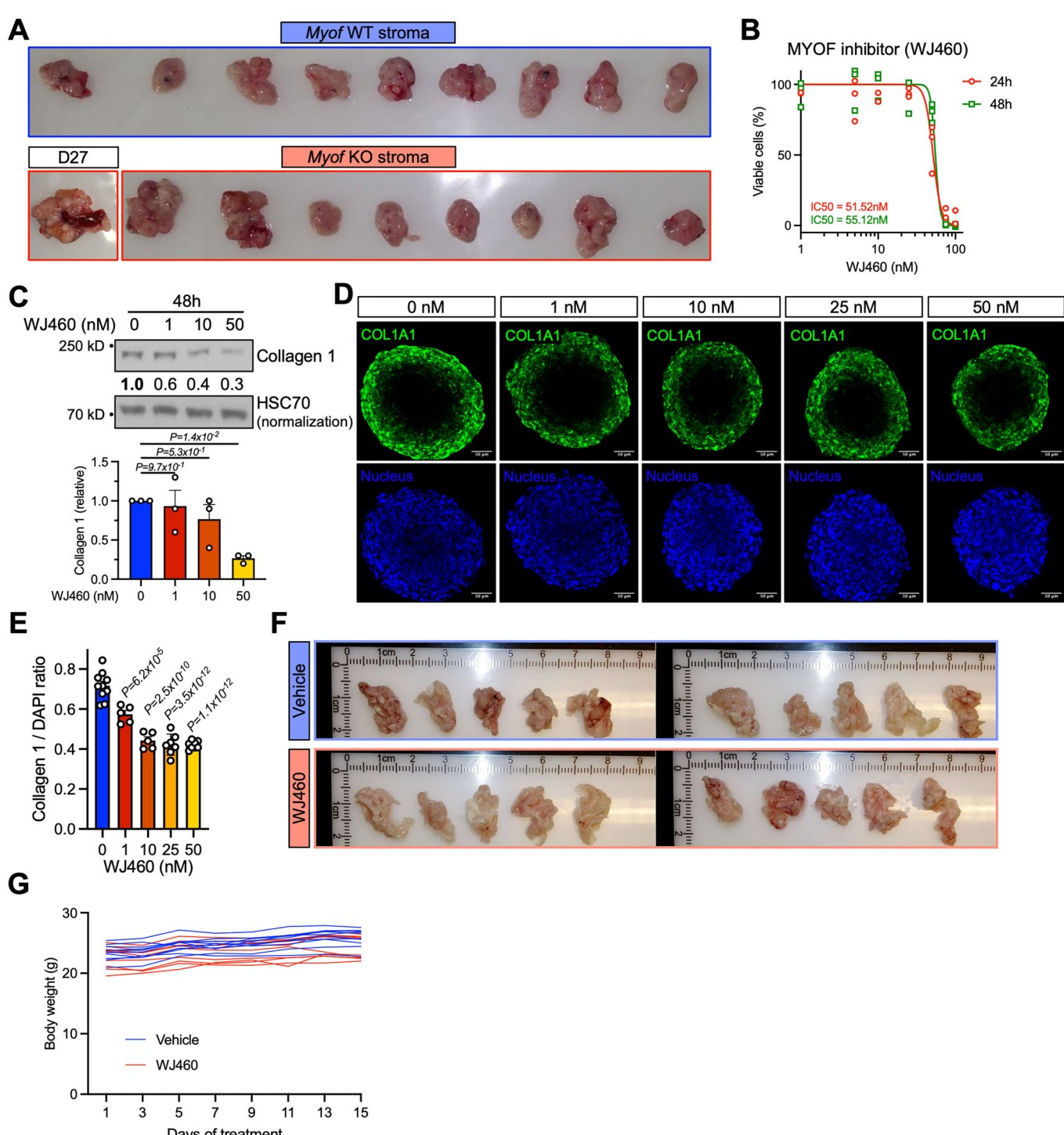

**Figure EV7.   Pharmacological targeting of myoferlin impairs ECM production.**

(A) Resected tumors (D28) of orthotopic KPC allografts into *Myof*^WT (*n* = 9) or *Myof*^KO (*n* = 9) mice. (B) IC50 analysis of myCAFs treated for 24 h or 48 h with increasing concentrations of WJ460 (*n* = 3). (C) Western blot analysis and quantification of total-cell lysates (CAF01) from 48 h WJ460-treated myCAFs (*n* = 3). One representative western blot of three independent experiments is shown, HSC70 was used as loading control. Mean ± SEM, one-way ANOVA (Tukey's test), *P* value relative to control group (0 nM). (D) Immunofluorescence microscopy of homotopic spheroids (*n* ≥ 5) generated from myCAFs and treated with increasing concentrations of WJ460. Representative pictures are shown. Nuclei (DAPI) = blue, collagen 1 = green, scale bar = 50 µm. (E) Mean fluorescence quantification of spheroids (*n* ≥ 5) shown in (D), collagen 1 intensity was normalized to DAPI. Mean ± SEM, one-way ANOVA (Tukey's test, *P* values relative to 0 nM). (F) Resected pancreata of vehicle (DMSO, *n* = 10) and WJ460-treated (*n* = 10) mice orthotopically injected with KPC cells (D21). (G) Body weight (g) evolution of vehicle (*n* = 10) and WJ460-treated (*n* = 10) mice orthotopically injected with KPC cells. Source data are available online for this figure.

