## [Peer Review File · The EMBO Journal]

Targeting myoferlin in ER/Golgi vesicle trafficking reprograms pancreatic cancer-associated fibroblasts

Raphaël Peiffer, Emilie Laverdeur, Anthoula Gaigneaux, Yasmine BOUMAHD, Charlotte Gullo, Gilles Rademaker, Rebekah CRAKE, Arnaud Lavergne, Naima Maloujahmoum, Ferman Agirman, Michael Herfs, Atsushi Masamune, Elisabeth Letellier, Akeila Bellahcene, and Olivier Peulen

Corresponding author(s): Olivier Peulen (olivier.peulen@uliege.be) , Raphaël Peiffer (peiffer.r@wehi.edu.au)

Review Timeline:

Submission Date:	22nd Aug 24
Editorial Decision:	27th Sep 24
Appeal Received:	30th Sep 24
Editorial Decision:	10th Oct 24
Appeal Received:	4th Jul 25
Editorial Decision:	26th Aug 25
Revision Received:	1st Sep 25
Accepted:	5th Sep 25

Editor: Ieva Gailite

Transaction Report:

Dear Dr. Peulen,

Thank you for submitting your manuscript for consideration by The EMBO Journal. I sincerely apologise for the protracted assessment process due to delays in report submission. We have now received three reviewer reports on your manuscript, which are included below for your information. Based on these comments, we unfortunately had to conclude that the study is not a sufficiently strong candidate for publication in The EMBO Journal.

As you can see, while the reviewers find the proposed role of myoferlin in regulation of extracellular matrix production via TGFBR1 trafficking from ER to Golgi interesting, they also find that the in vivo evidence for its role in cancer-associated fibroblasts in pancreatic cancer context remains currently rather weak. Reviewers #1 and #3 also find that further experiments to strengthen the evidence for the interaction of MYOF with Sec24, its role in ER-to-Golgi trafficking and its effect on TGFbeta signalling would be needed. Since addressing these concerns, and in particular the in vivo aspect, would require extensive additional experimental work with an unclear outcome, I am afraid that we cannot offer to explicitly invite a revision of your manuscript at The EMBO Journal.

Nevertheless, if you find that you can address all or most of the reviewers' concerns, and especially those on the in vivo aspect, with conclusive data that support the proposed mechanism, I would be happy to reconsider the revised manuscript. In such a case, I would send it back to the same reviewers, if possible, but would allow them to make new comments on the data, which might then have to be further addressed if the reviewers are more positive in this round of assessment.

Alternatively, I have discussed your manuscript and referee comments with my colleague Martina Rembold at our sister journal EMBO Reports. I am glad to say that she is potentially interested in your findings but also notes the limitations regarding the in vivo data. Since this concern as such does not preclude publication in EMBO Reports, she would invite you to provide a point-by-point response for further evaluation in their editorial team.

You do not need to revise the manuscript prior to transfer. You can also contact Martina at m.rembold@emboreports.org in advance to find out more about the transfer and the required revisions. If you find the transfer option of interest, you can use the link below to transfer the manuscript:

Link Not Available

Thank you in any case for the opportunity to consider this manuscript. I regret that I could not offer better news this time, but I nevertheless hope that you will find the transfer offer of interest.

With kind regards,

Ieva Gailite

Referee #1:

In this study, Peiffer wt al. set out to discover a potential function of myoferlin in stromal cells in pancreatic cancer, specifically in CAFs of PAADs.

Starting with patient samples, they showed nicely that Myoferlin is expressed in stromal CAFs and its expression is correlated with disease outcome. They then applied multiple in-vitro assays to show that myoferlin is required for ECM production, and to dissect the mechanism by which it operates- through TGFB1 signaling. They show that Myoferlin affects the migration of the TGFB1 receptor to the ER, thus reducing TGFB signaling in the CAFs and reducing the secretion of multiple structural ECM proteins. They then test the effect of Inhibition of myoferlin in-vivo, and claim that tumor burden as well as collagen 1 levels are reduced.

Overall this is an interesting study, revealing a very clear and streamlined mechanism for the action of myoferlin in CAFs, and its

affect on CAF activity. The authors used multiple in-vitro methods to assert their findings in a methodical and comprehensive manner, demonstrating the possible implications of myoferlin inhibition without overstating its importance. Nevertheless, there are several issues that should be addressed. Mostly, the in vivo data is weak and its interpretation, even if modest, is still overstated. The experimental evidence in this part must be strengthened or this whole part should be removed, in which case the potential clinical implications will be significantly reduced.

Major comments:

1. In the abstract the authors mention MYOF inhibition as a reprogramming factor for CAFs. They performed RNA-seq on MYOF WT/KD cells but don't show if any other functions of CAF are changed except ones related directly to TGF β and ECM. Does it "normalize" other CAF functions, reverting them back to a normal phenotype? Since Myoferlin is expressed by all CAF subsets, this is important to address. Following up on this. In Ext. Fig 3e the authors show no change in ECM genes, but then in Fig 3C show, based on RNA-seq, significant changes in ECM genes. How do they explain this discrepancy?
2. The kd of MYOF has very limited effect, and many of the panels in Fig 4 are therefore not convincing. For example, in Fig 4f-g SMAD2 does not seem to be affected, only SMAD3 and the overall effects are small. Perhaps Crispr would work better? Also, in Fig 4B the expression of pSMAD2 is so weak even without the kd that it is very hard to tell if the decrease in the kd cells is meaningful. And in Fig 4J, the authors state that "Strikingly, even under exogenous TGF β , MYOFKD myCAF's still showed lower SMAD3 target gene expression than CTRL myCAF's" when in fact they show 3 genes. They should name those genes rather than referring to a general SMAD3 signature.
3. The mechanism presented in Figure 5 is very interesting, but requires several clarifications and modifications. Figure 5C - how many times was the exp repeated? Were all 50 cells quantified from one technical repeat or multiple wells/experiments? Also, the BFA result is very nice but it does not "validate" the results as claimed in row 368, it supports the conclusion. Fig 5D - This is very nice but there is no control - what would the colocalization with another COP2 protein look like? Does myoferlin staining change with BFA? Sec24 staining in Fig 5H is very weak. Does it also work in Co-IP? Ext Fig 5G shows 5 genes that were less phosphorylated, but it also shows 2 that are more. and 2 of the less are borderline in terms of fold change. Given many potential targets, is this considered enriched between the full groups?
4. How is Fig 6f-h different from the analysis in Fig 3h?
5. The in vivo data presented in Fig 7 is weak and not convincing. In Fig 7b-c, the biggest difference is between the control and with CAFs condition, the knockdown has a very modest effect. Fig 7d-e is not significant and this reviewer thinks that presenting it is misleading, even if the authors do not state that this is significant. And in Fig 7h-j there is indeed effect on Collagen, but since there is no effect on tumor burden, and the siRNA does not affect Collagen, how can these two be tied? Figure 7 should be substantially modified by much more rigorous experimentation, or removed.

Minor comments:

Fig 1 - How were the cut-offs to the myoferlin high/low groups decided?

Supp. Figures 1h-l This analysis is unclear, how many patients were used in this analysis? Is the n in the correlation figure (i) the number of areas taken?

I think a more accurate measurement would be per patient/sample then averaging the score for the intersection of myoferlin+mason staining and giving one value for each sample. As it is presented, I would avoid drawing conclusions from the presented analysis.

Figures 4f-g, 5C - as mentioned in the major comments, it is unclear from the figure legend if the experiment was repeated as biological/technical repeats or each condition was one biological repeat in which the individual values presented are per cell. If the latter is true you have to change the analysis to average cells per each biological/technical repeat- showing values of individual cells as the number of reps in an experiment is conflating the p-value.

Figure 7e, j - It is mentioned in the figure legends that the collagen abundance was calculated for CK positive regions, why not for the entire tumor volume?

Referee #2:

Peiffer et al investigate vesicle trafficking in cancer associated fibroblasts (CAFs) in pancreatic cancer. They uncover that myoferlin (MYOF) activity in CAFs is associated with pancreatic cancer progression. The authors exploit omics data, in vitro studies and in vivo experiments to present data that supports loss of MYOF in CAFs alters TGF β receptor 1 trafficking. TGF β signaling in the tumor microenvironment is a key pathway that facilitates tumor progression; however, therapeutic strategies directly targeting the pathway have universally failed in the clinic. Thus, the authors present an intriguing strategy to interfere with TGF β signaling in CAFs, which could be beneficial therapeutically. There are multiple strengths associated with the manuscript, including the clear text and solid presentation of the data. Figures 1-4 provide a clear rationale for the focus on MYOF through the use of expression data and loss of function studies. Figure 5 provides a rationale to investigate the effect of loss of MYOF on TGF β signaling and TGF β R1 trafficking. Data presented in Figure 6 is solid. There are however several challenges as detailed below.

Comments:

1. Figure 7 does not present strong evidence of the loss of MYOF in CAFs is impactful in vivo. This is somewhat surprising given the overall importance of TGF β signaling in the microenvironment of pancreatic cancer.
a-c) The effect of admixing CAFs with Panc1 cells is mild in terms of pancreas weight. Would a larger number of Panc1 cells

provide a greater delta? The data presented in panel b indicate that loss of MYOF has no effect on pancreas weight. The incidence of tumor presence at day 21 (panel c) also shows modest effect of loss of MYOF in CAFs.

d-e) The data displayed do not show any difference between the groups?

f-j) the use of a pharmacologic strategy to inhibit MYOF is exciting. However the data shown with WJ460 is not impressive. There is no effect on pancreas weight by the drug and while there could be some changes in histology (panel h) the change is modest and there is no change in collagen deposition (panel i, j).

The effects shown in extended data fig 7 are also modest at best.

I encourage repetition of the experiments with WJ460. If a higher dose is feasible that would be useful. A longer time of exposure (more than 2 weeks) might also enable demonstration of significant effects. Further evaluation of TGF β canonical signaling in CAFs and a trichrome stain to evaluate collagen deposition could provide more robust validation of MYOF as a therapeutic candidate. Given the fact that TGF β is a significant driver of immune suppression in tumors, it might also be useful to evaluate the immune landscape of the tumors, this could be done via IHC.

Referee #3:

- general summary and opinion about the principle significance of the study, its questions and findings

The manuscript entitled "Targeting myoferlin in ER/Golgi vesicle trafficking reprograms cancer-associated fibroblasts in PAAD" describes a novel MYOF function in myofibroblast-like CAFs, contributing to desmoplasia. PAAD patients data show that MYOF^{high} stroma is associated with tumor aggressiveness and desmoplasia. The authors describe for the first time MYOF expression in CAFs, and especially show an increase of its expression in the myCAF lineage during tumor progression. Using MYOF-KD cells for transcriptomics analysis, imaging and biochemistry approaches demonstrated that MYOF in myCAF plays a role in TGFBR1 trafficking to the cell surface, TGF β signaling and TGF β -mediated ECM production. Additionally, it impacts myCAF migration and contractility. Finally, using mice models shows that MYOF is important for tumor establishment, and its targeting reduces collagen I production in tumors, making it a potential therapeutic target in association with other therapeutic strategies. The strength of this study is, in addition to have found a new role of MYOF in CAFs, to have used a whole scale approach, from patients data analysis to in vitro and mice in vivo experiments, to thoroughly investigate its mechanism of action. However, not all conclusions are fully supported by the data shown, especially concerning the investigation of MYOF role in trafficking, where MYOF involvement in TGFBR1 ER to Golgi transport should be further clarified. Should the concerns below be addressed to strengthen the manuscript, this study will be a substantial step forward to understand CAFs contribution to PAAD and further studies will be necessary to determine the potential of targeting MYOF for therapy.

- specific major concerns essential to be addressed to support the conclusions

1) Figure 4-a is the only patient data analysis that was not entirely convincing. Despite a clear clustering of high expression of FN and collagens with MYOF^{high}, the clustering of the overall survival, and to a lesser extent of TGF β genes, are not striking. Maybe this is due to the heterogeneity of MYOF^{high} patients that might have high MYOF in either cancer cells or in the stroma. If possible, it would be useful to have this analysis for patients with high MYOF in the stroma only or in comparison to patients with high MYOF in the cancer cells.

2) The pulse TGF β treatment experiment of Figure 4-d/e and extended figure 4-d has several issues leading to conclusions not fully reflecting the data shown. First, the authors claim that the TGF β response is reduced and delayed, but only one time point is shown in the quantification. A quantification of all time points showing a shift of the response peak is essential to assess the delay of the response. Second, the quantification on Figure 4-e does not show a significant decrease of SMAD2 phosphorylation, which does not support the conclusion that both SMAD2 and 3 phosphorylation are decreased. Finally, the extended data figure 4-d with sh #2 does not show any effect of shMYOF on p-SMAD2/3 while the quantification on Figure 4-e shows a decrease of p-SMAD3. Could the authors show a more representative blot corresponding to the quantification?

3) In figure 5-c and n Golgin-97 was used as a Golgi marker. However, this is a trans-Golgi marker. Using an early Golgi marker such as GM130 would be more in accordance with the COP2 vesicles function at the interface of ER and Golgi. Moreover, staining with an ER marker would be useful to determine if in MYOF-KD cells SEC24C and TGFBR1 are retained in the ER. This is essential to support the claim that the ER to Golgi transport of those proteins is affected. Finally, in 5-c the zoom from the shMYOF2 condition is not taken from a representative cell of the field of view as most cells there seem to still have a perinuclear pool of SEC24C. Is this heterogeneity due to cells having different MYOF levels after KD?

4) While the PLA experiments convincingly demonstrate that the proteins interact, the images in figures 5-e and j do not indicate in which cellular compartment they interact. The dots seem to be present all over the cells and not just in the perinuclear region where the proteins are shown to colocalize. Is it possible that they interact in the ER? Having immunofluorescent staining of Golgi and ER markers in cells where the PLA is performed and quantify the dots inside and outside those structures would be helpful to confirm where the proteins interact.

5) Some experiments are missing to fully support the conclusion that TGFBR1 trafficking from ER to Golgi is impaired. At least staining of an ER marker to show that TGFBR1 is retained there in MYOF-KD cells is necessary. In addition or alternatively, is it possible to assess the presence of high-mannose glycan chains on TGFBR1 that would indicate that it is not processed in the Golgi with an EndoH/PGNase digestion assay?

- minor concerns that should be addressed

1) In Figure 3-f a control for cell contamination of the conditioned media is missing. Those blots should show a protein that should be present in whole cell lysates but not in the CM such as GAPDH, with additional loading of a cell lysate, to ensure that the CM is free of signal that might come from cells or cell debris that are not fully removed by centrifugation.

2) In Figure 3-j I could not find in the methods nor in the figure legends how the myCAFs nuclei is stained to be differentiated from the cancer cells.

3) Figure 4-b shows only p-SMAD2 in unstimulated cells, what about p-SMAD3? Additionally, showing p-SMAD2 bands with a higher contrast/higher exposure time would allow to see them more easily.

4) The choice of green and red colors to show immunofluorescence images is not ideal for color blind people. I would advise to use a different combination of colors, such as green and magenta, to ensure that everyone is able to see the different markers on those images.

5) In Figure 5-f and k please define Tm1 and Tm2.

6) The authors use the immunoprecipitation experiment in Figure 5-h to show that MYOF and SEC24C directly interact. However, this is not sufficient to prove a direct interaction as it does not allow to exclude an indirect interaction by being part of the same complex. If the authors really want to prove a direct interaction, they should perform immunoprecipitation of recombinant proteins or at least change the text to not claim a direct interaction from the co-immunoprecipitation. Moreover, the legend on this figure does not indicate if this is a representative image of several replicates or if this has been done only once.

7) The authors conclude from Figures 6-a and b and Extended data figure 6 that MYOF has different main functions in CAFs and cancer cells. However, MYOF-KD CAFs and cancer cells both have a decrease of expression of COP2 and ER related genes, indicating that the MYOF function described here could be shared in tumor cells and CAFs, unlike its mitochondrial functions that are specific to tumor cells. I would suggest to re-write the conclusion here to be clearer and in accordance with the data shown.

8) Regarding the test of the effect of MYOF pharmacological inhibition on already established tumors, the authors say in the discussion that no significant reduction of tumor size was observed. This should be said in the results part when concluding on the data. Is this conclusion made from measuring the pancreas weight in Figure 7-g?

- any additional non-essential suggestions for improving the study (which will be at the author's/editor's discretion)

1) Will the whole data from the transcriptomic profiling of MYOF-KD CAFs and cancer cells be available publicly on a repository or together with the paper? This would be valuable for others to look into specific genes, related to vesicle trafficking or other functions.

2) Some phenotypes observed with the MYOF-KD cells are quite mild (reduction of SMAD2/3 phosphorylation and nuclear localization and loss of SEC24C Golgi localization). Would a KO result in stronger phenotypes?

3) The authors wrote that they could not find good flowcytometry antibodies for human TGFBR1. I would suggest as an alternative to perform an immunofluorescence staining of TGFBR1 in unpermeabilized cells and to use a plasma membrane marker to assess its presence at the cell surface.

4) Co-culture of MYOF-WT and KD CAFs with cancer cells in a model where the cancer cells are able to migrate, or in an invasion assay, would be a great addition to assess how MYOF expressed in CAF contributes to tumor cells migration and invasive abilities.

** As a service to authors, EMBO Press provides authors with the possibility to transfer a manuscript that one journal cannot offer to publish to another EMBO publication or the open access journal Life Science Alliance launched in partnership between EMBO Press, Rockefeller University Press and Cold Spring Harbor Laboratory Press. The full manuscript and if applicable,

reviewers' reports, are automatically sent to the receiving journal to allow for fast handling and a prompt decision on your manuscript. For more details of this service, and to transfer your manuscript please click on Link Not Available. **

Liège, 28th August 2024.

Dear Dr Gailite,

Thank you for considering our manuscript entitled “Targeting myoferlin in ER/Golgi vesicle trafficking reprograms cancer-associated fibroblasts in PAAD” in *The EMBO Journal* and for sending our manuscript out for peer-review.

We have read the reviewer reports thoroughly and find that the suggestions are largely relevant and would upon completion lead to an improved manuscript. However, we do want to underline that the main findings of the manuscript using patient data and *in vitro* CAF models are overall supported by all reviewers. While we recognize that mice studies have clear scientific relevance in a manuscript, we do not believe that the absence of *in vivo* studies diminishes the translational application of the findings (Reviewer #1), as long as relevant patient data complements *in vitro* experiments.

As aforementioned, we do believe that stronger *in vivo* data would indeed lead to a scientifically improved manuscript. During submission, we were aware that our *in vivo* studies present a potential weakness of the study. In fact, we have been generating a *Myo*^{KO} mouse using CRISPR in the past year. As CRISPR knockout of embryos in combination with breeding is time consuming, the mice were not ready before the finalizing and submission of the manuscript.

We are pleased to say that we have been able to generate homozygous *Myo*^{KO} mice in our lab which are currently being bred for amplification. We believe that the use of a CRISPR KO mouse model in combination with an extension of the pharmacological model will satisfy the reviewers demands for more thorough *in vivo* studies.

Below you will find a synthesis of the major reviewer demands including a feasibility assessment from our side. Additionally, we have attached a timeline for the experiments necessary to completely fulfill the reviewer reports.

In light of the feasibility assessment in this letter, and the availability of the *Myo*^{KO} mouse that neither you nor the reviewers could be aware of, we hope that you revise your decision and consider a formal review invitation at *The EMBO Journal*.

Sincerely yours,
Prof Olivier Peulen, corresponding author.

Synthesis and feasibility of major reviewer requests:

Reviewer 1

(1) Reviewer asks to assess if other CAF functions in addition to ECM production have been reversed back to a normal phenotype upon *MYOF* KD. Furthermore, the reviewer points towards a discrepancy between Fig. 3c and Ext. Fig. 3e.

We indeed assessed other CAF functions in the manuscript, including contraction and migration (Fig. 6c-h). Both functions showed a “normalization” upon *MYOF* KD. Regarding the discrepancy pointed out by the reviewer, Fig. 3c and Ext. Fig. 3e refer to different gene sets: Fig. 3c being major ECM components in PAAD, while Ext. Fig. 3e refers to myCAF/iCAF signature genes published by Öhlund et al. This signature includes only one ECM component gene, namely *COL2A1*. Consequently, this is the only apparent discrepancy that we could indeed mention. However, *COL2A1* not being part of Fig. 3c we are unable to draw a conclusion regarding the fate of *COL2A1* upon *MYOF* KD. Furthermore, *COL2A1* is only described as minor ECM component in PAAD, while *COL1A1*, *COL1A2* and *COL3A1* contribute to 90% of collagen mass (Tian et al., 2019, pmid: 31484774).

***COL2A1* expression upon *MYOF* KD will be assessed using RT-qPCR to assess whether gene expression is affected in a similar extend than major ECM components shown in Fig. 3c.**

(2) Reviewer points towards a weak pSMAD2 signal in Fig. 4B and the “small” effect of *MYOF* KD on SMAD3 phosphorylation during exogeneous TGFβ stimulation (Fig. 4d-e). Furthermore, reviewer highlights that we use three SMAD3 target genes as readout for the loss of TGFβ signaling and that we refer to these three genes as “SMAD3 signature”.

The weak pSMAD2 signal in Fig. 4B was also pointed out by reviewer 3 (in addition to the absence of pSMAD3 in this panel). SMAD2/3 phosphorylation is indeed hard to assess via western blot in non-TGFβ-treated conditions, hence why we did not obtain a specific signal for pSMAD3 and only a weak signal for pSMAD2. We propose to clarify the absence of TGFβ stimulation in Fig. 4B by improving the annotation of the panel. A better readout for TGFβ signal transduction in untreated cells is the nuclear translocation of SMAD2/3 via confocal microscopy (see Fig. 4f-g – timing 0 min). Accordingly, we found reduced nuclear translocation of SMAD2/3 upon *MYOF* KD in the absence of exogenous TGFβ.

We would like to pinpoint that the “small” effect of *MYOF* KD on pSMAD during the TGFβ pulse-chase has an amplitude ranging between 40 and 60%. However, we do agree that only assessing the peak timing can be oversimplified, hence we will quantify and plot all time points (see also comment of Reviewer 3).

We used indeed three SMAD3 target genes as assessment for reduced TGF β signal transduction, but nowhere in the manuscript did we refer to these genes as “SMAD3 signature”. However, we agree that naming the genes in lines 310-311 would reduce a potential confusion for readers.

We will improve the quantification and visualization of the TGF β pulse-chase experiment (Fig. 4d-e) and amend the text regarding the SMAD3 target genes.

(3) Reviewer highlights that Fig. 5C needs clarifications regarding technical/biological replicates. Furthermore, reviewer suggests repeating colocalization and CoIP experiments with another COP2 protein. Finally, reviewer points towards less phosphorylated TGFBR1-target proteins in a phospho-antibody array upon *MYOF* KD.

We agree that further clarifications are needed for Fig. 5c. Dots are individual cells quantified across 5 fields of view in the same biological replicate. We will repeat Fig. 5C in additional biological replicates and include another COP2 protein (i.e. Sec31). This additional COP2 protein will be used for CoIP with myoferlin. Using another COP2 protein in addition to Sec24C will help elucidate whether the interaction of myoferlin with the COP2 machinery is specific to Sec24C or more universal than we initially suggested. Finally, we showed in Ext. Fig. 5g that 5 out of 7 TGFBR1 targets are less phosphorylated upon *MYOF* KD. We agree that 2 out of the 5 are borderline reduced. A potential differential abundance of these targets can influence the P/total ratio and therefore affect the quantification presented in Ext. Fig. 5g. Hence, we will reassess the raw data of Ext. Fig. 5g and evaluate whether a differential protein abundance affected the quantification of TGFBR1 targets. Additionally, we will improve the Ext. Fig. 5g legend in order to highlight the fact that among the displayed proteins, only the 7 considered ones are direct targets of TGFBR1.

In summary, we will repeat Fig. 5c and MYOF/COP2 colocalization studies using additional COP2 protein. The same COP2 protein will be used for CoIP and the phospho-antibody array presented in Fig. 5g will be re-evaluated to take into account a potentially differential protein abundance.

(4) Reviewer asks how Fig. 6f-h is different from Fig. 3h.

Fig. 3h is a spheroid suspension culture of *MYOF* KD CAFs to assess ECM production. Fig. 6f-h is a collagen gel-embedded culture of CAFs as single cells to evaluate gel retraction (CAF contractile force). Both experiments independently assess crucial CAF functions during PAAD progression, respectively ECM production in Fig. 3h and contractile force in Fig. 6f-h.

We will amend the manuscript text for more clarification to reduce a potential confusion between both panels.

(5) Reviewer points to the weakness of *in vivo* data presented in Fig. 7. In summary, the knockdown of *MYOF* in transplanted CAFs into NOD SCID mice has a modest effect on tumor biology. However, the reviewer acknowledges the impact of pharmacological targeting on collagen deposition.

We agree that the effect of *MYOF* KD in CAFs seems to have a modest effect on tumor burden and collagen deposition. The poor amplitude of the phenotype is likely due to the difficulty to engraft and maintain CAFs into mice. In fact, human CAFs transplanted into mice are quickly replaced by their murine counterparts, thus diluting the effect of *MYOF* KD. To tackle this issue, we will orthotopically inject *Myof* KO mice with KPC cancer cells and extend the experimental time to 4 weeks, resulting in model with *Myof* WT cancer cells and a *Myof* KO stroma. Tumor burden and tumor desmoplasia will be assessed in this model using the same approach as described in Fig. 7b-e. Furthermore, to monitor tumor growth and tumor burden overtime, we transduced KPC cancer cells with luciferase. Allowing a more thorough interpretation of the effect of a *Myof* KO stroma on tumor establishment and progression. Regarding the pharmacological model, reviewer 1 seemed satisfied, but we will nevertheless repeat the experiment and extend the treatment to 3 weeks (see response to reviewer 3).

We will orthotopically inject luciferase-positive KPC cancer cells into *Myof* KO mice (4 weeks total duration instead of 3) and evaluate tumor establishment, progression and desmoplasia. The pharmacological mouse model will be extended to 3 weeks.

Reviewer 2

(1) Reviewer highlights that the *in vivo* evidence of the impact of *MYOF* KD in CAFs is weak. While reviewer pinpoints that the pharmacological targeting of myoferlin *in vivo* is exciting, reviewer admits that the effect on collagen deposition is modest. Furthermore, reviewer suggests assessing TGF β *in vivo*, complementing the collagen analysis with a trichrome Masson stain and evaluating immune infiltration.

We fully agree with reviewer 2. To improve the *in vivo* aspect of our manuscript, we will amend as highlighted for reviewer 1. In addition, we will assess the impact of *Myof* KO or pharmacological targeting *in vivo* on TGF β signaling using IHC, as suggested by reviewer 2.

We have already performed a trichrome Masson stain on all tumors used in this manuscript. We decided not to include the Masson stains due to the known difficulty of quantification. To fully satisfy the reviewer, we will also perform Masson stain on the new tumor samples and include this analysis in the revised manuscript.

Regarding the immune infiltration into tumors upon loss of tumor desmoplasia we admit that this represents an exciting opportunity. However, the impact of myoferlin

KD in CAFs on the immune landscape in PAAD is a project currently being carried out in our lab. Thus, this point disappointingly falls under conflict of interest, but reviewer can be assured that it is being addressed by our group. However, we can include a CD45 staining in our IHC panel, paving the way for our follow-up studies. **Using a *Myof-KO in vivo* model we will perform IHC of TGF β pathway activators such as pSMAD3 and quantify abundance in α SMA-positive cells (myCAFs). Masson stain will be added for all tumors.**

Reviewer 3

(1) Reviewer highlights that Fig. 4a is the only patient data that is not entirely convincing. Reviewer proposes that the modest clustering of TGF β genes with high/low *MYOF* patients might be diluted by high expression of *MYOF* in cancer cells while the expression might be low in the stroma for the same sample.

The point raised by reviewer 3 is relevant and presents a drawback of bulk RNAseq tumor samples. To address this issue, we propose to reperform the analysis and heatmap on TGF β genes but including only patients with a high stromal score. Thus, the bulk RNAseq data will largely be driven by the stromal compartment, while cancer cells represent only a minor contamination.

We will reperform the clustering including only patients with high stromal content.

(2) Reviewer points out that the conclusions from Fig. 4d-e could be strengthened. All time points should be quantified for the TGF β pulse-chase assay and the conclusions should be reworded as the effect is mainly driven by SMAD3 rather than SMAD2/3. Finally, reviewer suggests using a better representative WB for Ext. Fig. 4d.

Again, we fully agree with the reviewer. As highlighted for reviewer 1, the quantification of all time points for the TGF β pulse-chase will be added and the conclusion will be centered around SMAD3. Finally, regarding the WB for Ext. Fig. 4d, we have performed the experiment 3 times and will choose a more representative image.

We will reperform the quantification of WB in Fig. 4d-e, center the conclusion around SMAD3 and choose better representative images.

(3) Reviewer suggests complementing Fig. 5c using GM130 as Golgi marker. Moreover, showing a retention of COP2 and TGFBR1 on the ER using an ER marker would be of interest.

Reviewer raised an important point regarding the Golgi marker. Using GM130 as complement to Golgin-97 is indeed of interest when assessing COP2 vesicle trafficking. Also, evaluating a potential retention of COP2 and TGFBR1 on the ER represents an interesting approach. Although, while exciting, we do not necessarily expect a retention of COP2 and TGFBR1 on the ER. It is plausible that COP2 vesicles still undergo budding from the ER but fail to fuse to the Golgi. Latter being in accordance with suggested functions of myoferlin in membrane fusion. Nevertheless, we will perform the experiments of reviewer 3, as they will give valuable information about the implication of myoferlin in COP2 vesicle trafficking.

We will perform the colocalization between Sec24C and GM130. Furthermore, a potential retention of Sec24C or TGFBR1 on the ER will be assessed via colocalization with an anti-KDEL antibody.

(4) Reviewer suggests that identifying the subcellular localization of PLA interactions could be of interest.

Indeed, PLA alone does give any information about the subcellular localization of these interactions. As proposed by the reviewer, repeating the PLA but including an immunostaining for the ER and Golgi apparatus will give information regarding the subcellular localization of the interactions.

We will reperform the PLA and include a terminal antibody incubation using fluorescent anti-KDEL or anti-GM130 antibodies to respectively highlight the ER and Golgi apparatus.

(5) Similar to comment (3) reviewer suggests assessing if TGFBR1 is retained on the ER upon *MYOF* KD thanks to an analysis of high-mannose content using differential deglycosylation with PNGase/EndoH.

Indeed, a possible retention of TGFBR1 on the ER will be assessed as described in comment (3). Investigating the presence of high-mannose glycan chains on TGFBR1 is also an interesting approach. While we did not detect any TGFBR1 size modification upon *MYOF* KD, we will follow the suggestion of the reviewer.

We will perform EndoH/PNGase digestion on *MYOF* KD CAFs and assess a potential differential deglycosylation of TGFBR1 via western blot.

Proposed timeline of experiments

Month	Oct				Nov				Dec				Jan		
Week	1	2	3	4	1	2	3	4	1	2	3	4	1	2	3
In vivo															
Myof KO mouse breeding	X	X	X	X	X	X	X								
Myof KO tumor transplantation								X	X	X	X				
Myof KO tumor processing/IHC												X	X	X	
KPC- Luc tumor transplantation			X	X	X	X									
WJ460 mouse treatment				X	X	X									
WJ460 tumor processing/IHC							X	X	X	X					
In vitro															
COL2A1 RT-qPCR		X													
Pulse-chase quantification		X													
IF colocalization studies		X	X	X	X										
Antibody array reevaluation			X												
PLA + colocalization						X	X	X							
Patient clustering								X							
EndoH/PGNase assay									X	X	X	X			
Manuscript preparation															X

Dear Dr. Peulen,

Thank you for contacting me regarding the recent editorial decision on your manuscript. I apologise for the delay in responding to you due to the currently high manuscript submission rate to our office, as new submissions have to be treated with a priority.

I have now gone through your revision plan, and I find it very sensible. I am also glad to see that you have meanwhile developed a Myof knockout mouse model, which has the potential to provide more clarity on the relevance of myoferlin for cancer progression. However, since the outlined scope of revisions is rather broad, and the previous decision was also based on the rather unpredictable outcome of these experiments, I am afraid that I cannot explicitly invite a revised manuscript.

Nevertheless, should the revision work be successful and provide substantial additional support to your proposed model, I would be happy to send your manuscript back to the original reviewers for evaluation. It would have to be submitted as a new submission in our system, but we would then link it back to the original manuscript and treat it as a revised version depending on the provided data.

I appreciate that you contacted us again for further discussion of your work, and I hope that the proposed approach sounds reasonable to you.

With kind regards,

Ieva Gailite

** As a service to authors, EMBO Press provides authors with the possibility to transfer a manuscript that one journal cannot offer to publish to another EMBO publication or the open access journal Life Science Alliance launched in partnership between EMBO Press, Rockefeller University Press and Cold Spring Harbor Laboratory Press. The full manuscript and if applicable, reviewers' reports, are automatically sent to the receiving journal to allow for fast handling and a prompt decision on your manuscript. For more details of this service, and to transfer your manuscript please click on Link Not Available. **

Reviewing report

We thank all 3 reviewers for taking the time to read and review our manuscript "Targeting myoferlin in ER/Golgi vesicle trafficking reprograms cancer-associated fibroblasts in PAAD" for The EMBO Journal.

We appreciate the reviewers' compliments regarding our work and their enthusiasm towards the impact of myoferlin and vesicle trafficking in pancreatic CAFs. We have taken the time to address and discuss all comments that were raised. The revisions have led to a significantly improved manuscript, as we have amended conclusions that were unclear, improved quantifications, generated new *in vitro* results (qPCR, quad-IF microscopy, counterstained-PLA and BODIPY staining) and implemented a completely novel *Myof* CRISPR KO mouse model, that replaced the CAF xenografts in figure 7 while amending the existing WJ460-treated mice results. All modifications and amendments are highlighted in red throughout the manuscript.

We hope that the reviewers are satisfied with this improved version of our manuscript.

Reviewer 1

In this study, Peiffer et al. set out to discover a potential function of myoferlin in stromal cells in pancreatic cancer, specifically in CAFs of PAADs.

Starting with patient samples, they showed nicely that Myoferlin is expressed in stromal CAFs and its expression is correlated with disease outcome. They then applied multiple *in-vitro* assays to show that myoferlin is required for ECM production, and to dissect the mechanism by which it operates-through TGFB1 signaling. They show that Myoferlin affects the migration of the TGFB1 receptor to the ER, thus reducing TGFB signaling in the CAFs and reducing the secretion of multiple structural ECM proteins. They then test the effect of Inhibition of myoferlin *in-vivo*, and claim that tumor burden as well as collagen 1 levels are reduced.

Overall, this is an interesting study, revealing a very clear and streamlined mechanism for the action of myoferlin in CAFs, and its effect on CAF activity. The authors used multiple *in-vitro* methods to assert their findings in a methodical and comprehensive manner, demonstrating the possible implications of myoferlin inhibition without overstating its importance. Nevertheless, there are several issues that should be addressed. Mostly, the *in vivo* data is weak and its interpretation, even if modest, is still overstated. The experimental evidence in this part must be strengthened or this whole part should be removed, in which case the potential clinical implications will be significantly reduced.

We thank reviewer 1 for her/his comments and suggestions. We also thank the reviewer for her/his appreciation regarding our *in vitro* data. The detailed revision made by reviewer 1 have led to a significant improvement of the manuscript. Most importantly, we have included a new myoferlin KO mouse model generated using CRISPR, we hope that the bulk of these new results will satisfy the reviewer. Please find below a detailed answer to all comments.

Major comments

1. In the abstract the authors mention MYOF inhibition as a reprogramming factor for CAFs. They performed RNA-seq on MYOF WT/KD cells but don't show if any other functions of CAF are changes except ones related directly to TGFB and ECM. Does it "normalize" other CAF functions, reverting them back to a

normal phenotype? Since Myoferlin is expressed by all CAF subsets, this is important to address. Following up on this. In Ext. Fig 3e the authors show no change in ECM genes, but then in Fig 3C show, based on RNA-seq, significant changes in ECM genes. How do they explain this discrepancy?

Reviewer raised an interesting question relative to the normalization of CAF function. We indeed assessed other CAF functions in the manuscript, including contraction and migration (Fig. 6c-h). Both functions showed a normalization upon *MYOF* KD.

The point made by reviewer raised our curiosity regarding a normal phenotype of *MYOF*^{KD} CAFs. We therefore assessed lipid droplet amount in *MYOF*^{KD} CAFs via confocal microscopy and BODIPY staining. Lipid droplets are considered a quiescence marker for pancreatic stellate cells and therefore can be linked to a “normal” phenotype of CAFs (Vonlaufen *et al.*, 2010, *Pancreatology*). In agreement with the flow cytometry data that was already in our manuscript (Extended Data Fig. 3j), we did not observe a significant effect of *MYOF*^{KD} on lipid droplet amount and distribution in CAFs (Extended Data Fig. 3k).

Regarding the discrepancy pointed out by the reviewer, Fig. 3c and Ext. Fig. 3e refer to different gene sets: Fig. 3c being major ECM components in PAAD, while Ext. Fig. 3e refers to myCAF/iCAF signature genes published by Öhlund et al. The only discrepancy that we could indeed observe is the ECM gene *COL2A1* in the myCAF signature. *COL2A1* is only described as minor ECM component in PAAD, while *COL1A1*, *COL1A2* and *COL3A1* contribute to 90% of collagen mass (Tian et al., 2019, PNAS). However, as *COL2A1* was not part of Fig. 3c in the previous version of the manuscript, we have now assessed *COL2A1* expression in *MYOF* KD CAFs. In agreement with reduced ECM production upon *MYOF* KD in CAFs we also observed less *COL2A1* expression. Of note, CTs for *COL2A1* were high (close to 30), underpinning the low contribution to total ECM content. These new results have been included in Fig. 3c.

- The kd of MYOF has very limited effect, and many of the panels in Fig 4 are therefore not convincing. for example, in Fig 4f-g SMAD2 does not seem to be affected, only SMAD3 and the overall effects are small. Perhaps Crispr would work better? Also, In Fig 4B the expression of pSMAD2 is so weak even without the kd that it is very hard to tell if the decrease in the kd cells is meaningful. And in Fig 4J, the authors state that "Strikingly, even under exogenous TGF β , MYOFKD myCAFs still showed lower SMAD3 target gene expression than CTRL myCAFs" when in fact they show 3 genes. They should name those genes rather than referring to a general SMAD3 signature.

Reviewer is right, SMAD2/3 phosphorylation is indeed hard to assess via western blot in non-TGF β -treated conditions, hence why we did not obtain a specific signal for pSMAD3 (data not shown) and only a weak signal for pSMAD2. A better readout for TGF β signal transduction in untreated cells is the nuclear translocation of SMAD2/3 via confocal microscopy. Accordingly, we found reduced nuclear

translocation of SMAD2/3 upon MYOF KD in the absence of exogenous TGFβ (Fig. 4f-g).

We would like to pinpoint that the effect of MYOF KD on pSMAD during the TGFβ pulse-chase has an amplitude of 40-60%. However, we do agree that only assessing the peak timing can be oversimplified, hence why we quantified and plotted all time points (see plot below). The conclusion of this data remains unchanged as the effect of *MYOF* KD is primarily driven through loss of pSMAD3 in both shRNAs and the most significant difference between CTRL myCAFs and *MYOF* KD myCAFs across both shRNA remains the peak phosphorylation time point (45min). Of note, 2-way ANOVA considered *MYOF* KD as a significant source of pSMAD2/3 variation ($p < 0.05$). These new quantifications have been added into the manuscript as extended figure 4f.

We used indeed three SMAD3 target genes as assessment for reduced TGFβ signal transduction, instead of referring to these genes as “SMAD3 signature”, and following reviewer’s recommendation, we explicitly named the genes in lines 309-313 to reduce a potential confusion for readers.

Myoferlin depletion as source of variation $p < 0.0001$

Myoferlin depletion as source of variation $p = 0.0001$

Myoferlin depletion as source of variation $p = 0.0281$

- The mechanism presented in Figure 5 is very interesting, but requires several clarifications and modifications. Figure 5C - how many times was the exp

repeated? Were all 50 cells quantified from one technical repeat or multiple wells/experiments? Also, the BFA result is very nice but it does not "validate" the results as claimed in row 368, it supports the conclusion. Fig 5D - This is very nice but there is no control - what would the colocalization with another COP2 protein look like? does myoferlin staining change with BFA? Sec24 staining in Fig5H is very weak. Does it also work in Co-IP? Ext Fig 5G shows 5 genes that were less phosphorylated, but it also shows 2 that are more. and 2 of the less are borderline in terms of fold change. Given many potential targets, Is this considered enriched between the full groups?

We agree that further clarifications are needed for Fig. 5c. We have amended the legend for figure 5C accordingly:

Immunofluorescence microscopy (CAF01) with pixel-based colocalization quantification (tM2 Mander's coefficients after threshold regression) and object-based colocalization quantification (SODA) of COP2 vesicles and the Golgi apparatus in CTRL myCAFs (Irrel. shRNA; n>50), MYOF^{KD} myCAFs (shMYOF#1 and shMYOF#2; n>50 each) and myCAFs treated with Brefeldin A (BFA; n≥50). Representative pictures from 5 distinct observations across 2 independent wells are shown, n represents individual cells across the 5 fields of view.

Row 368 has been corrected as suggested by the reviewer:

BFA treatment mimicked MYOF^{KD} in myCAFs, as colocalization between COP2 vesicles and the Golgi apparatus was impaired (Fig. 5c), thus supporting that MYOF^{KD} disrupts anterograde vesicle trafficking of COP2 vesicles between the ER and Golgi apparatus.

To address reviewer's comment regarding a potential colocalization of myoferlin with other COP2 proteins we have performed a PLA between myoferlin and Sec31. Indeed, we observed PLA dots between myoferlin and Sec31, suggesting that these two proteins are in close proximity. Intrigued by this finding, we assessed a direct interaction between myoferlin and Sec31 via CoIP. However, while Sec24C successfully precipitated (as shown in the manuscript), we were unable to precipitate Sec31. These results support that myoferlin is involved with the COP2 machinery (i.e. Sec24C and Sec31) but that the physical interaction of myoferlin with COP2 vesicles is mediated via Sec24C. These results have been integrated in figure 5 and extended data figure 5, we have updated the manuscript accordingly:

A potential implication of myoferlin in COP2 vesicle function was further supported via proximity ligation assay (PLA) between myoferlin and Sec24C or Sec31. The abundant presence of MYOF/Sec24C and MYOF/Sec31 PLA dots in myCAFs proved a close relationship (<40nm distance) between myoferlin and COP2 vesicles which was lost upon MYOF^{KD} (Fig. 5e, Fig. 5g and Extended Data Fig. 5f-g). Finally, non-crosslink coimmunoprecipitation experiments of myoferlin confirmed that myoferlin and Sec24C are part of a same complex, while a close interaction with Sec31 was imperceptible (Fig. 5h). Altogether, these results strongly suggest a

direct function of myoferlin in COP2 vesicle trafficking likely mediated through Sec24C.

Reviewer asked whether myoferlin staining was affected by BFA treatment. We found no striking difference in myoferlin staining when myCAFy were treated with BFA. This result was not included in the manuscript.

Reviewer commented on the signal intensity of Sec24C in figure 5h, as this is a western blot on CoIP eluates, high signal can sometimes be hard to achieve. However, we do believe that the intensity of the band and the specificity of the molecular weight are sufficient to support that myoferlin interacts with Sec24C, in combination with colocalization and PLA data.

Finally, we showed in Extended Data Fig. 5i that 5 out of 7 TGFBR1 targets are less phosphorylated upon *MYOF* KD. We agree that 2 out of the 5 are borderline reduced. It is important to remember that the phosphoarray has been performed under TGF β stimulation. Potential differential abundance of these targets can influence the P/total ratio and therefore affect the quantification presented in Extended Data Fig. 5i. Accordingly, we presented the results as a heatmap, latter clearly shows that the two borderline reduced phosphorylation sites are very poorly phosphorylated in both conditions, suggesting that TGFBR1 likely not signals through Shc in our myCAF model. Additionally, the two highest phosphorylated sites in CTRL myCAFs are not increased under shMYOF#1. Altogether these results support that the major phosphorylation targets of TGFBR1 in our myCAF model present reduced phosphorylation upon *MYOF*^{KD}. This additional representation of TGFBR1 phosphorylation targets has been included in the manuscript as Extended Data Figure 5j.

4. How is Fig 6f-h different from the analysis is fig 3h?

Fig. 3h is a spheroid culture of MYOF KD CAFs in suspension to assess ECM production. Fig. 6f-h is a collagen gel-embedded culture of CAFs as single cells to evaluate gel retraction (CAF contractile force). Both experiments independently assess crucial CAF functions during PAAD progression, namely ECM production in Fig. 3h and contractile force in Fig. 6f-h.

5. The in vivo data presented in Fig 7 is weak and not convincing. In fig 7b-c, the biggest difference is between the control and with CAFs condition, the knockdown has a very modest effect. Fig 7d-e is not significant and this reviewer thinks that presenting it is misleading, even if the authors do not state that this is significant. And in Fig 7h-j there is indeed effect on Collagen, but since there is no effect on tumor burden, and the siRNA does not affect Collagen, how can these two be tied? Figure 7 should be substantially modified by much more rigorous experimentation, or removed.

We agree with the reviewer, the effect of MYOF KD in CAFs seems to have a modest effect tumor burden and collagen deposition. The poor amplitude of the

phenotype is likely due to the difficulty to engraft CAFs into mice. In fact, human CAFs transplanted into mice can quickly be replaced by murine counterparts, thus diluting the effect of MYOF KD. We have removed data related to figure 7a-e to reduce confusion.

To tackle the issue of engrafting *MYOF^{KD}* CAFs, we collaborated with our institutional CRISPR facility to generate *Myof^{KO}* mice, this project was started only a few months before the initial submission of this manuscript, explaining why the model was not included in the first submission. During the revision process, *Myof^{KO}* mice were amplified and prepared for experiments. *Myof* KO was validated during breeding via PCR and western blot (see below). This validation was not included in the manuscript.

We orthotopically injected *Myof^{KO}* mice with KPC cancer cells, resulting in a tumor model with *Myof^{WT}* cancer cells and a *Myof^{KO}* stroma. The absence of myoferlin in the tumor stroma was validated by IHC.

While depletion of myCAFs in pancreatic cancer mouse models has resulted in increased tumor burden in the past (Rhim et al., 2014, *Cancer Cell*; Özdemir et al., 2014, *Cancer Cell*), myoferlin depletion in the tumor stroma did not affect pancreas weight.

We have assessed tumor desmoplasia in this model via Masson's trichrome staining and immunofluorescence aSMA staining. Quantification of stained area was performed using the machine learning pixel classifier in QPath. *Myof*^{KO} mice presented significantly reduced collagen area than WT mice, while the aSMA-positive area (representative for myCAFs) was not significantly reduced. This novel mouse data supports the previous results from our manuscript that myoferlin depletion in myCAFs impairs ECM production without leading to myCAF death. We hope that the novel *in vivo* data in combination with the *in vitro* and patient data convinces the reviewer that myoferlin is relevant for myCAF biology *in vitro* and *in vivo*. The manuscript has been amended with figures 7a-7e and the results section has been updated:

Owing to the suggested implication of myoferlin in stromal aggressiveness and tumor desmoplasia, we aimed to assess the impact of MYOF^{KD} myCAFs on PAAD tumor biology in vivo. We injected murine pancreatic cancer cells (KPC) orthotopically in Myof^{WT} or Myof^{KO} C57BL/6 mice, resulting in a tumor model with Myof^{WT} cancer cells and Myof^{KO} stromal cells (Fig. 7a, Extended Data Fig. 7a). At sacrifice, the pancreas weight was unaffected by stromal Myof^{KO} (Fig. 7b). While stromal Myof^{WT} tumors were characterized by intense myoferlin abundance in cancer cells and stromal cells, stromal Myof^{KO} tumors lacked stromal myoferlin (Fig. 7c). Importantly, stromal Myof^{KO} tumors presented significantly reduced desmoplasia, while the abundance of myCAFs (aSMA-positive cells) was not affected (Fig. 7c-e), thus

supporting the *in vitro* data that showed a loss of ECM production without induced cell death in *MYOF^{KD}* myCAFs (Fig. 3d, Extended Data Fig. 3f).

Furthermore, for the sake of consistency, we have replaced the immunofluorescent staining of collagen 1 in WJ460-treated tumors with a Masson's trichrome stain, as latter covers a broad range of collagens. As above, quantification of stained area was performed using the machine learning pixel classifier in QPath. The results obtained with WJ460 globally align with the stromal *Myof^{KO}*, as tumor desmoplasia was reduced without affecting myCAF abundance. These new results have been implemented into the manuscript as figures 7h-j:

WJ460-treated tumors were characterized by reduced tumor fibrosis without impairing aSMA-positive myCAF abundance (Fig. 7h-j), thus supporting the observations made in tumors composed of a Myof^{KO} stroma (Fig. 7c-e).

Minor comments

1. Fig 1 - How were the cut-offs to the myoferlin high/low groups decided?

Quartiles (P25 and P75) of *MYOF* expression z-scores were chosen as cut-offs for myoferlin high/low groups.

2. Supp. Figures 1h-I This analysis is unclear, how many patients were used in this analysis? Is the n in the correlation figure (i) the number of areas taken?

Patient numbers and number of subTME regions are highlighted in the figure legend: *h) Myoferlin IHC and Masson trichrome staining (collagens stained in blue) in human PAAD sections. Representative images from 99 subTME regions across 33 patients. Scale bar = 100 μ m. i) IHC analysis, correlation and linear regression between Masson trichrome scores and Myoferlin stromal scores (n=99). Pearson (R) and Spearman (ρ) correlation.*

3. I think a more accurate measurement would be per patient/sample then averaging the score for the intersection of myoferlin+mason staining and giving one value for each sample. As it is presented, I would avoid drawing conclusions from the presented analysis.

The pancreatic tumor stroma presents high heterogeneity within patient sections; hence, the introduction of spatially confined sub-tumor microenvironments (subTMEs) in a recent landmark study (Grünwald et al., 2021, Cell). We believe that assessing fibrosis and myoferlin abundance in subTMEs presents the most robust strategy of taking intra-patient heterogeneity into account. However, we acknowledge the point that reviewer raised and reanalyzed the cohort per patient rather than per subTME. These results should be interpreted with caution, due to patient heterogeneity. Accordingly, the conclusions made in the manuscript explicitly state “subTME”:

We further extended these transcriptomic findings via myoferlin IHC and Masson trichrome staining of tumor sections from an internal PAAD patient cohort. Using 99 subTME regions with variations in collagen content (loose stroma vs dense stroma), we found that stromal myoferlin abundance significantly correlated with collagen deposition, as subTME regions with high stromal myoferlin abundance presented denser fibrotic tissue (Extended Data Fig. 1h-i).

4. Figures 4f-g, 5C - as mentioned in the major comments, it is unclear from the figure legend if the experiment was repeated as biological/technical repeats or each condition was one biological repeat in which the individual values

presented are per cell. If the latter is true you have to change the analysis to average cells per each biological/technical repeat- showing values of individual cells as the number of reps in an experiment is conflating the p-value.

As corrected in the major comments, we have amended the figure legends to reduce confusion and explicitly mentioned that data points shown are per cell:

Fig4 g) *Quantification of Fig. 4f. Nuclear and cytosolic mean fluorescence was quantified for individual cells across ≥ 3 fields of view. Mean \pm SEM, one-way ANOVA (Tukey's test), p-value relative to control group (Irrel. shRNA).*

Fig5 c) *Immunofluorescence microscopy (CAF01) with pixel-based colocalization quantification (tM2 Mander's coefficients after threshold regression) and object-based colocalization quantification (SODA) of COP2 vesicles and the Golgi apparatus in CTRL myCAFs (Irrel. shRNA; $n > 50$), MYOF^{KD} myCAFs (shMYOF#1 and shMYOF#2; $n > 50$ each) and myCAFs treated with Brefeldin A (BFA; $n \geq 50$). Representative pictures from 5 distinct observations across 2 independent wells are shown, n represents individual cells across the 5 fields of view.*

We believe that representing individual cells as data points is the only possible way of taking into account the complex biology of these intracellular processes. Even within a same culture, individual cells behave differently due to differences in cell cycle phases and cell/cell contacts across the cultures. We tried to select regions of interest across all our experiments with similar cell densities and avoided the presence of apoptotic cells to introduce the least amount of variation possible. The variation that remains is likely caused by inter-cellular variations and may be amplified by calculating averages for each field of view. Cell-by-cell analysis is the only option to robustly assess intracellular phenomenon and take into account biological variations.

5. Figure 7e, j - It is mentioned in the figure legends that the collagen abundance was calculated for CK positive regions, why not for the entire tumor volume?

In the revised version of the manuscript Figure 7e was removed. Concerning Figure 7j, collagen abundance has now been assessed via Masson's trichrome and quantified "per mouse" (Fig. 7i).

Reviewer 2

Peiffer et al investigate vesicle trafficking in cancer associated fibroblasts (CAFs) in pancreatic cancer. They uncover that myoferlin (MYOF) activity in CAFs is associated pancreatic cancer progression. The authors exploit omics data, in vitro studies and in vivo experiments to present data that supports loss of MYOF in CAFs alters TGF β receptor 1 trafficking. TGF β signaling in the tumor microenvironment is a key pathway that facilitates tumor progression; however, therapeutic strategies directly targeting the pathway have universally failed in the clinic. Thus, the authors present an intriguing strategy to interfere with TGF β signaling in CAFs, which could be beneficial therapeutically. There are multiple strengths associated with the manuscript, including the clear text and solid presentation of the data. Figures 1-4 provide a clear rationale for the focus on MYOF through the use of expression data and loss of function studies. Figure 5 provides a rationale to investigate the effect of loss of MYOF on TGF β signaling and TGF β R1 trafficking. Data presented in Figure 6 is solid. There are however several challenges as detailed below.

We thank reviewer 2 for her/his compliments regarding the presentation and robustness of our work. Please find below a detailed answer to all comments.

Major comments

1. Figure 7 does not present strong evidence of the loss of MYOF in CAFs is impactful in vivo. This is somewhat surprising given the overall importance of TGF β signaling in the microenvironment of pancreatic cancer.

We fully agree with reviewer 2. We agree that the effect of MYOF KD in CAFs seems to have a modest effect tumor burden and collagen deposition. The poor amplitude of the phenotype is likely due to the difficulty to engraft CAFs into mice. In fact, human CAFs transplanted into mice can quickly be replaced by murine counterparts, thus diluting the effect of MYOF KD. We have removed this data from our manuscript.

To improve the in vivo aspect of our manuscript we collaborated with our institutional CRISPR facility to generate *Myof*^{KO} mice, this project was started only a few months before the initial submission of this manuscript, explaining why this model was not included in the first submission. During the revision process, *Myof*^{KO} mice were amplified and prepared for experiments. *Myof* KO was validated during breeding via PCR and western blot (see below). This validation was not included in the manuscript.

We orthotopically injected *Myof*^{KO} mice with KPC cancer cells, resulting in model with *Myof*^{WT} cancer cells and a *Myof*^{KO} stroma. The absence of myoferlin in the tumor stroma was validated via IHC.

We have assessed tumor desmoplasia in this model via Masson's trichrome staining and immunofluorescence aSMA staining. Quantification of stained area was performed using the machine learning pixel classifier in QPath. *Myof*^{KO} mice presented significantly reduced collagen area than WT mice, while the aSMA-positive area (representative for myCAFs) not significantly reduced. This novel mouse data supports the previous results from our manuscript that myoferlin depletion in myCAFs impairs ECM production without leading to myCAF death. Using this new model, we could indeed show that the loss of myoferlin in CAFs has a significant impact on tumor architecture and leads to reduced tumor fibrosis.

We hope that the novel *in vivo* data in combination with the *in vitro* and patient data convinces the reviewer that myoferlin is relevant for myCAF biology *in vitro* and *in vivo*. The manuscript has been amended with figures 7a-7e and the results section has been updated:

Owing to the suggested implication of myoferlin in stromal aggressiveness and tumor desmoplasia, we aimed to assess the impact of MYOF^{KD} myCAFs on PAAD tumor biology *in vivo*. We injected murine pancreatic cancer cells (KPC) orthotopically in *Myof*^{WT} or *Myof*^{KO} C57BL/6 mice, resulting in a tumor model with *Myof*^{WT} cancer cells and *Myof*^{KO} stromal cells (Fig. 7a, Extended Data Fig. 7a). At sacrifice, the pancreas weight was unaffected by stromal *Myof*^{KO} (Fig. 7b). While stromal *Myof*^{WT} tumors were characterized by intense myoferlin abundance in cancer cells and stromal cells, stromal *Myof*^{KO} tumors lacked stromal myoferlin (Fig. 7c). Importantly, stromal *Myof*^{KO} tumors presented significantly reduced desmoplasia, while the abundance of myCAFs (aSMA-positive cells) was not affected (Fig. 7c-e), thus supporting the *in vitro* data that showed a loss of ECM production without induced cell death in MYOF^{KD} myCAFs (Fig. 3d, Extended Data Fig. 3f).

- Figure 7 a-c) The effect of admixing CAFs with Panc1 cells is mild in terms of pancreas weight. Would a larger number of Panc1 cells provide a greater delta? The data presented in panel b indicate that loss of MYOF has no effect of pancreas weight. The incidence of tumor presence at day 21 (panel c) also shows modest effect of loss of MYOF in CAFs.

We have removed these results from our manuscript and replaced them with the novel *Myof* KO mouse model that resulted in a much higher success rate of tumor establishment (18/18 mice).

3. Figure 7 d-e) The data displayed do not show any difference between the groups?

As mentioned above, the effect of *MYOF*^{KO} CAFs during xenografts may be drastically reduced due to a low number of successful grafts and the replacement of human CAFs by murine counterparts (that would be myoferlin WT). We have removed these results and replaced them with our *Myof*^{KO} mouse (see comment #1).

4. Figure 7 f-j) the use of a pharmacologic strategy to inhibit MYOF is exciting. However, the data shown with WJ460 is not impressive. There is no effect on pancreas weight by the drug and while there could be some changes in histology (panel h) the change is modest and there is no change in collagen deposition (panel i, j). The effects shown in extended data fig 7 are also modest at best.

We are relying largely on the results from the *Myof* KO mouse to validate the key message of our manuscript, as figures 1 to 6 were also generated using genetic interference in human CAFs. We hope that the novel *in vivo* data convinces the reviewer.

Regarding the pharmacological inhibition of myoferlin using WJ460, we agree that this represents an exciting opportunity. For the development of a powerful inhibitor, many aspects need to be taken into account, ranging from binding energy, to pharmacokinetics to bioavailability. As the development of a new inhibitor extends far beyond the scope of this work, we used the most widely accepted myoferlin inhibitor WJ460. While we indeed showed an impact of WJ460 on tumor fibrosis (see comment #6) we agree that a more potent inhibitor, or a nanobody-based targeting, may be necessary for further clinical assessment. Nevertheless, our results from the stromal *Myof*^{KO} in combination with WJ460 results underpin the therapeutic potential of myoferlin to tackle tumor fibrosis in pancreatic cancer.

For the sake of consistency, we have replaced the immunofluorescent staining of collagen 1 in WJ460-treated tumors with a Masson's trichrome stain, as latter covers a broad range of collagens. As above, quantification of stained area was performed using the machine learning pixel classifier in QPath. The results obtained with WJ460 globally align with the stromal *Myof*^{KO}, as tumor desmoplasia was reduced without affecting myCAF abundance. These new results have been implemented into the manuscript as figures 7h-j.

*WJ460-treated tumors were characterized by reduced tumor fibrosis without impairing α SMA-positive myCAF abundance (Fig. 7h-j), thus supporting the observations made in tumors composed of a *Myof*^{KO} stroma (Fig. 7c-e).*

- I encourage repetition of the experiments with WJ460. If a higher dose is feasible that would be useful. A longer time of exposure (more than 2 weeks) might also enable demonstration of significant effects.

We have repeated the *in vivo* experiment with WJ460 and extended the treatment time. However, KPC cells were highly aggressive, and the extended timing resulted in ethically unacceptable tumors, thus we were unable to use this data. We hope the novel data generated using the *Myof*^{KO} mouse and the new IHC results from WJ460-treated mice (see comment #6) will convince the reviewer.

- Further evaluation of TGFβ canonical signaling in CAFs and a trichrome stain to evaluate collagen deposition could provide more robust validation of MYOF as a therapeutic candidate. Given the fact that TGFβ is a significant driver of immune suppression in tumors, it might also be useful to evaluate the immune landscape of the tumors, this could be done via IHC.

We agree with reviewers' comment and have used a Masson trichrome stain in the new *Myof* KO mouse model to assess tumor desmoplasia (see comment #1). Furthermore, we have replaced the immunofluorescent staining of collagen 1 in WJ460-treated tumors with a Masson's trichrome stain. The results obtained with WJ460 globally align with the stromal *Myof*^{KO}, as tumor desmoplasia was reduced without affecting myCAF abundance. These new results have been implemented into the manuscript as figures 7h-j:

WJ460-treated tumors were characterized by reduced tumor fibrosis without impairing aSMA-positive myCAF abundance (Fig. 7h-j), thus supporting the observations made in tumors composed of a Myof^{KO} stroma (Fig. 7c-e).

Indeed, investigating the effect of myoferlin targeting on TGFβ signaling in an immune context represents an exciting research axis. We have currently a PhD student in the lab that is addressing this exact question. As this works focusses on

the link between myoferlin and ECM production in CAFs, we hope to be able to show new data in the future in an immunology context.

Reviewer 3

The manuscript entitled "Targeting myoferlin in ER/Golgi vesicle trafficking reprograms cancer-associated fibroblasts in PAAD" describes a novel MYOF function in myofibroblast-like CAFs, contributing to desmoplasia. PAAD patient data show that MYOF^{high} stroma is associated with tumor aggressiveness and desmoplasia. The authors describe for the first time MYOF expression in CAFs, and especially show an increase of its expression in the myCAF lineage during tumor progression. Using MYOF-KD cells for transcriptomics analysis, imaging and biochemistry approaches demonstrated that MYOF in myCAF plays a role in TGFBR1 trafficking to the cell surface, TGF β signaling and TGF β -mediated ECM production. Additionally, it impacts myCAF migration and contractility. Finally, using mice models shows that MYOF is important for tumor establishment, and its targeting reduces collagen I production in tumors, making it a potential therapeutic target in association with other therapeutic strategies. The strength of this study is, in addition to have found a new role of MYOF in CAFs, to have used a whole scale approach, from patient data analysis to *in vitro* and mice *in vivo* experiments, to thoroughly investigate its mechanism of action. However, not all conclusions are fully supported by the data shown, especially concerning the investigation of MYOF role in trafficking, where MYOF involvement in TGFBR1 ER to Golgi transport should be further clarified. Should the concerns bellow be addressed to strengthen the manuscript, this study will be a substantial step forward to understand CAFs contribution to PAAD and further studies will be necessary to determine the potential of targeting MYOF for therapy.

We thank reviewer 3 for her/his encouraging words regarding our work. We especially thank the reviewer noticing the whole scale approach of our manuscript, ranging from patients to *in vitro* and *in vivo* experiments. Please find below a detailed answer to all comments.

Major comments

1. Figure 4-a is the only patient data analysis that was not entirely convincing. Despite a clear clustering of high expression of FN and collagens with MYOF^{high}, the clustering of the overall survival, and to a lesser extent of TGF β genes, are not striking. Maybe this is due to the heterogeneity of MYOF^{high} patients that might have high MYOF in either cancer cells or in the stroma. If possible, it would be useful to have this analysis for patients with high MYOF in the stroma only or in comparison to patients with high MYOF in the cancer cells.

We thank for the careful observation of our analysis. The heatmap indeed includes a certain patient heterogeneity with regarding to sample composition that could interfere with the expression of TGF β related genes. While we showed in figure 1d that the stromal content is equal between MYOF^{high} and MYOF^{low} patients, we are unable to distinguish between patients that have high myoferlin expression in their stroma or high myoferlin expression in their cancer cells due to the bulk RNAseq nature of TCGA data. This highlights a drawback of bulk RNAseq tumor samples. Nevertheless, we exploited a study (Grünwald et al., 2021, Cell) that performed microdissection on PAAD samples prior to bulk RNAseq, resulting in "stromal" and "cancer cell" samples. We have downloaded the data and performed a similar analysis on the samples.

Globally we found that myoferlin expression was higher in cancer cells compared to the stroma. This may be due to the absence of myoferlin in immune cells, thus diluting a stromal expression. Furthermore, we found that not all TGF β -related genes were higher expressed in the stroma, underpinning the complexity of cancer cell/stroma crosstalk. In the Grünwald dataset we observed only slight variations of myoferlin expression in stromal samples (right half of heatmap), making a correlation between myoferlin expression and TGF β -related genes challenging. Intriguingly, we observed a higher variability of myoferlin in cancer cells (left half of heatmap). This raises the hypothesis that high/low myoferlin expression in cancer cells may contribute to the fibrotic phenotype of CAFs. However, due to the considerably small size of this cohort, we prefer to interpret these results with caution and did not include these results in the manuscript.

2. The pulse TGF β treatment experiment of Figure 4-d/e and extended figure 4-d has several issues leading to conclusions not fully reflecting the data shown. First, the authors claim that the TGF β response is reduced and delayed, but only one time point is shown in the quantification. A quantification of all time points showing a shift of the response peak is essential to assess the delay of the response. Second, the quantification on Figure 4-e does not show a significant decrease of SMAD2 phosphorylation, which does not support the conclusion that both SMAD2 and 3 phosphorylation are decreased. Finally, the extended data figure 4-d with sh #2 does not show any effect of shMYOF on p-SMAD2/3 while the quantification on Figure 4-e shows a decrease of p-SMAD3. Could the authors show a more representative blot corresponding to the quantification?

We thank the reviewer for her/his considerate suggestions. We have reperformed the quantification of figure 4d-e and extended figure 4d to address reviewers' concerns. While we removed local background signal from the analysis (which can be substantial during western blots on phosphorylated proteins) we also analyzed and plotted all time points. Furthermore, we have selected another representative western blot for extended figure 4d.

While our main conclusion is unchanged (less SMAD3 phosphorylation at 45min peak), we have toned down our statement and removed “reduced and delayed” while centering the conclusion around pSMAD3:

We conducted pulse-chase experiments with recombinant TGFβ1 on MYOF^{KD} myCAFs and assessed TGFβ signal transduction kinetics (≤60 min) via c-terminal SMAD2/3 phosphorylation. Upon exogenous stimulation, TGFβ response was impaired in MYOF^{KD} myCAFs, as highlighted by reduced SMAD3 phosphorylation, particularly at the phosphorylation peak of 45 minutes (Fig. 4d-e, Extended Data Fig. 4d-f).

Myoferlin depletion as source of variation $p < 0.0001$

Myoferlin depletion as source of variation $p = 0.0001$

Myoferlin depletion as source of variation $p = 0.0281$

- In figure 5-c and n Golgin-97 was used as a Golgi marker. However, this is a trans-Golgi marker. Using an early Golgi marker such as GM130 would be more in accordance with the COP2 vesicles function at the interface of ER and Golgi. Moreover, staining with an ER marker would be useful to determine if in MYOF-KD cells SEC24C and TGFBR1 are retained in the ER. This is essential to support the claim that the ER to Golgi transport of those proteins is affected. Finally, in 5-c the zoom from the shMYOF2 condition is not taken

from a representative cell of the field of view as most cells there seem to still have a perinuclear pool of SEC24C. Is this heterogeneity due do cells having different MYOF levels after KD?

We thank reviewer for the important point she/he raises. GM130 represents indeed an interesting candidate to assess the fusion of TGFBR1 vesicles with the Golgi. The assessment of an ER retention of TGFBR1 vesicles using an ER marker also represents an exciting idea.

To address the reviewers' comment we have performed a quadruple IF (quad-IF) staining: Nuclei (grey), TGFBR1 (green), GM130 (red), KDEL (blue). Next, we have quantified the amount of TGFBR1 objects that colocalized with GM130 (Golgi) or with KDEL (ER) objects. First of all, we validated our findings that were made with Golgin-97, as we observed reduced colocalization of TGFBR1 vesicles with GM130-positive objects, suggesting an impaired fusion of TGFBR1 vesicles with the Golgi. However, we did not observe an increased colocalization of TGFBR1 vesicles with KDEL-positive objects (ER), which would have suggested a retention of TGFBR1 vesicles in the ER. Of note, even BFA treatment did not induce a retention of TGFBR1 vesicles at the ER. Altogether, these results support the idea that myoferlin is involved in membrane fusion and thus important for the fusion of TGFBR1 vesicles with the Golgi, while myoferlin does not seem to be part of the budding-machinery of TGFBR1 vesicles from the ER. These novel results have been added into the manuscript as figure 5p:

Incomplete shuttling of TGFBR1 to the Golgi apparatus was further validated using an early Golgi marker (GM130) via quad-IF staining, while a retention of TGFBR1 on the ER (KDEL) was undetectable (Fig. 5p), implying a potential role of myoferlin during the fusion of TGFBR1-carrying COP2 vesicles with the Golgi apparatus rather than during the budding from the ER.

Regarding the zoom in the shMYOF#2 condition of figure 5c, we have selected a more representative cell with a perinuclear pool of Sec24C. We think that minor

differences across cells are simply a reflection of biology rather than differences in *MYOF*^{KD}. Transduced CAFs are selected after every thawing (\pm monthly) using puromycin, thus minimizing a potential contamination of *MYOF*^{WT} CAFs.

- While the PLA experiments convincingly demonstrate that the proteins interact, the images in figures 5-e and j do not indicate in which cellular compartment they interact. The dots seem to be present all over the cells and not just in the perinuclear region where the proteins are shown to colocalize. Is it possible that they interact in the ER? Having immunofluorescent staining of Golgi and ER markers in cells where the PLA is performed and quantify the dots inside and outside those structures would be helpful to confirm where the proteins interact.

Once again, reviewer raised an intriguing point. We have reperformed both PLA experiments (*MYOF*/*Sec24C* and *MYOF*/*TGFBR1*) while adding fluorescent antibodies against the ER (*CALR*) and the Golgi (*GM130*). On one hand we did observe some PLA dots in the Golgi area, while others on the other hand were closer to the ER or cytosolic. This aligns with the volatile nature of *COP2* vesicles, shuttling between the ER and the Golgi within the cytosol. We believe that it is realistic to see some vesicles close to the ER or the Golgi, but also some vesicles “in transit”. Our results support the hypothesis of a general presence of myoferlin on *COP2* vesicles and not only at the ER exit sites or Golgi fusion sites. However, these experiments did not increase our knowledge regarding a precise localization of PLA dots, they were not included in the manuscript.

5. Some experiments are missing to fully support the conclusion that TGFBR1 trafficking from ER to Golgi is impaired. At least staining of an ER marker to show that TGFBR1 is retained there in MYOF-KD cells is necessary. In addition or alternatively, is it possible to assess the presence of high-mannose glycan chains on TGFBR1 that would indicate that it is not processed in the Golgi with an EndoH/PGNase digestion assay?

Indeed, we have assessed a potential retention of TGFBR1 on the ER via IF confocal microscopy. We did not observe an increased colocalization of TGFBR1 vesicles with KDEL (ER), which would have suggested a retention of TGFBR1 vesicles in the ER. Of note, even BFA treatment did not induce a retention of TGFBR1 vesicles at the ER. These results support the idea that myoferlin is involved in membrane fusion and thus important for the fusion of TGFBR1 vesicles with the Golgi, while myoferlin does not seem to be part of the budding-machinery of TGFBR1 vesicles from the ER. In other words, it seems that under $MYOF^{KD}$ TGFBR1 vesicles are still able to leave the ER but are unable to fuse with the Golgi. This important detail is also depicted in the graphical abstract of our manuscript. While the assessment of protein glycosylation upon $MYOF^{KD}$ represents an exciting thought, we believe that this deserves an entire manuscript. The biological

implication of altering the glycosylome can be major and a potential link with myoferlin is indeed possible. As we have addressed reviewers concerns via a quad-IF staining we hope that a PGNase digestion assay is not necessary in this situation. However, we hope to be able to investigate the impact of *MYOF*^{KD} on protein glycosylation in the future.

Minor concerns that should be addressed

1. In Figure 3-f a control for cell contamination of the conditioned media is missing. Those blots should show a protein that should be present in whole cell lysates but not in the CM such as GAPDH, with additional loading of a cell lysate, to ensure that the CM is free of signal that might come from cells or cell debris that are not fully removed by centrifugation.

Reviewer raised a relevant point. We have reperformed a western blot on the same samples that were used in figure 3f, while including a cell lysate as control, and probed for GAPDH. While GAPDH was abundantly present in the control cell lysate, we could not detect any signal in the conditioned media samples. Thus, we are confident that our conclusions were not biased by a cellular contamination in the conditioned media samples.

2. In Figure 3-j I could not find in the methods nor in the figure legends how the myCAFs nuclei is stained to be differentiated from the cancer cells.

Indeed, we missed to highlight this information. In fact, we received CAF01 from our collaborator already transduced with a nuclear red fluorescent protein (mKate). This enabled us to easily distinguish between CAFs and cancer cells in coculture models. We have amended this in the figure 3j legend and in methods section:

Immortalized human pancreatic stellate cells (PSCs) [HPSC127 (male IPMC, CAF01 hereafter, mKate red-fluorescent nucleus), HPSC21 (female PAAD, CAF02 hereafter), HPSC128 (male PAAD) and HPSC130 (female PAAD)] were established at Division of Gastroenterology, Tohoku University Graduate School of Medicine, Japan.

Fig. 3j) *Immunofluorescence microscopy of heterotopic spheroids (n≥10) generated from PAAD cells (PANC-1) in coculture with mKate red-fluorescent CTRL myCAFs (CAF01, Irrel. shRNA) or MYOF^{KD} myCAFs (CAF01, shMYOF#1 and shMYOF#2). Representative pictures are shown. Nuclei (DAPI) = blue, collagen 1 = green, myCAF nucleus = red, scale bar = 100µm.*

3. Figure 4-b shows only p-SMAD2 in unstimulated cells, what about p-SMAD3? Additionally, showing p-SMAD2 bands with a higher contrast/higher exposure time would allow to see them more easily.

SMAD2/3 phosphorylation is indeed hard to assess via western blot in non-TGFβ-treated conditions, hence why we did not obtain a specific signal for pSMAD3 (data not shown) and only a weak signal for pSMAD2. A better readout for TGFβ signal transduction in untreated cells is the nuclear translocation of SMAD2/3 via confocal microscopy. Accordingly, we found reduced nuclear translocation of SMAD2/3 upon MYOF KD in the absence of exogenous TGFβ (Fig. 4f-g).

4. The choice of green and red colors to show immunofluorescence images is not ideal for color blind people. I would advise to use a different combination of colors, such as green and magenta, to ensure that everyone is able to see the different markers on those images.

We agree with reviewers' concern and apologize to have omitted this aspect. However, due to the large number of images in this manuscript that would have to be altered (in addition to the complexity of quad-IF), we leave this comment to the editors' discretion.

5. In Figure 5-f and k please define $Tm1$ and $Tm2$.

$tM1$ and $tM2$ are Mander's coefficients after threshold regression. In other words, pixels with low intensity (below threshold) are excluded from the analysis of coefficients to minimize biases. We highlighted this also in the figure legend of figure 5:

f) Quantification of colocalization ($tM1$ and $tM2$ Mander's coefficients after threshold regression) shown in Fig. 5d and Extended Data Fig. 5d. Violin plot.

k) Quantification of colocalization ($tM1$ and $tM2$ Mander's coefficients after threshold regression) shown in Fig. 5i and Extended Data Fig. 5e. Violin plot.

6. The authors use the immunoprecipitation experiment in Figure 5-h to show that MYOF and SEC24C directly interact. However, this is not sufficient to prove a direct interaction as it does not allow to exclude an indirect interaction by being part of the same complex. If the authors really want to prove a direct interaction, they should perform immunoprecipitation of recombinant proteins or at least change the text to not claim a direct interaction from the co-immunoprecipitation. Moreover, the legend on this figure does not indicate if this is a representative image of several replicates or if this has been done only once.

Reviewer raised a valid point. We have toned down our statement in the results section regarding a direct interaction between myoferlin and Sec24C.

Finally, non-crosslink coimmunoprecipitation experiments of myoferlin confirmed that myoferlin and Sec24C are part of a same complex, while a close interaction with Sec31 was imperceptible (Fig. 5h). Altogether, these results strongly suggest a direct function of myoferlin in COP2 vesicle trafficking likely mediated through Sec24C.

Finally, we have amended the figure legend, as the WB shown is indeed one representative from two. The second replicate was part of an optimization process for our co-IP (of note, 100mM NaCl was selected for further experiments). This second replicate WB has been added in the "uncropped WB" file.

h) Western blot analysis of myoferlin co-IP eluants in myCAF01. Input = total cell lysate. Irrelevant IgG was used as IP control. One representative western blot of two independent experiments is shown.

Fig 5h
- The authors conclude from Figures 6-a and b and Extended data figure 6 that MYOF has different main functions in CAFs and cancer cells. However, MYOF-KD CAFs and cancer cells both have a decrease of expression of COP2 and ER related genes, indicating that the MYOF function described here could be shared in tumor cells and CAFs, unlike its mitochondrial functions that are specific to tumor cells. I would suggest to re-write the conclusion here to be clearer and in accordance with the data shown.

We apologize that the description of these findings may have been confusing, we have clarified this section accordingly:

To understand whether the function of myoferlin in CAFs could be different to cancer cells, we performed transcriptomic profiling on MYOF^{KD} cancer cells (PANC1) and crossed the differentially expressed genes (DEGs) between MYOF^{KD} and CTRL cancer cells with DEGs from MYOF^{KD} CAFs (Fig. 6a). Overrepresentation analysis (ORA) of overlapping downregulated DEGs revealed that COP2 vesicles and ER-related genes are both downregulated in CAFs and cancer cells upon MYOF^{KD}, suggesting that a COP2-related function of myoferlin may be shared between CAFs and cancer cells (Fig. 6b). However, while the overlapping DEGs represent a majority of DEGs in CAFs (65.6%), they represent only a minority of DEGs in cancer cells (13.2%). Accordingly, ORA on cancer cell-specific DEGs showed a downregulation of mitochondria-related genes, in agreement with the described mitochondrial role of myoferlin in pancreatic cancer cells^{19,20} (Extended Data Fig. 6), suggesting that a COP2-related function of myoferlin is only minor in cancer cells. Taken together, this data supports a cell type specific function of myoferlin, as the main function of myoferlin in CAFs is to support COP2 vesicle trafficking, while cancer cell myoferlin is predominantly linked to mitochondrial fitness.

- Regarding the test of the effect of MYOF pharmacological inhibition on already established tumors, the authors say in the discussion that no significant reduction of tumor size was observed. This should be said in the

results part when concluding on the data. Is this conclusion made from measuring the pancreas weight in Figure 7-g?

Indeed, the conclusion highlighted by the reviewer was made based on the pancreas weight. We have added the conclusion in the results section and replaced “size” with “weight”.

WJ460 administration was well tolerated by mice (Extended Data Fig. 7g) and did not result in increased pancreas weight, in contrast to what has been previously observed with CAF ablation⁶⁵ (Fig. 7g, Extended Data Fig. 7f).

Any additional non-essential suggestions for improving the study (which will be at the author's/editor's discretion)

1. Will the whole data from the transcriptomic profiling of MYOF-KD CAFs and cancer cells be available publicly on a repository or together with the paper? This would be valuable for others to look into specific genes, related to vesicle trafficking or other functions.

Yes, the transcriptomic data will be published together with the paper as count tables.

2. Some phenotypes observed with the MYOF-KD cells are quite mild (reduction of SMAD2/3 phosphorylation and nuclear localization and loss of SEC24C Golgi localization). Would a KO result in stronger phenotypes?

A KO could indeed result in more pronounced phenotypes. We aim to include KO models for myoferlin in our future studies.

3. The authors wrote that they could not find good flowcytometry antibodies for human TGFBR1. I would suggest as an alternative to perform an immunofluorescence staining of TGFBR1 in unpermeabilized cells and to use a plasma membrane marker to assess its presence at the cell surface.

We have indeed tried to perform a staining on unpermeabilized cells. Upon fixation, cells were incubated with anti-TGFBR1 antibody (same that was used in IF microscopy), washed and then incubated with a FITC-conjugated secondary antibody. However, we were unable to obtain a clear signal shift (right side) when compared to the unstained control (left side).

4. Co-culture of MYOF-WT and KD CAFs with cancer cells in a model where the cancer cells are able to migrate, or in an invasion assay, would be a great addition to assess how MYOF expressed in CAF contributes to tumor cells migration and invasive abilities.

Reviewer raised a relevant point. We have a current student in the lab that has looked at exactly this question in the past. We have performed a transwell migration assay with *MYOF*^{KD} CAFs in the bottom chamber and *MYOF*^{WT} cancer cells (CFPAC1) in the top chamber. Overall, we did not observe an effect of *MYOF* KD in CAFs on the migratory potential of cancer cells.

Dear Olivier,

Thank you for submitting a revised version of your manuscript. We have now received input from all original reviewers, who are satisfied with the revisions and now recommend acceptance of the manuscript after final minor revisions. Additionally, there remain a few editorial points that need to be addressed before I can extend official acceptance of the manuscript:

1. Please submit a complete author checklist, which you can download from our author guidelines (<https://www.embopress.org/pb-assets/embo-site/EMBO%20Press%20Author%20Checklist-1642513524327.xlsx>). Please insert information in the checklist that is also reflected in the manuscript. The completed author checklist will also be part of the Review Process File.
2. Please rename the extended view figures into "Figure EV1" etc. Please place the figure legends after the main figure legends and under the heading "Expanded View Figure Legends".
3. Please rename the heading "Main" to "Introduction".
4. Please merge "Funding" section" with "Acknowledgments".
5. Please remove the disclaimer for BioRender from "Acknowledgments" and move to the "Methods" section, using following the following format:
Graphics:
(some of the... OR Figure #... OR synopsis) Graphics were created with BioRender.com.
6. Please compile the information included in Extended Tables 1 and 2 into a Reagents and Tools Table. Please use the template (.docx), which you can find in our author guidelines:
<https://www.embopress.org/page/journal/14602075/authorguide#structuredmethods>
When submitting your revised manuscript, please upload it as a separate file choosing the file type "Reagent Table".
An example of a Method paper with Structured Methods can be found here:
<https://www.embopress.org/doi/10.15252/msb.20178071>.
7. Datasets and computer code that were generated in the reported study should be listed in a structured manner in the "Data Availability" section placed after the Methods section. If your study does not include datasets, please insert the following statement: "This study includes no data deposited in external repositories". More information about the format of this section can be found here: <https://www.embopress.org/page/journal/14602075/authorguide#dataavailability>.
8. Please update references according to The EMBO Journal style - it should be alphabetical. Please also remove the DOIs. Please see further information here: <https://www.embopress.org/page/journal/14602075/authorguide#referencesformat>
9. In our standard image check, we have found that microscopy and blot panels across the entire figure set appear pixelated under image analysis. This is a common result of converting original 16-bit TIFF images to RGB format for publication. While not a direct cause for integrity concern, it can give the impression of image alteration to critical readers.
To avoid any misunderstanding and to ensure compliance with EMBO Press standards, please resubmit the complete figure set at its original data resolution, preferably in TIFF format, to preserve image quality.
10. Our data editors have flagged the following issues in figure legends that need correcting:
 - Please provide the exact p values in the legends of figures 1F-H; 2B, F; 3C, E, I; 4G, J; 5C, G, I, M, O, 6D, extended data figures 1A, C; 2I; 4F, H; 5B, 7E.
 - Please indicate the statistical test used for data analysis in the legends of figures 4A, 5A.
 - Please define the box plots in terms of minima, maxima, centre, bounds of box and whiskers, and percentile in the legends of extended data figures 1A, 5B.
 - Please provide information on the number and nature of replicates in the legends of figures 2B, C, D, F, H.
11. Papers published in The EMBO Journal are accompanied online by a 'Synopsis' to enhance discoverability of the manuscript. It consists of A) a short (1-2 sentences) summary of the findings and their significance, B) 3-4 bullet points highlighting key results and C) a synopsis image that is 550x300-600 pixels large (width x height, jpeg or png format). You can either show a model or key data in the synopsis image. Please note that the image size is rather small and that text needs to be readable at the final size.

With kind regards,

Ieva

Ieva Gailite, PhD
Senior Scientific Editor
The EMBO Journal

Meyerhofstrasse 1
D-69117 Heidelberg
Tel: +4962218891309
i.gailite@embojournal.org

We realize that it is difficult to revise to a specific deadline. In the interest of protecting the conceptual advance provided by the work, we recommend a revision within 3 months (24th Nov 2025). Please discuss the revision progress ahead of this time with the editor if you require more time to complete the revisions. Use the link below to submit your revision:

Referee #1:

I appreciate the efforts and additions Peiffer et al. had made to this work, it has certainly improved the overall message of their work and cleared some questions we raised. The addition of the MYOF KO in-vivo model help highlight the role of MYOF in the TME, without over-stating its' effects on overall tumour progression. The paper now shows a very coherent mechanism of action on MYOF in fibroblasts, and as the authors mentioned in their discussion, leading to further questions on the effect this might have in combination with chemotherapies or ICB treatment, as they did show a significant reduction in ECM but no significant effect on tumour burden as a stand-alone factor.

Some minor comments -

Figure 2E - you show krt18 as an epithelial marker but it's also a mesothelial marker (as you can see in extended figure 2e), which express high levels of MYOF as seen in extended figure 2f. Consider changing the marker used for epithelial cells.

Extended figure 2e - it's unclear what are the corresponding clusters from extended figure 2d to cluster numbers on this figure, please add them to the legend or change the cluster numbers to cluster names.

Extended figure 2g - please add gene name to y axis legend (as done in all other similar figures)

Referee #2:

The authors have done a commendable job at addressing prior concerns. The addition of studies in the Myof-null mice significantly strengthens the manuscript.

The manuscript presents compelling data that support the overall conclusions of the study.

I only have a couple of minor issues that could be dealt with editorially.

1. the term 'stromal/stroma aggressiveness' is not definable and as such should not be used in my opinion. I think it would increase clarity to indicate what the characteristic that has changed, such as ECM deposition or an attribute that is definable.

2. Please define the genotype of KPC used in this study. What is the p53 mutation, what is the cre.

3. Please clarify if metastatic disease was present and quantified in the orthotopic tumor studies and whether there was a difference in metastatic burden in tumors grown in WT vs Myof-null mice.

Referee #3:

The authors made a substantial effort to improve the manuscript and answer the reviewers' concerns. The new in vivo data are a significant improvement and show convincingly the relevance of Myoferlin function in fibroblasts in a tumor in vivo. All my concerns were carefully addressed and I appreciate the efforts to have performed many new experiments that strengthen the conclusions. However, I still have a couple of concerns to be addressed before publication of the manuscript:

Major concerns:

The new extended Figure 5j and the corresponding text modification help to answer to reviewer 1's concern, however given the very small change for Shc phosphorylation, which does not seem to be much phosphorylated in the control condition, I don't think it is correct to say that "5 out of 7 TGFBR1 phosphorylation targets were less phosphorylated in MYOFKD 421 myCAFs when 422 compared to CTRL myCAFs". Moreover, the authors themselves wrote in the response letter that "the two borderline reduced phosphorylation sites are very poorly phosphorylated in both conditions, suggesting that TGFBR1 likely not signals

through Shc in our myCAF model." Therefore, the text needs to be modified to not claim that Shc is less phosphorylated in the shMYOF condition.

On the same figure, SMAD3 phosphorylation is not impacted by the shMYOF, which goes against the observations from Figure 4 that SMAD3 phosphorylation is reduced in MYOF KD cells. This questions the validity of those experiments and is an important concern to be addressed or at least discussed.

Additional suggestion:

In Figure 5P the KDEL staining in blue over black background is difficult to see, another color showing more contrast with the background would make it easier to appreciate the staining.

Reviewing report (August 2025)

To the editors

We thank the editors, and especially Dr Gailite, for the efficient handling of our manuscript. We appreciate the last comments they made on the manuscript and tried to address them (see green annotations in the manuscript).

- 1 – Author checklist was completed and included in the submission.
- 2 – Supplemental figures were renamed as EV figure, and legends were added after the main legend under the header “Expanded View Figure Legends”.
- 3 – The heading “main” was change to “Introduction”.
- 4 – Acknowledgements and funding sections were merged.
- 5 – Biorender.com was moved out of the acknowledgements and added to the methods section under the heading “Graphics”.
- 6 – Extended table 2 was adapted to the “Reagents and Tools Table” template. However, Extended table 1 was kept as a separate table but transform as a level 3 appendix.
- 7 – RNAseq dataset were submitted to Gene Expression Omnibus and the data availability section was adapted consequently.
- 8 – Reference list was adapted using the EMBO Journal style for Zotero.
- 9 – Figures and extended view figures were submitted as TIFF
- 10 – All changes requested by the data editors have been made, except for the addition of exact p-values for Figures EV1A and EV5B, which were produced by the online tool GEPIA2. Unfortunately, this tool does not calculate the exact p-values.
- 11 – We add a synopsis of the manuscript, including a small image.

In addition, and at the discretion of the editors, we suggest replacing the abbreviation "PAAD" in the title with "pancreatic cancer" to improve the discoverability of the manuscript.

To the reviewers

Once again, we thank all 3 reviewers for taking the time to read and review our manuscript “Targeting myoferlin in ER/Golgi vesicle trafficking reprograms cancer-associated fibroblasts in PAAD” for The EMBO Journal.

We appreciate the positive feedback regarding our revised manuscript and value their attention to detail. We have addressed all remaining comments (see modifications in red in the manuscript) and hope that the reviewers are satisfied with this final version of our manuscript.

Reviewer 1

I appreciate the efforts and additions Peiffer et al. had made to this work, it has certainly improved the overall message of their work and cleared some questions we raised. The addition of the MYOF KO in-vivo model help highlight the role of MYOF in the TME, without over-stating its' effects on overall tumour progression. The paper now shows a very coherent mechanism of action on MYOF in fibroblasts, and as the authors mentioned in their discussion, leading to further questions on the effect this might have in combination with chemotherapies or ICB treatment, as they did show a significant reduction in ECM but no significant effect on tumour burden as a stand-alone factor.

We thank reviewer 1 for his/her positive feedback.

Minor comments

1. Figure 2E - you show *krt18* as an epithelial marker but it's also a mesothelial marker (as you can see in extended figure 2e), which express high levels of MYOF as seen in extended figure 2f. Consider changing the marker used for epithelial cells.

We thank the reviewer for their attention to detail, we have adjusted figure 2e accordingly and replaced *Krt18* with *Cpa1*, which is not expressed in mesothelial cells (Extended Data Fig. 2f). The low expression of *Cpa1* confirms the EPCAM+ depletion performed before scRNAseq analysis (Dominguez et al. 2020).

2. Extended figure 2e - it's unclear what are the corresponding clusters from extended figure 2d to cluster numbers on this figure, please add them to the legend or change the cluster numbers to cluster names.

Indeed, cluster annotations were missing in extended figure 2e, we have amended the figure accordingly and fitted the cluster colors to the ones showed in extended figure 2d.

3. Extended figure 2g - please add gene name to y axis legend (as done in all other similar figures).

The y axis label has been added as suggested.

Reviewer 2

The authors have done a commendable job at addressing prior concerns. The addition of studies in the Myof-null mice significantly strengthens the manuscript. The manuscript presents compelling data that support the overall conclusions of the study.

I only have a couple of minor issues that could be dealt with editorially.

We appreciate the reviewer's feedback and have addressed the remaining issues accordingly.

1. The term 'stromal/stroma aggressiveness' is not definable and as such should not be used in my opinion. I think it would increase clarity to indicate what the characteristic that has changed, such as ECM deposition or an attribute that is definable.

We thank the reviewer for his/her relevant comment. While the term “stromal aggressiveness” indeed lacks a precise definition, it had been chosen by us to account for ECM production, CAF contraction and CAF migration, which are all affected by myoferlin KO. Consequently, as our *in vivo* data only addressed tumor desmoplasia, we have removed stromal aggressiveness from the abstract, from the main text, from figure 6 and figure 7. However, to ensure that our graphical abstract addresses all key points from our manuscript, covering *in vitro* and *in vivo* findings, we did not remove the term “stromal aggressiveness” from figure 8.

2. Please define the genotype of KPC used in this study. What is the p53 mutation, what is the cre.

We have amended the KPC genotype in figure 7 and in the methods: *Mouse KPC cells (Kras^{G12D}; Trp53^{R172H}; Pdx1^{CRE}) were a gift from Dr Van Audenaerde (UAntwerp, Belgium).*

3. Please clarify if metastatic disease was present and quantified in the orthotopic tumor studies and whether there was a difference in metastatic burden in tumors grown in WT vs Myof-null mice.

Metastases were macroscopically examined in the liver and lungs; none were found across all conditions. We have amended the text accordingly: *At sacrifice, no macroscopic metastases were visible (data not shown) and the pancreas weight was unaffected by stromal Myof^{KO} (Fig. 7b).*

Reviewer 3

The authors made a substantial effort to improve the manuscript and answer the reviewers' concerns. The new *in vivo* data are a significant improvement and show convincingly the relevance of Myoferlin function in fibroblasts in a tumor *in vivo*. All my concerns were carefully addressed and I appreciate the efforts to have performed many new experiments that strengthen the conclusions. However, I still have a couple of concerns to be addressed before publication of the manuscript.

We thank the reviewer for his/her comforting feedback and appreciate the recognition of our additional data. We have addressed all final concerns accordingly.

1. The new extended Figure 5j and the corresponding text modification help to answer to reviewer 1's concern, however given the very small change for Shc phosphorylation, which does not seem to be much phosphorylated in the control condition, I don't think it is correct to say that "5 out of 7 TGFBR1

phosphorylation targets were less phosphorylated in MYOFKD myCAFs when compared to CTRL myCAFs". Moreover, the authors themselves wrote in the response letter that "the two borderline reduced phosphorylation sites are very poorly phosphorylated in both conditions, suggesting that TGFBR1 likely not signals through Shc in our myCAF model." Therefore, the text needs to be modified to not claim that Shc is less phosphorylated in the shMYOF condition.

On the same figure, SMAD3 phosphorylation is not impacted by the shMYOF, which goes against the observations from Figure 4 that SMAD3 phosphorylation is reduced in MYOF KD cells. This questions the validity of those experiments and is an important concern to be addressed or at least discussed.

Indeed, reviewer raised a relevant point. We have adjusted our conclusion for figure 5j as follows: *Furthermore, using a phospho-antibody array in combination with TGFβ stimulation, we found 3 out of 5 TGFBR1 phosphorylation targets⁶⁰ were less phosphorylated in MYOF^{KD} myCAFs when compared to CTRL myCAFs (Extended Data Fig. 5i-j), excluding Shc which is unphosphorylated in both conditions.*

Regarding the SMAD3 discrepancy, TGFBR1 phosphorylates SMAD3 on S423 and S425, both relevant for the transcriptional activity of the SMAD complex. While the antibody used in figure 4 indeed captures both phosphorylation sites, the antibody used in the phospho-array from extended figure 5 only targets S425. This may explain the discrepancy raised by reviewer. We have addressed this issue in the manuscript: *Surprisingly, SMAD3 S425 phosphorylation was not reduced upon MYOF^{KD} as reported earlier (Fig. 4d). This discrepancy can be explained by the simultaneous detection of S423 and S425 phosphorylation sites by western blot (Fig. 4d), while only S425 phosphorylation was evaluated in the phospho-antibody array (Extended Data Fig. 5i-j).*

2. In Figure 5P the KDEL staining in blue over black background is difficult to see, another color showing more contrast with the background would make it easier to appreciate the staining.

We thank the reviewer for this careful observation. We have changed the color attribution to improve the visibility of the KDEL staining.

Dear Olivier,

Thank you for addressing the final editorial requests. I am now pleased to inform you that your manuscript has been accepted for publication - congratulations!

Before we forward your manuscript to our publishers, we would like to propose some minor edits in the manuscript title, abstract and synopsis, which you can find in the attached text file. I have also written a short blurb that will accompany the title of your manuscript in our online table of contents. Please take a look and let me know if any corrections or adjustments are needed.

If you have any questions, please do not hesitate to contact the Editorial Office. Thank you for this contribution to The EMBO Journal and congratulations on a nice study!

With best wishes,

Ieva

Ieva Gailite, PhD
Senior Scientific Editor
The EMBO Journal
Meyerohofstrasse 1
D-69117 Heidelberg
Tel: +4962218891309
i.gailite@embojournal.org
